# Learning Orthogonal Multi-Index Models: A Fine-Grained Information Exponent Analysis

**Yunwei Ren**
Princeton University
yunwei.ren@princeton.edu

**Jason D. Lee**
Princeton University
jasonlee@princeton.edu

## Abstract

The information exponent ([BAGJ21]) and its extensions — which are equivalent to the lowest degree in the Hermite expansion of the link function (after a potential label transform) for Gaussian single-index models — have played an important role in predicting the sample complexity of online stochastic gradient descent (SGD) in various learning tasks. In this work, we demonstrate that, for multi-index models, focusing solely on the lowest degree can miss key structural details of the model and result in suboptimal rates.

Specifically, we consider the task of learning target functions of form $f_*(\boldsymbol{x}) = \sum_{k=1}^{P} \phi(\boldsymbol{v}_k^* \cdot \boldsymbol{x})$, where $P \ll d$, the ground-truth directions $\{\boldsymbol{v}_k^*\}_{k=1}^{P}$ are orthonormal, and the information exponent of $\phi$ is $L$. Based on the theory of information exponent, when $L = 2$, only the relevant subspace (not the exact directions) can be recovered due to the rotational invariance of the second-order terms, and when $L > 2$, recovering the directions using online SGD require $\tilde{O}(Pd^{L-1})$ samples. In this work, we show that by considering both second- and higher-order terms, we can first learn the relevant space using the second-order terms, and then the exact directions using the higher-order terms, and the overall sample and complexity of online SGD is $\tilde{O}(dP^{L-1})$.

## 1 Introduction

In many learning problems, the target function exhibits or is assumed to exhibit a low-dimensional structure. A classical model of this type is the multi-index model, where the target function depends only on a $P$-dimensional subspace of the ambient space $\mathbb{R}^d$, with $P$ typically much smaller than $d$. When the relevant dimension $P = 1$, the model is known as the single-index model, which dates back to at least [Ich93]. Both single- and multi-index models have been widely studied, especially in the context of neural network and stochastic gradient descent (SGD) in recent years, sometimes under the name "feature learning" [BAGJ21, BBSS22, DLS22, AAM22, AAM23, DKPS24, DPVLB24, OSSW24, DTA+24].

In [BAGJ21], the authors show that for single-index models, the behavior of online SGD can be split into two phases: an initial "searching" phase, where most of the samples are used to boost the correlation with the relevant (one-dimensional) subspace to a constant, and a subsequent "descending" phase, where the correlation further increases to $1$. They introduce the concept of the information exponent (IE), defined as the index of the first nonzero coefficient in the Taylor expansion of the population loss around $0$, which also corresponds to the lowest degree in the Hermite expansion of the link function in Gaussian single-index models. They prove that the sample complexity of online SGD is $\tilde{O}(d^{(\mathrm{IE}-1)\vee 1})$. After that, various lower and upper bounds have been established for single-index models in [BBSS22, DNGL23, DPVLB24]. Similar results for certain multi-index models have also been derived in [AAM22, AAM23, BBPV23, OSSW24]. In all cases, the sample complexity of online SGD scales with $d^{\mathrm{IE}-1}$ when $\mathrm{IE} \geq 3$.

39th Conference on Neural Information Processing Systems (NeurIPS 2025).

Later, it was realized that the notion of information exponent is not stable under modifications of the algorithm. In particular, the information exponent of a link function can be greatly reduced by reusing batches or applying a suitable label transformation [ADK$^+$24, DTA$^+$24, LOSW24, DPVLB24]. For example, the IE of any fixed degree polynomial can be reduced to at most 2 via monomial transformations. This observation leads to the notion of generative exponent (GE) [DPVLB24], which is defined as the lowest information exponent among all $L^2$ transform of the original link function. It yields bounds that match the previous results for non-gradient-based methods [CM20, TDD$^+$24, BKM$^+$19]. Despite the improvement over the vanilla information exponent, in the framework of generative exponents, still only the lowest order is considered. As we will discuss later, this makes it suffer from the same issue of information exponent in the context of multi-index models.

Consider multi-index models of form $f_*(\boldsymbol{x}) = \sum_{k=1}^{P} \phi(\boldsymbol{v}_k^* \cdot \boldsymbol{x})$, where $\{\boldsymbol{v}_k^*\}_k$ are orthonormal vectors. In this setting, there are two types of recovery: recovering each direction $\boldsymbol{v}_k^*$ and recovering the subspace spanned by $\{\boldsymbol{v}_k^*\}_k$. The former notion is stronger, and once the directions are learned, the learning task essentially reduces to learning the one-dimensional $\phi : \mathbb{R} \to \mathbb{R}$. However, directional recovery is not always possible. To see this, consider the case $\phi(z) = h_2(z)$, where $h_l$ is the $l$-th (normalized) Hermite polynomial. One can show that this corresponds to decomposing the projection matrix (a second-order tensor) of the subspace $\mathrm{span}\{\boldsymbol{v}_k^*\}_k$. Hence, recovering the directions is impossible due to the rotational invariance (see Section 3.1 for more discussion). In other words, in any framework that considers only the lowest order (IE or GE), if the lowest order is 2,[1] we cannot get guarantees beyond subspace recovery due to the existence of the $\phi = h_2$ example.

On the other hand, if $\phi$ contains some higher-order terms, e.g., $\phi = h_2 + h_4$, then one should expect that directional convergence is possible, even though $\mathrm{IE}(\phi)$ is still 2, because of identifiability property of (higher-order) orthogonal tensor decomposition problem [GLM18, LMZ20, GRWZ21]. In addition, we should be able to first recover the subspace using the second-order terms, which should require $\tilde{O}(d\,\mathrm{poly}\,P)$ samples, and then recover the directions using the higher-order terms. Moreover, since we have learned the relevant subspace, the number of samples needed in the second step should be much smaller than what is needed if there were no second-order terms.

In this work, we formally prove the above conjecture and show that the overall sample complexity is $\tilde{O}(dP^{L-1})$, where $L$ is the (lowest) order of the higher-order terms. Note that this bound scales linearly with the ambient dimension $d$, and it matches the sample complexity of separately learning $P$ independent single-index models with the relevant subspace known but the noise scales with $d$, up to potential logarithmic terms. More formally, we prove the following theorem.

**Theorem 1.1** (Informal version of Theorem 2.1). *Suppose that the target function is given by* $f_*(\boldsymbol{x}) = \sum_{k=1}^{P} \phi(\boldsymbol{v}_k^* \cdot \boldsymbol{x})$ *where* $\phi = \hat{\phi}_2 h_2 + \sum_{l=L}^{\infty} \hat{\phi}_l h_l$, *with* $L \geq 3$, $\hat{\phi}_2^2, \hat{\phi}_L^2 > 0$, *and* $\{\boldsymbol{v}_k^*\}_{k=1}^{P}$ *are orthonormal, and the input* $\boldsymbol{x}$ *follows the standard Gaussian distribution* $\mathcal{N}(0, \boldsymbol{I}_d)$. *Then, we can use online SGD (followed by a ridge regression step) to train a two-layer network of width* $\tilde{O}(P)$ *to learn (with high probability) this target function using* $\tilde{O}(dP^{L-1})$ *samples and steps.*

**Organization**  The rest of the paper is organized as follows. First, we review the related works and summarize our contributions. Then, we describe the detailed setting and state the formal version of the main theorem in Section 2. In Section 3, we discuss the easier case where the training algorithm is population gradient flow. Then, in Section 4, we show how to convert the gradient flow analysis to an online SGD one. Finally, we conclude and discuss the limitations in Section 5. The proofs, simulation results (Appendix F), and a table of contents can be found in the appendix.

## 1.1   Related work

In this subsection, we discuss works that are directly related to ours or were not covered earlier in the introduction.

Along the line of information exponent, the paper most related to ours is [OSSW24]. They show that for near orthogonal multi-index models, the sample complexity of recovering all ground-truth directions using online SGD is $\tilde{O}(Pd^{\mathrm{IE}-1})$ when $\mathrm{IE} \geq 3$. Their results do not apply to the case

---

[1]The IE = 2 case is particularly important as the information exponent of many functions, including all fixed degree polynomials, can be reduced to at most 2 by a suitable label transform [DPVLB24], and the information exponent of any even functions is at least 2.

IE $= 2$ for the reason we have discussed earlier. They propose first removing the second-order terms using the technique in [DLS22, DKPS24], which requires $d^2$ samples. Our result considers the situation where both the second and $L$-th order terms are present and show that in this case, the sample complexity of online SGD (without any preprocessing) is $\tilde{O}(dP^{L-1})$.

Another recent related work is [BAGP24]. Our main results are not directly comparable since the settings are different. They run SGD on the Stiefel manifold, which automatically prevents the model from collapsing to a single direction, but allow the target model to have condition number larger than $1$. In addition, only the lowest degree is considered in their work. However, they also show (in their setting) that when the second order term is isotropic, the subspace and only the subspace can be recovered. A similar idea is also used in our analysis of Stage 1.1 (cf. Section 3.1).

Another related line of research is learning two-layer networks in the teacher-student setting ([ZSJ$^{+}$17, LY17, Tia17, LMZ20, ZGJ21, GRWZ21]). Among them, the ones most relevant to this work are [LMZ20] and the follow-up [GRWZ21], both of which consider orthogonal models similar to ours and use similar ideas in the analysis of the population process. However, they do not assume a low-dimensional structure and only provide very crude $\text{poly}(d)$-style sample complexity bounds.

## 1.2 Our contributions

We summarize our contributions as follows:

- We demonstrate that information/generative exponent alone is insufficient to characterize certain structures in the learning task and show that for a specific orthogonal multi-index model, if we consider both the lower- and higher-order terms, the sample complexity of directional recovery using online SGD can be greatly improved over the vanilla information exponent-based analysis.

- As a by-product, we derive a collection of user-friendly technical lemmas to analyze the difference between noisy one-dimensional processes and their deterministic counterparts, which may be of independent interests (cf. Section 4.1 and Appendix E).

## 2 Setup and main result

In this section, we describe the setting of our learning task and the training algorithm, and then formally state our main result. We will also convert the problem to an orthogonal tensor decomposition task using the standard Hermite argument as in [GLM18].

**Notations**    We use $\|\cdot\|_p$ to denote the $p$-norm of a vector. When $p = 2$, we often drop the subscript and simply write $\|\cdot\|$. For $a, b, \delta \in \mathbb{R}$, $a = b \pm \delta$ means $|a - b| \le |\delta|$ and $a \vee b = \max\{a, b\}$ and $a \wedge b = \min\{a, b\}$. Beside the standard asymptotic (big $O$) notations, we also use the notation $f_d = O_\phi(g_d)$, which means there exists a constant $C_\phi > 0$ that can depend only on $\phi$ such that $f_d \le C_\phi g_d$ for all large enough $d$. Sometimes we also write $f_d \lesssim_\phi g_d$ for $f_d = O_\phi(g_d)$. The actual value of $C_\phi$ can vary between lines.

### 2.1 Input and target function

We assume the input $\boldsymbol{x}$ follows the standard Gaussian distribution $\mathcal{N}(0, \boldsymbol{I}_d)$ and the target function has form $f_*(\boldsymbol{x}) = \sum_{k=1}^{P} \phi(\boldsymbol{v}_k^* \cdot \boldsymbol{x})$, where $\log^C d \le P \le d$ for a large universal constant $C > 0$, $\{\boldsymbol{v}_k^*\}_{k=1}^{P}$ are orthonormal and $\phi : \mathbb{R} \to \mathbb{R}$ is the link function. In addition, we assume $\phi$ satisfies the following.

**Assumption 1** (Assumptions on the link function). *Let $h_k$ denote the degree-$k$ normalized Hermite polynomial and $\phi = \sum_{l=0}^{\infty} \hat{\phi}_k h_k$ denote the Hermite expansion of $\phi \in L^2(\mathcal{N}(0, \boldsymbol{I}_d))$.*

*(a)* **(IE structure)** *For some constant $L > 2$, $\phi(z) = \hat{\phi}_2 h_2(z) + \hat{\phi}_L h_L(z) + \sum_{l > L} \hat{\phi}_l h_l(z)$.*

*(b)* **(IE regularity)** *$\hat{\phi}_2, \hat{\phi}_L = \Omega(1)$ and $\|\phi'\|_{L^2}^2 = \sum_{l=1}^{\infty} l\hat{\phi}_l^2 \le C_\phi^2$ for some constant $C_\phi > 0$.*

*(c)* **(Polynomial growth)** *There exists universal constants $C, q > 0$ such that $|\phi(x)| \vee |\phi'(x)| \le C(1 + x^2)^{q/2}$ for all $x \in \mathbb{R}$.*

Our target model and algorithm will all be invariant under rotation. Hence, we will assume w.l.o.g. that $\boldsymbol{v}_k^* = \boldsymbol{e}_k$ where $\{\boldsymbol{e}_k\}_k$ is the standard basis of $\mathbb{R}^d$.

## 2.2 Learner model and the training algorithm

Our learner model is a width-$m$ two-layer network

$$f(\boldsymbol{x}) := f(\boldsymbol{x}; \boldsymbol{a}, \boldsymbol{V}) := \sum_{i=1}^m a_i \phi(\boldsymbol{v}_i \cdot \boldsymbol{x}),$$

where $\boldsymbol{a} = (a_1, \ldots, a_m) \in \mathbb{R}^m$ and $\boldsymbol{V} = (\boldsymbol{v}_1, \ldots, \boldsymbol{v}_m) \in (\mathbb{S}^{d-1})^m$ are the trainable parameters. We call $\{\boldsymbol{v}_i\}_{i \in [m]}$ the first-layer neurons. We measure the difference between the learner and the target model using the correlation loss. Given a sample $(\boldsymbol{x}, f_*(\boldsymbol{x}))$, we define the per-sample and population MSE loss as

$$l_{\mathrm{MSE}}(\boldsymbol{x}) := l(\boldsymbol{x}; \boldsymbol{a}, \boldsymbol{V}) := \frac{1}{2} \left( f_*(\boldsymbol{x}) - f(\boldsymbol{x}; \boldsymbol{a}, \boldsymbol{V}) \right)^2, \quad \mathcal{L}_{\mathrm{MSE}}(\boldsymbol{a}, \boldsymbol{V}) := \mathbb{E}_{\boldsymbol{x}} \, l_{\mathrm{MSE}}(\boldsymbol{x}; \boldsymbol{a}, \boldsymbol{V}).$$

Now, we describe the training algorithm. First, we initialize each output weight $a_i$ to be 1. Then, we symmetrically initialize the first layer neurons. That is, for $i \in [m/2]$, we initialize $\boldsymbol{v}_i \sim \mathrm{Unif}(\mathbb{S}^{d-1})$ independently and set $\boldsymbol{v}_{m/2+i} = -\boldsymbol{v}_i$ for the other half of the neurons. After the initialization, we fix the output weights $\boldsymbol{a}$ and train the first-layer weight $\boldsymbol{v}_i$ using online (spherical) SGD with the correlation loss $l_{\mathrm{corr}}(\boldsymbol{x}) = -f_*(\boldsymbol{x})f(\boldsymbol{x})$ and step size $\eta > 0$ for $T$ iterations. Then, we fix the first-layer weights and use ridge regression to train the output weights $\boldsymbol{a}$.

Let $\{(\boldsymbol{x}_t, f_*(\boldsymbol{x}_t))\}_{t \in \mathbb{N}}$ be our samples where $\{\boldsymbol{x}_t\}$ are i.i.d. standard Gaussian vectors, and let $\tilde{\nabla}_{\boldsymbol{v}} = (\boldsymbol{I} - \boldsymbol{v}\boldsymbol{v}^\top)\nabla_{\boldsymbol{v}}$ denote the spherical gradient. Then, we can formally describe the training procedure as follows:

$$
\begin{array}{lll}
\text{Initialization:} & a_{0,i} = 1, \quad \boldsymbol{v}_{0,i} \overset{\text{i.i.d.}}{\sim} \mathrm{Unif}(\mathbb{S}^{d-1}), \quad \boldsymbol{v}_{0,m/2+i} = -\boldsymbol{v}_{0,i} & \forall i \in [m/2]; \\[2mm]
\text{Stage 1:} & \begin{cases} \hat{\boldsymbol{v}}_{t+1,i} = \boldsymbol{v}_{t,i} + \eta f_*(\boldsymbol{x}_t)\nabla_{\boldsymbol{v}_i}\phi(\boldsymbol{v}_i \cdot \boldsymbol{x}), \\ \boldsymbol{v}_{t+1,i} = \hat{\boldsymbol{v}}_{t+1,i} / \|\hat{\boldsymbol{v}}_{t+1,i}\|, \end{cases} & \forall i \in [m], t \in [T]; \\[4mm]
\text{Stage 2:} & \boldsymbol{a} = \underset{\boldsymbol{a}'}{\operatorname{argmin}} \, \frac{1}{2N} \sum_{n=1}^N l(\boldsymbol{x}_{T+n}; \boldsymbol{a}', \boldsymbol{V}_T) + \lambda \|\boldsymbol{a}'\|^2 . &
\end{array}
$$

$$(1)$$

Here, the hyperparameters are the network width $m > 0$, step size $\eta > 0$, time horizon $T > 0$, the number of samples $N$ in Stage 2, and the regularization strength $\lambda > 0$.

We will show that after the first stage, for each ground truth direction $\boldsymbol{v}_k^*$, there will be some neurons $\boldsymbol{v}_i$ that has converged to that direction. As a result, in the second stage, we can use ridge regression to pick out those neurons and use them to fit the target function. The analysis of this second stage is standard and has been done in [DLS22, AAM22, BES+22, LOSW24, OSSW24]. Hence, we will not further discuss this stage in the main text and defer the proofs of this stage to Appendix C.

For the gradient update in Stage 1, we have the following lemma on its expectation and tail. The proof of this lemma is rather standard and can be found in, for example, [GLM18, OSSW24]. We also provide a proof in Appendix A.1 for completeness.

**Lemma 2.1** (First-layer gradients). *Consider the setting described above. Suppose that $\phi$ satisfies Assumption 1 and $a_i = 1$ for all $i \in [m]$ and $\{\boldsymbol{v}_k^*\}_k$ are orthonormal. Then, for each $i \in [m]$, we have*

$$\mathbb{E}\left[f_*(\boldsymbol{x})\nabla_{\boldsymbol{v}_i}\phi(\boldsymbol{v}_i \cdot \boldsymbol{x})\right] = 2\hat{\phi}_2^2 \sum_{k=1}^P \langle \boldsymbol{v}_k^*, \boldsymbol{v}_i \rangle \boldsymbol{v}_k^* + \sum_{l \geq L} \sum_{k=1}^P l\hat{\phi}_l^2 \langle \boldsymbol{v}_k^*, \boldsymbol{v}_i \rangle^{l-1} \boldsymbol{v}_k^*. \tag{2}$$

*Then, for each fixed neuron $(a, \boldsymbol{v})$ and direction $\boldsymbol{u} \in \mathbb{S}^{d-1}$ that is independent of $\boldsymbol{x} \sim \mathcal{N}(0, \boldsymbol{I}_d)$, we have*

$$\mathbb{E}\langle f_*(\boldsymbol{x})\nabla_{\boldsymbol{v}_i}\phi(\boldsymbol{v}_i \cdot \boldsymbol{x}), \boldsymbol{u} \rangle^2 \lesssim_\phi P,$$

$$|\langle f_*(\boldsymbol{x})\nabla_{\boldsymbol{v}_i}\phi(\boldsymbol{v}_i \cdot \boldsymbol{x}), \boldsymbol{u} \rangle| \lesssim_\phi P^{1/2} \log^{2(1+q)} \log(m/\delta_\mathbb{P}) \quad \text{with probability at least } 1 - \delta_\mathbb{P}.$$

**Remark.** We say a random variable $X$ is $(\sigma^2, \theta)$-subweibull [VGNA20, KC22] if

$$\mathbb{P}(|X| \geq M) \lesssim \exp\left(-(M/\sigma)^{1/\theta}\right), \quad \forall M \geq 0. \tag{3}$$

Hence, this lemma implies that $\langle f_*(\boldsymbol{x})\nabla_{\boldsymbol{v}_i}\phi(\boldsymbol{v}_i \cdot \boldsymbol{x}), \boldsymbol{u}\rangle$ is $(P, 1/(2(1+q)))$-subweibull. ♣

### 2.3 Main result

The following is our main result. The proof of it can be found in Appendix D.

**Theorem 2.1** (Main Theorem). *Consider the setting and algorithm described above. Let $C > 0$ be a large universal constant. Suppose that $\log^C d \leq P \leq d$ and $\{\boldsymbol{v}_k^*\}_{k\in[P]}$ are orthonormal. Let $\delta_{\mathbb{P}} \in (\exp(-\log^C d), 1)$ and $\varepsilon_* > 0$ be given. Suppose that we choose $a_0, \eta, T, N$ satisfying*

$$m = \tilde{\Theta}(P), \quad N = \tilde{\Theta}\left(\frac{P^2}{\varepsilon_*^2 \delta_{\mathbb{P}}^2}\right), \quad \eta = \tilde{\Theta}_\phi\left(\frac{\varepsilon_*^2 \delta_{\mathbb{P}}}{PdP^{L/2-1}}\right), \quad T = \tilde{O}_\phi\left(\frac{P^{L/2-1}}{\eta \varepsilon_*^4 \delta_{\mathbb{P}}}\right).$$

*Then, there exists some $\lambda > 0$ such that at the end of training, we have $\mathcal{L}_{\mathrm{MSE}}(\boldsymbol{a}, \boldsymbol{V}) \leq \varepsilon_*$ with probability at least $1 - O(\delta_{\mathbb{P}})$.*

**Remark.** Note that $N \ll T$. Hence, the total number of samples needed is $T = \tilde{O}_\phi(dP^{L-1})$, which matches the sample complexity of separately learning $P$ single-index models with the relevant subspace known *a priori* and the noise scales with the ambient dimension $d$. ♣

## 3 The gradient flow analysis

In this section, we consider the situation where the training algorithm in Stage 1 is gradient flow over the population correlation loss instead of online SGD. The discussion here is non-rigorous and our formal proof does not rely on anything in this section. Nevertheless, this gradient flow analysis will provide valuable intuition on the behavior of online SGD.

For notational simplicity, we will assume w.l.o.g. that $\boldsymbol{v}_k^* = \boldsymbol{e}_k$. In addition, we will assume $\phi = h_2 + h_L$ with $L > 2$ for ease of presentation. Let $\boldsymbol{v}$ be an arbitrary first-layer neuron. By Lemma 2.1, the dynamics of $\boldsymbol{v}$ are controlled by[2]

$$\dot{\boldsymbol{v}}_\tau \approx 2\sum_{k=1}^P v_k(\boldsymbol{I} - \boldsymbol{v}\boldsymbol{v}^\top)\boldsymbol{e}_k + L\sum_{k=1}^P v_k^{L-1}(\boldsymbol{I} - \boldsymbol{v}\boldsymbol{v}^\top)\boldsymbol{e}_k.$$

The second term on the RHS comes from the normalized/projection. For each $k \in [d]$, we have

$$\frac{\mathrm{d}}{\mathrm{d}\tau}v_k^2 \approx 2\mathbb{1}\{k \leq P\}\left(2 + Lv_k^{L-2}\right)v_k^2 - 2\left(2\|\boldsymbol{v}_{\leq P}\|^2 + L\|\boldsymbol{v}_{\leq P}\|_L^L\right)v_k^2. \tag{4}$$

We further split Stage 1 into two substages. In Stage 1.1, the second-order terms dominate and $\|\boldsymbol{v}_{\leq P}\|^2 / \|\boldsymbol{v}_{>P}\|^2$ grows from $\Theta(P/d)$ to $\Theta(1)$. In Stage 1.2, $\boldsymbol{v}$ converges to one ground-truth direction relying on the signal from the higher-order terms.

The direction to which $\boldsymbol{v}$ will converge depends on the index of the largest $v_k^2$ at the beginning of Stage 1.2. With some standard concentration/anti-concentration argument, one can show that $\max_{k\in[P]} v_k^2$ is at least $1 + c$ times larger than the second-largest $v_k^2$ for a small constant $c > 0$ with probability at least $1/\mathrm{poly}(P)$ at the initialization (of Stage 1.1). Hence, as long as this gap can be preserved throughout Stage 1, we can choose $m = \mathrm{poly}(P)$ to ensure all ground-truth directions can be found after Stage 1.2.

### 3.1 Stage 1.1: learning the subspace and preservation of the gap

In this substage, we track $\|\boldsymbol{v}_{\leq P}\|^2 / \|\boldsymbol{v}_{>P}\|^2$ and $v_p^2/v_q^2$ where $p, q \in [P]$ are arbitrary. The goal is to show that $\|\boldsymbol{v}_{\leq P}\|^2 / \|\boldsymbol{v}_{>P}\|^2$ will grow to a constant while $v_p^2/v_q^2$ stay close to its initial value.

---

[2]In the main text, we use $\tau$ to index the time in this continuous-time process (as $t$ has been used to index the steps in the discrete-time process) and will often omit it when it is clear from the context.

For the norm ratio, by (4), we have

$$\frac{\mathrm{d}}{\mathrm{d}\tau}\frac{\|\boldsymbol{v}_{\leq P}\|^2}{\|\boldsymbol{v}_{>P}\|^2} = \frac{\frac{\mathrm{d}}{\mathrm{d}\tau}\|\boldsymbol{v}_{\leq P}\|^2}{\|\boldsymbol{v}_{>P}\|^2} - \frac{\|\boldsymbol{v}_{\leq P}\|^2}{\|\boldsymbol{v}_{>P}\|^2}\frac{\frac{\mathrm{d}}{\mathrm{d}\tau}\|\boldsymbol{v}_{>P}\|^2}{\|\boldsymbol{v}_{>P}\|^2} = \frac{4\|\boldsymbol{v}_{\leq P}\|^2}{\|\boldsymbol{v}_{>P}\|^2} + \frac{2L\|\boldsymbol{v}\|_L^L}{\|\boldsymbol{v}_{>P}\|^2}$$

$$-\frac{2\left(2\|\boldsymbol{v}_{\leq P}\|^2 + L\|\boldsymbol{v}_{\leq P}\|_L^L\right)\cancel{\|\boldsymbol{v}_{\leq P}\|^2}}{\cancel{\|\boldsymbol{v}_{>P}\|^2}} + \frac{\|\boldsymbol{v}_{\leq P}\|^2}{\cancel{\|\boldsymbol{v}_{>P}\|^2}}\frac{2\left(2\|\boldsymbol{v}_{\leq P}\|^2 + L\|\boldsymbol{v}_{\leq P}\|_L^L\right)\cancel{\|\boldsymbol{v}_{>P}\|^2}}{\|\boldsymbol{v}_{>P}\|^2}.$$

In particular, note that the terms coming from normalization cancel with each other. Moreover, this implies $\frac{\mathrm{d}}{\mathrm{d}\tau}\frac{\|\boldsymbol{v}_{\leq P}\|^2}{\|\boldsymbol{v}_{>P}\|^2} \geq 4\frac{\|\boldsymbol{v}_{\leq P}\|^2}{\|\boldsymbol{v}_{>P}\|^2}$, and therefore, it takes only at most $\frac{1+o(1)}{4}\log(d/P) = \Theta(\log(d/P))$ amount of time for the ratio to grow from $\Theta(P/d)$ to $\Theta(1)$. If we choose a small step size $\eta$ so that online SGD closely tracks the gradient flow, then the number of steps one should expect is $O(\log(d/P)/\eta)$.

Meanwhile, for any $p, q \in [P]$, we have

$$\frac{\mathrm{d}}{\mathrm{d}\tau}\frac{v_p^2}{v_q^2} = 2\left(2 + Lv_p^{L-2}\right)\frac{v_p^2}{v_q^2} - 2\left(2\|\boldsymbol{v}_{\leq P}\|^2 + L\|\boldsymbol{v}_{\leq P}\|_L^L\right)\frac{v_p^2}{v_q^2}$$

$$-\frac{v_p^2}{v_q^2}\left(2\left(2 + Lv_q^{L-2}\right) - 2\left(2\|\boldsymbol{v}_{\leq P}\|^2 + L\|\boldsymbol{v}_{\leq P}\|_L^L\right)\right) = 2L\left(v_p^{L-2} - v_q^{L-2}\right)\frac{v_p^2}{v_q^2}.$$

Note that not only those terms coming from normalization cancel with each other, but also the second-order terms. In particular, this also implies that we cannot learn the directions using only the second-order terms. At initialization, with high probability $v_k^2 = \tilde{O}(1/d)$ for all $k \in [P]$. Hence, if we assume the induction hypothesis $v_p^2 = \tilde{O}(1/P)$, then above will become $\frac{\mathrm{d}}{\mathrm{d}\tau}v_p^2/v_q^2 \lesssim P^{-1}v_p^2/v_q^2$. As a result, $v_{t,p}^2/v_{t,q}^2 \leq (1+o(1))v_{0,p}^2/v_{0,q}^2$ for any $t \leq \Theta(\log(d/P))$, as long as $P \geq \operatorname{poly}\log d$.

### 3.2 Stage 1.2: learning the directions

Let $\boldsymbol{v}$ be a first-layer neuron with $v_1^2 \geq (1+c)\max_{2\leq k\leq P}v_k^2$ for some small constant $c > 0$ at initialization. By our previous discussion, we know at the end of Stage 1.1, the above bound still holds with a potentially smaller constant $c > 0$. In addition, since $\|\boldsymbol{v}_{\leq P}\|^2 = \Theta(1)$, we also have $v_1^2 \geq \Omega(1/P)$ at the end of Stage 1.1. We claim that $\boldsymbol{v}$ will converge to $\boldsymbol{e}_1$. The argument here is similar to the proofs in [LMZ20] and [GRWZ21].

Again, by (4), we have

$$\frac{\mathrm{d}}{\mathrm{d}\tau}v_1^2 \approx 2\left(2 - 2\|\boldsymbol{v}_{\leq P}\|^2 + Lv_1^{L-2} - L\|\boldsymbol{v}_{\leq P}\|_L^L\right)v_1^2 \geq 2L\left(v_1^{L-2} - \|\boldsymbol{v}_{\leq P}\|_L^L\right)v_1^2$$

Assume the induction hypothesis $v_1^2 \geq (1+c)\max_{2\leq k\leq P}v_k^2$ and write

$$v_1^{L-2} - \|\boldsymbol{v}_{\leq P}\|_L^L = v_1^{L-2}\left(1 - v_1^2\right) - \left(\|\boldsymbol{v}_{\leq P}\|^2 - v_1^2\right)\sum_{k=2}^{P}\frac{v_k^2}{\|\boldsymbol{v}_{\leq P}\|^2 - v_1^2}v_k^{L-2}$$

Note that the summation is a weighted average of $\{v_k^{L-2}\}_{k\geq 2}$ and therefore can be upper bounded by $\left(v_1^2/(1+c)\right)^{L/2-1} \leq (1-c_L)v_1^{L-2}$ for some constant $c_L > 0$ that can only depend on $L$. Thus, we have

$$\frac{\mathrm{d}}{\mathrm{d}\tau}v_1^2 \gtrsim 2L\left(v_1^{L-2}\left(1 - v_1^2\right) - \left(\|\boldsymbol{v}_{\leq P}\|^2 - v_1^2\right)(1-c_L)v_1^{L-2}\right)v_1^2 \geq 2c_LL\left(1 - v_1^2\right)v_1^L.$$

When $v_1^2 \leq 3/4$, this implies $\frac{\mathrm{d}}{\mathrm{d}\tau}v_1^2 \gtrsim_L v_1^L$. As a result, it takes at most $O_L(P^{L/2-1})$ amount of time for $v_1^2$ to grow from $\Omega(1/P)$ to $3/4$ under gradient flow. It is important that $v_1^2 = \Omega(1/P)$ instead of $\Omega(1/d)$ at the start of Stage 1.2, since otherwise the time needed will be $O_L(d^{L-1})$. After $v_1^2$ reaches $3/4$, we have $\frac{\mathrm{d}}{\mathrm{d}\tau}(1 - v_1^2) \lesssim_L -\left(1 - v_1^2\right)$. Thus, $v_1^2$ will converge linearly to 1 afterwards.

# 4 From gradient flow to online SGD

In this section, we discuss how to convert the previous gradient flow analysis to an online SGD one. Our actual proof will be based directly on the online SGD analysis, but the overall idea is still proving that the online SGD dynamics of certain important quantities closely track their population gradient descent (GD) counterparts. Our choice of learning rate $\eta$ will be much smaller than what needed for GD to track GF, so the bottleneck comes from the GD-to-SGD conversion, not the GF-to-GD one. Provided that SGD tracks GD well, the number of steps/samples it needs to finish each substage is roughly the amount of time GF needs, divided by the step size $\eta$.

The rest of this section is organized as follows. In Section 4.1, we collect a few useful lemmas for controlling the difference between noisy dynamics and their deterministic counterparts. The idea behind them has appeared in [BAGJ21] and is also used in [AAM22]. Here, we simplify and slightly generalize their argument and provide a user-friendly interface. When used properly, it reduces the GD-to-SGD proof to routine calculus. Then, in Section 4.2, we discuss how to apply those general results to analyze the dynamics of online SGD in our setting.

## 4.1 Technical lemmas for analyzing general noisy dynamics

We start with the lemma that will be used to analyze $\|\boldsymbol{v}_{\leq P}\|^2 / \|\boldsymbol{v}_{>P}\|^2$. The formal proofs of it and all other lemmas in this subsection can be found in Section E.

**Lemma 4.1** (Stochastic Gronwall's lemma). *Suppose that $(X_t)_t$ satisfies*

$$X_{t+1} = (1 + \alpha)X_t + \xi_{t+1} + Z_{t+1}, \quad X_0 = x_0 > 0, \tag{5}$$

*where the signal growth rate $\alpha > 0$ and initialization $x_0 > 0$ are given, $(\xi_t)_t$ is an adapted process, and $(Z_t)_t$ is a martingale difference sequence. Define $x_t = (1 + \alpha)^t x_0$.*

*Let $T > 0$ and $\delta_{\mathbb{P}} \in (0, 1)$ be given. Suppose that there exists some $\delta_{\mathbb{P},\xi} \in (0, 1)$ and $\Xi, \sigma_Z > 0$ such that for every $t \geq 0$, if $X_t = (1 \pm 0.5)x_t$, then we have $|\xi_{t+1}| \leq (1 + \alpha)^t \Xi$ with probability at least $1 - \delta_{\mathbb{P},\xi}$ and $Z_{t+1}$ is conditionally $((1 + \alpha)^t \sigma_Z^2, \theta)$-subweibull. Then, if*

$$\Xi \lesssim \frac{x_0}{T} \quad and \quad \sigma_Z^2 \lesssim \frac{x_0^2}{T \log^{\theta+1}(T/\delta_{\mathbb{P}})}, \tag{6}$$

*we have $X_t = (1 \pm 0.5)x_t$ for all $t \in [T]$ with probability at least $1 - \delta_{\mathbb{P}} - T\delta_{\mathbb{P},\xi}$.*

**Condition** (6). One may interpret $Z_{t+1}$ as those terms coming from the difference between the population and mini-batch gradients, whose variance is typically quadratic in $\eta$, and $\xi_{t+1}$ as the higher-order error terms. $\alpha$ is usually small. In our case, it is proportional to the step size $\eta$. $T$ is usually the time needed for $X_t$ to grow from a small $x_0 > 0$ to $\Theta(1)$, which is roughly $\alpha^{-1} \log(1/x_0)$. Since the LHS' of (6) are $O(\eta^2)$ while the RHS' are $\Omega(\eta)$, (6) can be alternatively viewed as a condition on $\eta$. ♣

**Stochastic induction**. One important feature of this lemma is that it only requires the bounds $|\xi_{t+1}| \leq (1 + \alpha)^t \Xi$ and $\mathbb{E}[Z_{t+1}^2 \mid \mathcal{F}_t] \leq (1 + \alpha)^t \sigma_Z^2$ to hold when $X_t = (1 \pm 0.5)x_t$. This can be viewed as a form of induction and is particularly useful when considering the dynamics of, say, $v_k^2$. Similar to how the RHS of $\frac{\mathrm{d}}{\mathrm{d}\tau} v_{\tau,k}^2 = 2v_{\tau,k}\dot{v}_{\tau,k}$ depends on $v_{\tau,k}$, the size of $\xi_{t+1}, Z_{t+1}$ will usually depend on $X_t$. Hence, we will not be able to bound them without suitable induction hypotheses. ♣

**Remark on the subweibull condition**. We assume the martingale difference terms $(Z_{t+1})_t$ are conditionally subweibull. This allows us to get poly-logarithmic dependence on $\delta_{\mathbb{P}}$, which is important as we will eventually take union bound over $\operatorname{poly} P$ events. One may replace this condition with the weaker condition $\mathbb{E}[Z_{t+1}^2 \mid \mathcal{F}_t] \leq (1 + \alpha)^t \sigma_Z^2$. This will lead to a linear dependence on $\delta_{\mathbb{P}}$. ♣

*Proof sketch of Lemma 4.1.* For the ease of presentation, we assume that $|\xi_{t+1}| \leq (1 + \alpha)^t \Xi$ with probability at least $1 - \delta_{\mathbb{P},\xi}$ and $\mathbb{E}[Z_{t+1}^2 \mid \mathcal{F}_t] \leq (1 + \alpha)^t \sigma_Z^2$ always hold. This step can be made formal using a stopping time argument. See Section E for details. Then, Unroll (5) to obtain

$X_{t+1} = (1 + \alpha)^{t+1} x_0 + \sum_{s=1}^{t} (1 + \alpha)^{t-s} \xi_{s+1} + \sum_{s=1}^{t} (1 + \alpha)^{t-s} Z_{s+1}$. Divide both sides with $(1 + \alpha)^{t+1}$ and replace $t + 1$ with $t$. Then, the above becomes

$$X_t (1 + \alpha)^{-t} = x_0 + \sum_{s=1}^{t} (1 + \alpha)^{-s} \xi_s + \sum_{s=1}^{t} (1 + \alpha)^{-s} Z_s.$$

The second term is bounded by $T\Xi$ (uniformly over $t \leq T$) with probability at least $1 - T\delta_{\mathbb{P},\xi}$. Note that $(1 + \alpha)^{-s} Z_s$ is still a martingale difference sequence. Hence, by Doob's $L^2$-submartingale inequality, the third term is bounded by $x_0/4$ with probability at least $16\sigma_Z^2/(\alpha x_0^2)$. Thus, when (6) holds, the RHS is $(1 \pm 0.5) x_0$ with probability at least $1 - T\delta_{\mathbb{P},\xi} - \delta_{\mathbb{P}}$. Multiply both sides with $(1+\alpha)^t$, and we complete the proof. To improve the dependence on $\delta_{\mathbb{P}}$ from linear to poly-logarithmic, it suffices to replace Doob's $L^2$-submartingale inequality with a variant of Freedman's inequality that works with subweibull variables (cf. Appendix E). □

Using the same strategy, one can prove a similar lemma that deals with the case $\alpha = 0$, which will be used to show the preservation of the gap in Stage 1.1. Another interesting case is where the growth is not linear but polynomial. This is the case of Stage 1.2 in our setting. For this case, we have the following lemma.[3]

**Lemma 4.2.** *Let $(X_t)_t$ be a non-negative stochastic process satisfying*

$$X_{t+1} \geq X_t + \alpha X_t^p + Z_{t+1} + \xi_{t+1}, \quad X_0 = x_0 > 0, \tag{7}$$

*where $\alpha > 0$, $(Z_{t+1})_t$ is a martingale difference sequence, and $(\xi_t)_t$ is an adapted process. Let $\hat{x}_t$ be the solution to the deterministic recurrence relationship $\hat{x}_{t+1} = \hat{x}_t + \alpha \hat{x}_t^p, \hat{x}_0 = x_0/2$.*

*Let $\delta_{\mathbb{P}} \in (0,1)$ be given and $T := \inf \left\{ t \lesssim \left( p\alpha (x_0/2)^{p-1} \right)^{-1} : X_t \geq 1 \right\}$. Suppose that there exists $\Xi, \sigma_Z > 0$ and $\delta_{\mathbb{P},\xi} \in (0,1)$ such that if $X_t \geq \hat{x}_t$ and $t \leq T$, we have $|\xi_t| \leq \Xi X_t$ with probability at least $1 - \delta_{\mathbb{P},\xi}$ and $Z_{t+1}$ is conditionally $(\sigma_Z^2 X_t, \theta)$-subweibull. Then, if*

$$\alpha \lesssim x_0^{p-1}/p, \quad \Xi \lesssim p\alpha x_0^{p-1}, \quad \sigma_Z^2 \lesssim p\alpha x_0^p \operatorname{poly} \log(T/\delta_{\mathbb{P}}), \tag{8}$$

*we have $X_t \geq \hat{x}_t$ for all $t \leq T$ and $X_t \geq 1$ with probability at least $1 - T\delta_{\mathbb{P},\xi} - \delta_{\mathbb{P}}$.*

The proof of this lemma can be found in Section E. It is similar to the previous proof in spirit: we replace $(1 + \alpha)^t$ with $\prod_{s=0}^{t-1} (1 + \alpha X_s^{p-1})$ and unroll the recurrence. However, unlike the linear case, it is generally difficult to upper bound the difference between $X_t$ and $\hat{x}_t$, as this type of polynomial systems exhibit sharp transitions and blow up in finite time. Consequently, $|\xi_t| \leq \Xi X_t$ and $\mathbb{E}[Z_{t+1}^2 \mid \mathcal{F}_t] \leq \sigma_Z^2 X_t$ do not directly imply that $|\xi_t| \lesssim \Xi \hat{x}_t$ and $\mathbb{E}[Z_{t+1}^2 \mid \mathcal{F}_t] \lesssim \sigma_Z^2 \hat{x}_t$, and this makes the analysis tricky as the RHS' are not deterministic. To handle this issue, we use the following recoupling argument: whenever $X_t \geq 4\hat{x}_t$, we replace the current $\hat{x}_t$ with $X_t/2$. Clear that this can only increase $\hat{x}_t$, and it ensures $X_t \leq 4\hat{x}_t$ always holds. Moreover, after each recoupling, $\hat{x}_t$ will at least double. As a result, the conditions we need to absorb the noises will also become weaker.

## 4.2 Sample complexity of online SGD

In this subsection, we demonstrate how to use the previous results to obtain results for online SGD and discuss why the sample complexity is $\tilde{O}(dP^{L-1})$ instead of $\tilde{O}(d^{L-1})$ even though we are relying on the $L$-th order terms to learn the directions.

### 4.2.1 A simplified version of Stage 1.1

As an example, we consider the dynamics of $Pv_p^2/(dv_q^2)$ where $p \leq P$ and $q > P$ and assume both of $v_p$ and $v_q$ are small and $Pv_p^2/(dv_q^2) \leq 1$. This can be viewed as a simplified version of the analysis of $\|\boldsymbol{v}_{\leq P}\|^2 / \|\boldsymbol{v}_{>P}\|^2$ in Stage 1.1. The analysis of other quantities/stages is essentially the same — we rewrite the update rule to single out martingale difference terms and the higher-order error terms, and apply a suitable lemma from the previous subsection (or Section E) to complete the proof.

For the ease of presentation, in this subsection, we ignore the higher-order terms. In particular, we assume the approximation $\hat{v}_{t+1,k} \approx v_{t,k} + 2\eta \left( \mathbb{1}\{k \leq P\} - \|\boldsymbol{v}_{\leq P}\|^2 \right) + \eta Z_{t+1,k}$, for all $k \in [d]$,

---

[3] In an early version of this manuscript, we did not relax the conditions on the noises when $X_t$ grows as in Lemma 4.1. We thank Eshaan Nichani for pointing out that this would result in a suboptimal rate.

where $Z_{t+1,k}$ represents the difference between the population and mini-batch gradients. Then, we compute

$$\hat{v}_{t+1,k}^2 \approx \left(1 + 4\eta \left(\mathbb{1}\{k \leq P\} - \|\boldsymbol{v}_{\leq P}\|^2\right)\right) v_k^2 + 2\eta v_k Z_k \pm C_L \eta^2 (1 \vee Z_k^2).$$

Here, the last term is the higher-order term and will eventually be included in $\xi$. For simplicity, we will also ignore them in the following discussion. The second term is the martingale difference term. Its (conditional) variance depend on $v_k$, and this necessitates the induction-style conditions in Lemma 4.1. Note that $v_{t+1,p}^2/v_{t+1,q}^2 = \hat{v}_{t+1,p}^2/\hat{v}_{t+1,q}^2$. Hence, we have

$$\frac{v_{t+1,p}^2}{v_{t+1,q}^2} \approx \frac{\left(1 + 4\eta \left(1 - \|\boldsymbol{v}_{\leq P}\|^2\right)\right) v_p^2 + 2\eta v_p Z_p}{\left(1 - 4\eta \|\boldsymbol{v}_{\leq P}\|^2\right) v_q^2 + 2\eta v_q Z_q}.$$

Repeatedly use the elementary identity $\frac{1}{a+\delta} = \frac{1}{a}\left(1 - \frac{\delta}{a}\left(1 - \frac{\delta}{a+\delta}\right)\right) \approx \frac{1}{a}\left(1 - \frac{\delta}{a}\right)$ for any $a > 0$ and small $\delta > 0$, we can rewrite the above equation as

$$\frac{Pv_{t+1,p}^2}{dv_{t+1,q}^2} \approx (1 + 4\eta)\frac{Pv_p^2}{dv_q^2} - \frac{Pv_p^2}{dv_q^2}\frac{2\eta v_q Z_q}{v_q^2} + \frac{2P\eta v_p Z_p}{dv_q^2}.$$

Suppose that $v_p^2 \approx v_q^2$ at initialization and assume the induction hypothesis $Pv_p^2/(dv_q^2) = (1 \pm 0.5)(1 + 4\eta)^t Pv_{0,p}^2/(dv_{0,q}^2)$. Then, by Lemma 2.1, the conditional variance of the martingale difference terms (the last two terms) is bounded by $O_L((1 + 4\eta)^t \eta^2 P^2/d)$. Using the language of Lemma 4.1, this means $\sigma_Z^2 \leq O_L(\eta^2 P^2/d)$. Meanwhile, by our gradient flow analysis, the number steps Stage 1.1 needs is roughly $\log d/\eta$. Hence, in order for (the second condition of) (6) to hold, it suffices to choose $\eta = \tilde{O}(1/d)$. One can also show that for the higher-order terms to be small, it suffices to choose $\eta = \tilde{O}(1/(dP))$. As a result, for Stage 1.1, the sample complexity is $\tilde{O}(dP)$.

### 4.2.2 The improved sample complexity for Stage 1.2

To see why the existence of the second-order terms can reduce the sample complexity from $d^{\mathrm{IE}-1}$ to $d \operatorname{poly}(P)$, first note that after Stage 1.1, $\max_{p \in [P]} v_p^2$ will be $\Omega(1/P)$. Also note that the conditions in Lemma 4.2 depend on the initial value. With the initial value being $\Omega(1/P)$ instead of $\tilde{O}(1/d)$, the largest possible step size we can choose will be $O(1)/(PdP^{L/2-1})$, which is much larger than the usual $O(1/(Pd^{L/2}))$ requirement from the vanilla information exponent argument. Meanwhile, by our gradient flow analysis, we know the number of iterations needed is $O(P^{L/2-1}/\eta)$. Combine these and we obtain the $\tilde{O}(dP^{L-1})$ sample complexity.

## 5 Conclusion and limitations

In this work, we study the task of learning multi-index models of form $f_*(\boldsymbol{x}) = \sum_{k=1}^{P} \phi(\boldsymbol{v}_k^* \cdot \boldsymbol{x})$ with $P \ll d$, $\{\boldsymbol{v}_k^*\}_k$ be orthogonal and $\phi = \hat{\phi}_2 h_2 + \sum_{l=L}^{\infty} \hat{\phi}_l h_l$. By considering both the lower- and higher-order terms, we prove an $\tilde{O}(d \operatorname{poly}(P))$ bound on the sample complex for strong recovery of directions using online SGD, which improves the results one can obtain using vanilla information exponent-based analysis.

The main limitation of this work is the orthogonality condition. This can potentially be relaxed to near-orthogonality as in [OSSW24]. Extending this result beyond near-orthogonal teacher neurons is an interesting but challenging future direction, as when the teacher neurons are not near-orthogonal, this task is hard in general. However, we conjecture that when the target model has a hierarchical structure across different orders, online SGD can gradually learn the directions using those terms of different order sequentially.

Another limitation of this work is the assumption that the signal strengths are isotropic. When this is not true, training with the second-order terms and correlation loss will make all neurons collapse to the largest direction or require $d^2$ samples if we perform only one gradient step [DLS22, DKPS24]. That being said, it is still reasonable to expect the overall sample complexity to be improved if we can leverage the second-order terms properly. Formally establishing this is also a potential future direction.

## Acknowledgements

JDL acknowledges support of NSF CCF 2002272, NSF IIS 2107304, NSF CIF 2212262, ONR Young Investigator Award, and NSF CAREER Award 2144994. We thank Eshaan Nichani for pointing out an earlier version of Lemma 4.2 is suboptimal, and anonymous reviewer 4V3d for helpful discussions on relaxing Assumption 1.

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

# Contents

# A  Preliminaries

## A.1  Population and per-sample gradients

In this subsection, we show that the task of learning the multi-index target function $f_*(\boldsymbol{x}) = \sum_{k=1}^{P} \phi(\boldsymbol{v}_k^* \cdot \boldsymbol{x})$ can be reduced to tensor decomposition and prove the tail bounds in Lemma 2.1.

For the first goal, we will need the following classical result on Hermite polynomials (cf. Chapter 11.2 of [O'D14]) and correlated Gaussian variables.

**Lemma A.1** (Proposition 11.31 of [O'D14]). *For $k \in \mathbb{N}_{\geq 0}$ denote the normalized Hermite polynomials. Let $\rho \in [-1, 1]$ and $z, z'$ be $\rho$-correlated standard Gaussian variables. Then, we have*
$$\mathop{\mathbb{E}}_{z,z'} [h_k(z)h_j(z')] = \mathbb{1}\{k = j\}\rho^k.$$

**Lemma A.2.** *Under the setting described in Section 2, we have*
$$\mathop{\mathbb{E}}_{\boldsymbol{x}} [f_*(\boldsymbol{x})\nabla_{\boldsymbol{v}}\phi(\boldsymbol{v} \cdot \boldsymbol{x})] = \sum_{k=1}^{P}\sum_{l=1}^{\infty} l\hat{\phi}_l^2 \langle \boldsymbol{v}_k^*, \boldsymbol{v}\rangle^{l-1} \boldsymbol{v}_k^*.$$

*Proof.* Let $\phi = \sum_{k=0}^{\infty} \hat{\phi}_k h_k$ be the Hermite expansion of $\phi$ where the convergence is in $L^2$ sense. For any $\rho \in [-1, 1]$ and $\rho$-correlated standard Gaussian variables $z, z'$, we have
$$\mathop{\mathbb{E}}_{z,z'} \{\phi(z)\phi(z')\} = \sum_{k,l=0}^{\infty} \hat{\phi}_k\hat{\phi}_l \mathop{\mathbb{E}}_{z,z'} \{h_k(z)h_l(z')\} = \sum_{k=0}^{\infty} \hat{\phi}_k^2 \rho^k,$$

where the first equality comes from the Dominated Convergence Theorem and the second from Lemma A.1. Therefore, we have
$$\mathop{\mathbb{E}}_{\boldsymbol{x}} [f_*(\boldsymbol{x})\phi(\boldsymbol{v} \cdot \boldsymbol{x})] = \sum_{k=1}^{P} \mathop{\mathbb{E}}_{\boldsymbol{x}} [\phi(\boldsymbol{v}_k^* \cdot \boldsymbol{x})\phi(\boldsymbol{v} \cdot \boldsymbol{x})] = \sum_{k=1}^{P}\sum_{l=1}^{\infty} \hat{\phi}_l^2 \langle \boldsymbol{v}_k^*, \boldsymbol{v}\rangle^l.$$

Then, we compute
$$\mathop{\mathbb{E}}_{\boldsymbol{x}} [f_*(\boldsymbol{x})\nabla_{\boldsymbol{v}}\phi(\boldsymbol{v} \cdot \boldsymbol{x})] = \sum_{k=1}^{P}\sum_{l=1}^{\infty} \hat{\phi}_l^2 \nabla_{\boldsymbol{v}} \langle \boldsymbol{v}_k^*, \boldsymbol{v}\rangle^l = \sum_{k=1}^{P}\sum_{l=1}^{\infty} l\hat{\phi}_l^2 \langle \boldsymbol{v}_k^*, \boldsymbol{v}\rangle^{l-1} \boldsymbol{v}_k^*.$$
$\square$

Now, we consider the per-sample gradient. The goal here is to prove variance and tail bounds for $\langle f_*(\boldsymbol{x})\nabla_{\boldsymbol{v}}\phi(\boldsymbol{v} \cdot \boldsymbol{x}), \boldsymbol{u}\rangle$, where $\boldsymbol{u} \in \mathbb{S}^{d-1}$ is an arbitrary direction that is independent of $\boldsymbol{x}$. First, we upper bound $f_*$. To this end, we will use need the following concentration inequality for the sum of independent subweibull variables. It is a consequence of Theorem 3.2 of [KC22] and the discussion after Definition 2.3 of the same paper.

**Lemma A.3** ([KC22]). *Let $\psi_\alpha(x) = \exp(x^\alpha) - 1$ and $\|\cdot\|_{\psi_\alpha}$ denote the corresponding Orlicz norm. Let $\alpha \leq 1$ and $X_1, \ldots, X_n$ be i.i.d. mean zero random variables with variance $\sigma^2$ and $\|X_1\|_{\psi_\alpha} < \infty$. Then, for any $\delta_{\mathbb{P}} \in (0, 1)$, we have*
$$\left|\sum_{i=1}^{n} X_i\right| \lesssim_\alpha \sqrt{n}\sigma \log^{1/2}(1/\delta_{\mathbb{P}}) + \sqrt{n}\|X_1\|_{\psi_\alpha} \log^{1/\alpha}(n/\delta_{\mathbb{P}}), \quad \textit{with probability at least } 1 - \delta_{\mathbb{P}}.$$

**Lemma A.4.** *Suppose that Assumption 1 holds and $\{\boldsymbol{v}_k^*\}_k$ are orthonormal. Then, for any $\delta_{\mathbb{P}} \in (0, 1)$, we have*
$$|f_*(\boldsymbol{x})| \lesssim \sqrt{P}\log^q(P/\delta_{\mathbb{P}}), \quad \textit{with probability at least } 1 - \delta_{\mathbb{P}}.$$

*Proof.* Write $Y_k := \phi(\boldsymbol{v}_k^* \cdot \boldsymbol{x})$. By the orthonormality of $\{\boldsymbol{v}_k^*\}_k$, $\{Y_k\}_k$ are independent variables. For any $p \geq 1$, we have
$$\|Y_1\|_{L^p} \leq \left(\mathop{\mathbb{E}}_{z\sim\mathcal{N}(0,1)} |\phi|^p(z)\right)^{1/p} \leq C\left(\mathop{\mathbb{E}}_{z\sim\mathcal{N}(0,1)} \left[(1 + z^2)^{pq/2}\right]\right)^{1/p}$$
$$\leq Cp^q \sqrt{\mathop{\mathbb{E}}_{z\sim\mathcal{N}(0,1)} [(1 + z^2)^q]} \lesssim p^q,$$

where the first inequality in the second line comes from the Gaussian hypercontractivity. This implies that $\|Y_1\|_{\psi_{1/q}} \lesssim 1$ and $\mathbf{Var}\, Y_1 \lesssim 1$. Thus, by Lemma A.3, we have with probability at least $1 - \delta_{\mathbb{P}}$ that

$$|f_*(\boldsymbol{x})| \lesssim \sqrt{P}\log^{1/2}(1/\delta_{\mathbb{P}}) + \sqrt{P}\log^q(P/\delta_{\mathbb{P}}) \lesssim \sqrt{P}\log^q(P/\delta_{\mathbb{P}}).$$

$\square$

**Lemma A.5.** *Suppose that Assumption 1 holds and $\{\boldsymbol{v}_k^*\}_k$ are orthonormal. Then, we have*

$$\mathbb{E}\left\langle f_*(\boldsymbol{x})\nabla_{\boldsymbol{v}}\phi(\boldsymbol{v}\cdot\boldsymbol{x}), \boldsymbol{u}\right\rangle^2 \lesssim_\phi P,$$

$$|\left\langle f_*(\boldsymbol{x})\nabla_{\boldsymbol{v}}\phi(\boldsymbol{v}\cdot\boldsymbol{x}), \boldsymbol{u}\right\rangle| \lesssim_\phi P^{1/2}\log^{2(1+q)}\log(m/\delta_{\mathbb{P}}) \quad \textit{with probability at least } 1 - \delta_{\mathbb{P}},$$

*where $q$ is the degree of $\phi$ if it is a polynomial and $Q = 0$ if $\phi$ is Lipschitz.*

*Proof.* Note that $\langle\nabla_{\boldsymbol{v}}\phi(\boldsymbol{v}\cdot\boldsymbol{x}), \boldsymbol{u}\rangle = \phi'(\boldsymbol{v}\cdot\boldsymbol{x})\langle\boldsymbol{x}, \boldsymbol{u}\rangle$. First, for the variance, we have

$$\mathbb{E}(f_*(\boldsymbol{x})\phi'(\boldsymbol{v}\cdot\boldsymbol{x})\langle\boldsymbol{u}, \boldsymbol{x}\rangle)^2 \lesssim \mathbb{E}\,f_*^6 + \mathbb{E}(\phi'(\boldsymbol{v}\cdot\boldsymbol{x}))^6 + \mathbb{E}\langle\boldsymbol{u}, \boldsymbol{x}\rangle^6 \lesssim P,$$

where the second inequality comes from Assumption 1 and the hypercontractivity of Gaussian. This implies $\mathbb{E}\langle\nabla f(\boldsymbol{x}), \boldsymbol{u}\rangle^2 \lesssim P$. For the tail bound, first recall from the previous lemma that $|f_*(\boldsymbol{x})| \lesssim \sqrt{P}\log^q(P/\delta_{\mathbb{P}})$ with probability at least $1 - \delta_{\mathbb{P}}$. The proof of it also implies $|\phi'(\boldsymbol{v}\cdot\boldsymbol{x})| \lesssim \log^q(P/\delta_{\mathbb{P}})$ with probability at least $1 - \delta_{\mathbb{P}}$. Finally, since $\langle\boldsymbol{x}, \boldsymbol{u}\rangle$ is 1-subgaussian, we have $|\boldsymbol{x}\cdot\boldsymbol{u}| \lesssim \log^{1/2}(1/\delta_{\mathbb{P}})$ with probability at least $1 - \delta_{\mathbb{P}}$. Combine these bounds, take the union bound over $m$ learner neurons, and we complete the proof.

$\square$

## A.2 Typical structure at initialization

In this subsection, we use the results in Section A.3 to analyze the structure of $\boldsymbol{v}_1, \ldots, \boldsymbol{v}_m$ at initialization. Recall that we initialize $\boldsymbol{v}_i$ with $\mathrm{Unif}(\mathbb{S}^{d-1})$ independently. Meanwhile, note that for $\boldsymbol{v} \sim \mathrm{Unif}(\mathbb{S}^{d-1})$, we have $\boldsymbol{v} \overset{d}{=} \boldsymbol{Z}/\|\boldsymbol{Z}\|$ where $\boldsymbol{Z} \sim \mathcal{N}(0, \boldsymbol{I}_d)$.

We start with a lemma on the largest coordinate.

**Lemma A.6** (Largest coordinate). *Let $\boldsymbol{v} \sim \mathrm{Unif}(\mathbb{S}^{d-1})$. For any $K \geq 1$, we have*

$$\max_{i\in[d]}|v_i| \leq \frac{4\sqrt{2K\log d}}{\sqrt{d}} \quad \textit{with probability at least } 1 - \frac{4}{d^K}.$$

*As a corollary, for any $\delta_{\mathbb{P}} \in (0, 1)$, at initialization, we have*

$$\max_{i\in[m]}\|\boldsymbol{v}_i\|_\infty \leq \frac{4\sqrt{2\log(4m/\delta_{\mathbb{P}})}}{\sqrt{d}} \quad \textit{with probability at least } 1 - \delta_{\mathbb{P}}.$$

*Proof.* Let $\boldsymbol{Z} \sim \mathcal{N}(0, \boldsymbol{I}_d)$. Recall that $\boldsymbol{Z}/\|\boldsymbol{Z}\|$ follows the uniform distribution over the sphere. By Lemma A.11, we have $\|\boldsymbol{Z}\| \geq \sqrt{d}/2$ with probability at least $1 - 2\exp(-d/18)$. Then, by Lemma A.13, with probability at least $1 - 2e^{-d/18} - 2e^{-s^2/2}$, we have

$$\frac{\max_{i\in[d]}|Z_i|}{\|\boldsymbol{Z}\|} \leq \frac{\sqrt{2\log d} + s}{\sqrt{d}/2} = \frac{2\sqrt{2\log d}}{\sqrt{d}} + \frac{2s}{\sqrt{d}}.$$

Let $K \geq 1$ be arbitrary. Choose $s = \sqrt{2K\log d}$ and the above becomes

$$\frac{\max_{i\in[d]}|Z_i|}{\|\boldsymbol{Z}\|} \leq \frac{4\sqrt{2K\log d}}{\sqrt{d}} \quad \text{with probability at least } 1 - \frac{4}{d^K}.$$

For the corollary, use union bound and choose $K = \log(4m/\delta_{\mathbb{P}})/\log d$, we have

$$\max_{i\in[m]}\|\boldsymbol{v}_i\|_\infty \leq \frac{4\sqrt{2\log(4m/\delta_{\mathbb{P}})}}{\sqrt{d}} \quad \text{with probability at least } 1 - \frac{4m}{d^K} = 1 - \delta_{\mathbb{P}}.$$

$\square$

Suppose that we only have higher-order terms. Then, for a neuron $\boldsymbol{v} \in \mathbb{S}^{d-1}$ to converge to a ground-truth direction $\boldsymbol{e}_k$ in a reasonable amount of time, we need $v_k^2$ to be the largest among all $v_i^2$ and there is gap between it and the second largest $v_i^2$. The following lemma ensures that when $m$ is large, for every ground-truth direction $\{\boldsymbol{e}_k\}_{k \in [P]}$, there will be at least one neuron satisfying the above property. Note that in our case, we only need to ensure $v_k^2$ is the largest among all $\{v_i^2\}_{i \in [P]}$ instead of $\{v_i^2\}_{i \in [d]}$, as the second-order term will help us identify the correct subspace.

**Lemma A.7** (Existence of good neurons). *Let $\delta_{\mathbb{P}} \in (e^{-\log^C d}, 1)$ be given. Suppose that $m \geq 2P \log(P/\delta_{\mathbb{P}}) = \tilde{\Theta}(P)$. Then, at initialization, with probability at least $1 - \delta_{\mathbb{P}}$, we have*

$$\forall p \in [P] \, \exists i \in [m] \quad such that \quad \frac{v_{i,p}^2}{\max_{q \in [P] \setminus \{p\}} v_{i,q}^2} \geq 1 + \frac{1}{200 \log(48P)} = 1 + \tilde{\Theta}(1).$$

*Proof.* Let $\delta_0$ be a parameter to chosen later and let $\delta_{\mathbb{P},0}$ be the probability that $\max_{q \in [P]} v_q^2$ is smaller than $1 + \delta_0$ times the second largest $v_q^2$. For each $p \in [P]$, let $B_p$ be the event $\left\{ \forall k \in [m], v_{k,p}^2 \leq (1 + \delta_0) \max_{q \in [P] \setminus \{p\}} v_{k,q}^2 \right\}$. To bound $\mathbb{P}[B_p]$, we write

$$\mathbb{P}[B_p] = \left( \mathbb{P}_{\boldsymbol{v} \sim \mathrm{Unif}(\mathbb{S}^{d-1})} \left[ v_p^2 \leq (1+c) \max_{q \in [P] \setminus \{p\}} v_q^2 \right] \right)^m$$

$$= \left( \mathbb{P} \left[ v_p^2 \neq \max_{q \in [P]} v_q^2 \right] + \mathbb{P} \left[ v_p^2 \leq (1+c) \max_{q \in [P] \setminus \{p\}} v_q^2 \,\middle|\, v_p^2 = \max_{q \in [P]} v_q^2 \right] \mathbb{P} \left[ v_p^2 = \max_{q \in [P]} v_q^2 \right] \right)^m$$

$$= \left( 1 - \frac{1}{P} + \frac{\delta_{\mathbb{P},0}}{P} \right)^m.$$

By Corollary A.12, if we choose $\delta_{\mathbb{P},0} = 1/2$, then we can choose

$$\delta_0 = \frac{1}{200 \log(48P)}.$$

With the above choices of parameters, we have

$$\mathbb{P} \left[ \bigcup_{p \in [P]} B_p \right] \leq P \left( 1 - \frac{1}{2P} \right)^m \leq P \exp \left( -\frac{m}{2P} \right).$$

For the last term to be bounded by $\delta_{\mathbb{P}}$, it suffices to choose $m \geq 2P \log(P/\delta_{\mathbb{P}})$. $\qquad \square$

**Lemma A.8** (Typical structure at initialization). *Let $\delta_{\mathbb{P}} \in (e^{-\log^C d}, 1)$ be given and $c_g > 0$ be a small constant. Suppose that $\{\boldsymbol{v}_k\}_{k=1}^m \sim \mathrm{Unif}(\mathbb{S}^{d-1})$ independently with*

$$m = \Theta \left( P \log(P/\delta_{\mathbb{P}}) \right).$$

*Then, with probability at least $1 - 3\delta_{\mathbb{P}}$, we have*

$$\forall p \in [P] \, \exists i \in [m] \quad such that \quad \frac{v_{i,p}}{\max_{q \in [P] \setminus \{p\}} |v_{i,q}|} \geq 1 + \frac{\Theta(1)}{\log P}, \tag{9}$$

$$\forall i \in [m], \quad \|\boldsymbol{v}_i\|_\infty \leq \frac{20\sqrt{\log(P/\delta_{\mathbb{P}})}}{\sqrt{d}},$$

$$\forall i \in [m], \quad \frac{\sqrt{P}}{3\sqrt{d}} \leq \frac{\|\boldsymbol{v}_{\leq P}\|}{\|\boldsymbol{v}\|} \leq \frac{3\sqrt{P}}{\sqrt{d}}.$$

*Proof.* The first two bounds comes directly from Lemma A.6 and Lemma A.7 and the fact that we use symmetric initialization. By Lemma A.11, we have

$$\mathbb{P} \left( \big| \|\boldsymbol{Z}\| - \mathbb{E}\|\boldsymbol{Z}\| \big| \geq \sqrt{d}/2 \right) \leq 2e^{-d/8},$$

$$\mathbb{P} \left( \big| \|\boldsymbol{Z}_{\leq P}\| - \mathbb{E}\|\boldsymbol{Z}_{\leq P}\| \big| \geq \sqrt{P}/2 \right) \leq 2e^{-P/8}.$$

As a result, for any $v \sim \mathrm{Unif}(\mathbb{S}^{d-1})$, we have with probability at least $1 - 4e^{-P/8}$ that

$$\frac{\|v_{\leq P}\|}{\|v\|} \overset{d}{=} \frac{\|Z_{\leq P}\|}{\|Z\|} = \frac{\mathbb{E}\|Z_{\leq P}\| \pm \sqrt{P}/2}{\mathbb{E}\|Z\| \pm \sqrt{d}/2} = [1/3, 3] \times \sqrt{\frac{P}{d}}.$$

Since we assume $P \geq \log^{C'} d$ for a large $C'$, we have $4e^{-P/8} \leq \delta_{\mathbb{P}}/m$. This gives the third bound. $\qquad\square$

## A.3   Concentration and anti-concentration of Gaussian

**Lemma A.9.** *Let $Z_1, \ldots, Z_d$ be independent $\mathcal{N}(0,1)$ variables. Let $Y_1, Y_2$ be the largest and second largest of $|Z_1|, \ldots, |Z_d|$. For any $\delta_{\mathbb{P}} \in (0,1)$, we have*

$$\frac{Y_1}{Y_2} \geq 1 + \frac{\delta_{\mathbb{P}}}{12 \log\left(12d/\delta_{\mathbb{P}}\right)} \quad \text{with probability at least } 1 - \delta_{\mathbb{P}}.$$

*Proof.* The following proof is adapted from this MathOverflow answer [Pin20]. Let $f$ and $F$ denote the PDF and CDF of $|Z|$ with $Z \sim \mathcal{N}(0,1)$, respectively. We have $f(y) = 2\phi(y)\mathbb{1}\{y \geq 0\}$ and $F(y) = \mathrm{erf}(y/\sqrt{2})\mathbb{1}\{y \geq 0\}$, where $\phi$ is the PDF of $\mathcal{N}(0,1)$ and $\mathrm{erf}$ is the error function. We will use the following formula for the joint PDF of two order statistics:

$$f_{Y_1,Y_2}(y_1, y_2) = d(d-1)F^{d-2}(y_2)f(y_2)f(y_1)\mathbb{1}\{0 < y_2 < y_1\}.$$

Consider small $s > 0$. We compute

$$\mathbb{P}\left(\frac{Y_1}{Y_2} \geq 1 + s\right) = \int_0^\infty \int_0^\infty \mathbb{1}\{y_1 \geq (1+s)y_2\} f_{Y_1,Y_2}(y_1, y_2)\,\mathrm{d}y_2\mathrm{d}y_1$$

$$= \int_0^\infty \int_0^\infty f_{Y_1,Y_2}((1+s)y_2 + r, y_2)\,\mathrm{d}y_2\mathrm{d}r$$

$$= d(d-1)\int_0^\infty F^{d-2}(y_2)f(y_2)\left(\int_0^\infty f((1+s)y_2 + r)\,\mathrm{d}r\right)\mathrm{d}y_2$$

$$= d(d-1)\int_0^\infty F^{d-2}(y_2)f(y_2)\left(1 - F((1+s)y_2)\right)\mathrm{d}y_2.$$

Let $G = F^{-1}$. With the change-of-variables $u = F(y_2)$, $y_2 = G(u)$, we can rewrite the above as

$$\mathbb{P}\left(\frac{Y_1}{Y_2} \geq 1 + s\right) = d(d-1)\int_0^1 u^{d-2}\left(1 - F((1+s)G(u))\right)f(G(u))\,\mathrm{d}G(u)$$

$$= d(d-1)\int_0^1 u^{d-2}\left(1 - F((1+s)G(u))\right)f(G(u))\frac{1}{F'(G(u))}\,\mathrm{d}u$$

$$= d(d-1)\int_0^1 u^{d-2}\left(1 - F((1+s)G(u))\right)\mathrm{d}u.$$

Now, we analyze the last integral. We will use the following expansion of the (complementary) error function:

$$1 - F(y) = 1 - \mathrm{erf}\left(y/\sqrt{2}\right) = \frac{e^{-y^2/2}}{y\sqrt{\pi/2}}\left(1 - \frac{1}{\sqrt{\pi}}\int_y^\infty r^2 e^{-r^2}\,\mathrm{d}r\right).$$

For notational simplicity, put $w = G(u)$. Then, we have

$$1 - F((1+s)w) = \frac{e^{-(1+s)^2 w^2/2}}{(1+s)w\sqrt{\pi/2}}\left(1 - \frac{1}{\sqrt{\pi}}\int_{(1+s)w}^\infty r^2 e^{-r^2}\,\mathrm{d}r\right)$$

$$= \frac{\exp\left(-(s+s^2/2)w^2\right)}{1+s}\frac{e^{-w^2/2}}{w\sqrt{\pi/2}}\left(1 - \frac{1}{\sqrt{\pi}}\int_{(1+s)w}^\infty r^2 e^{-r^2}\,\mathrm{d}r\right)$$

$$= \frac{\exp\left(-(s+s^2/2)w^2\right)}{1+s}\left(1 - F(w)\right)\frac{1 - \frac{1}{\sqrt{\pi}}\int_{(1+s)w}^\infty r^2 e^{-r^2}\,\mathrm{d}r}{1 - \frac{1}{\sqrt{\pi}}\int_w^\infty r^2 e^{-r^2}\,\mathrm{d}r}.$$

Note that the last factor is at least 1 and $F(w) = F(G(u)) = F(F^{-1}(u)) = u$. Therefore,

$$1 - F((1+s)w) \geq \frac{\exp\left(-(s+s^2/2)w^2\right)}{1+s}(1-u).$$

As a result, we have

$$\mathbb{P}\left(\frac{Y_1}{Y_2} \geq 1+s\right) \geq d(d-1) \int_0^1 u^{d-2}(1-u)\frac{\exp\left(-(s+s^2/2)G^2(u)\right)}{1+s}\,du$$

$$\geq d(d-1) \int_0^{1-\varepsilon} u^{d-2}(1-u)\frac{\exp\left(-(s+s^2/2)G^2(u)\right)}{1+s}\,du,$$

where $\varepsilon > 0$ is a parameter to be chosen later. By the next lemma, when $u \leq 1-\varepsilon$, we have $G^2(u) \leq 2\log(2/\varepsilon)$. Therefore,

$$\mathbb{P}\left(\frac{Y_1}{Y_2} \geq 1+s\right) \geq d(d-1) \int_0^{1-\varepsilon} u^{d-2}(1-u)\,du\frac{\exp\left(-(2s+s^2)\log(2/\varepsilon)\right)}{1+s}$$

$$\geq d(d-1) \int_0^{1-\varepsilon} u^{d-2}(1-u)\,du\frac{1}{1+s}\left(\frac{\varepsilon}{2}\right)^{4s}$$

$$= (1-\varepsilon)^d\left(1+\frac{d\varepsilon}{1-\varepsilon}\right)\frac{1}{1+s}\left(\frac{\varepsilon}{2}\right)^{4s}.$$

Let $\delta_{\mathbb{P}} \in (0,1)$ be our target failure probability. We choose

$$(1-\varepsilon)^d\left(1+\frac{d\varepsilon}{1-\varepsilon}\right) \geq 1-\frac{\delta_{\mathbb{P}}}{3} \quad \Leftarrow \quad \varepsilon = \frac{\delta_{\mathbb{P}}}{6d}.$$

With this choice of $\varepsilon$, we compute

$$\left(\frac{\varepsilon}{2}\right)^{4s} \geq 1-\frac{\delta_{\mathbb{P}}}{3} \quad \Leftarrow \quad \left(\frac{\delta_{\mathbb{P}}}{12d}\right)^{4s} \geq 1-\frac{\delta_{\mathbb{P}}}{3} \quad \Leftarrow \quad 4s\log\left(\frac{\delta_{\mathbb{P}}}{12d}\right) \geq -\frac{\delta_{\mathbb{P}}}{3} \quad \Leftarrow \quad s \leq \frac{\delta_{\mathbb{P}}}{12\log\left(\frac{12d}{\delta_{\mathbb{P}}}\right)}.$$

Note that for $s$ satisfying this condition, we automatically have $1/(1+s) \geq 1-\delta_{\mathbb{P}}/3$. Thus, we have

$$\frac{Y_1}{Y_2} \geq 1+\frac{\delta_{\mathbb{P}}}{12\log\left(\frac{8d}{\delta_{\mathbb{P}}}\right)} \quad \text{with probability at least } 1-\delta_{\mathbb{P}}.$$

$\square$

**Lemma A.10.** *Let* $F(y) = \text{erf}(y/\sqrt{2})\mathbb{1}\{y \geq 0\}$ *be the CDF of* $|\mathcal{N}(0,1)|$*, respectively. Let* $G = F^{-1}$*. If* $u \leq 1-1/(d\log d)$*, then* $G(u) \leq \sqrt{2\log(2/\varepsilon)}$*.*

*Proof.* Note that $G(u) \leq M$ iff $u \leq F(M)$ iff $1-u \geq 1-F(M)$ iff $1-u \geq \mathbb{P}(|Z| \geq M)$. In other words, our goal here is to find the smallest $M$ such that $\mathbb{P}(|Z| \geq M) \leq \varepsilon$. By the standard Gaussian concentration, we have $\mathbb{P}(|Z| \geq M) \leq 2\exp(-M^2/2)$. For the RHS to be upper bounded by $\varepsilon$, it suffices to choose $M \geq \sqrt{2\log(2/\varepsilon)}$. $\square$

**Lemma A.11.** *Let* $z_1, \ldots, z_m$ *be independent* $\mathcal{N}(0, I_d)$ *random vectors. Then, for any* $\varepsilon > 0$*, we have*

$$\mathbb{P}\left(\forall k \in [m], \left|\frac{\|z_k\|}{\mathbb{E}\|z_1\|} - 1\right| \leq \varepsilon\right) \geq 1 - 2me^{-\varepsilon^2 d/3}.$$

*Proof.* It is well-known that any 1-Lipschitz function of $\mathcal{N}(0, I_d)$ is 1-subgaussian (see, for example, Theorem 5.2.2 of [Ver18]). Hence, for any $s > 0$, we have

$$\mathbb{P}\left(\max_{k \in [m]} |\|z_k\| - \mathbb{E}\|z_k\|| \geq s\right) \leq \sum_{k=1}^m \mathbb{P}\left(|\|z_k\| - \mathbb{E}\|z_k\|| \geq s\right) \leq 2me^{-s^2/2}.$$

Set $s = \varepsilon\,\mathbb{E}\|z_k\|$ and the above becomes

$$\mathbb{P}\left(\forall k \in [m], \left|\frac{\|z_k\|}{\mathbb{E}\|z_1\|} - 1\right| \leq \varepsilon\right) \geq 1 - 2me^{-\varepsilon^2(\mathbb{E}\|z_1\|)^2/2}.$$

To complete the proof, it suffices to note that $\|z_1\|$ follows the $\chi_d$ distribution, and therefore we have $\mathbb{E}\|z_1\| \geq \sqrt{d}(1-2/d)$. $\square$

**Corollary A.12.** *Let $v \sim \mathrm{Unif}(\mathbb{S}^{d-1})$ and $w_1$ and $w_2$ denote the largest and second largest of $|v_1|, \ldots, |v_P|$. Suppose that $\frac{d}{\log d} \gtrsim \frac{\log^2(P/\delta_{\mathbb{P}})}{\delta_{\mathbb{P}}^2}$ and $\delta_{\mathbb{P}} \in (e^{-\log^C d}, 1)$. Then, we have*

$$\frac{w_1}{w_2} \geq 1 + \frac{\delta_{\mathbb{P}}}{100 \log(24P/\delta_{\mathbb{P}})} \quad \text{with probability at least } 1 - \delta_{\mathbb{P}}.$$

*Proof.* Let $z \sim \mathcal{N}(0, I_d)$ vectors. Note that $v \overset{d}{=} z/\|z\|$. By Lemma A.11, we have, for any $\delta_{\mathbb{P}} \in (0, 1)$, that

$$\left| \frac{\|z_k\|}{\mathbb{E}\|z_1\|} - 1 \right| \leq \sqrt{\frac{3\log(2/\delta_{\mathbb{P}})}{d}} \quad \text{with probability at least } 1 - \delta_{\mathbb{P}}.$$

Suppose that $|z_{k_1}|$ and $|z_{k_2}|$ are the largest and second largest of $|z_1|, \ldots, |z_P|$. By Lemma A.9 (with $d$ replaced by $P$), we have

$$\frac{|z_{k_1}|}{|z_{k_2}|} \geq 1 + \frac{\delta_{\mathbb{P}}}{12\log(12P/\delta_{\mathbb{P}})} \quad \text{with probability at least } 1 - \delta_{\mathbb{P}}.$$

Write

$$\frac{w_1}{w_2} \geq \left( 1 + \frac{\delta_{\mathbb{P}}}{12\log(12P/\delta_{\mathbb{P}})} \right) \frac{1 - \sqrt{\frac{3\log(2P/\delta_{\mathbb{P}})}{d}}}{1 + \sqrt{\frac{3\log(2d/\delta_{\mathbb{P}})}{d}}} \geq 1 + \frac{\delta_{\mathbb{P}}}{12\log(12P/\delta_{\mathbb{P}})} - 3\sqrt{\frac{3\log(2d/\delta_{\mathbb{P}})}{d}}.$$

In order to merge the last term into the second last term, it suffices to require

$$\frac{\delta_{\mathbb{P}}}{12\log(12P/\delta_{\mathbb{P}})} \geq 6\sqrt{\frac{3\log(2d/\delta_{\mathbb{P}})}{d}} \quad \Longleftarrow \quad \frac{d}{\log d} \gtrsim \frac{\log^2(P/\delta_{\mathbb{P}})}{\delta_{\mathbb{P}}^2}.$$

Then, with probability at least $1 - 2\delta_{\mathbb{P}}$, we have

$$\frac{w_1}{w_2} \geq 1 + \frac{\delta_{\mathbb{P}}}{24\log(12P/\delta_{\mathbb{P}})}$$

Replace $\delta_{\mathbb{P}}$ with $\delta_{\mathbb{P}}/2$ and we complete the proof. $\qquad\square$

**Lemma A.13** (Upper tail for the maximum). *Let $Z_1, \ldots, Z_d \sim \mathcal{N}(0, 1)$ be independent. We have the upper tail*

$$\mathbb{P}\left( \max_{i \in [d]} |Z_i| \geq \sqrt{2\log d} + s \right) \leq 2e^{-s^2/2}, \quad \forall s \geq 0.$$

*Proof.* For notational simplicity, put $Z^* = \max_{i \in [d]} Z_i$. By union bound and the Chernoff bound, we have for each $s, \theta > 0$,

$$\mathbb{P}(Z^* \geq s) = \mathbb{P}\left( \bigvee_{i=1}^{d} Z_i \geq s \right) \leq d\,\mathbb{P}(Z_1 \geq s) \leq d\frac{\mathbb{E}\,e^{\theta Z_1}}{e^{\theta s}} = de^{\theta^2/2 - \theta s}.$$

Choose $\theta = s$ to minimize the RHS, and we obtain $\mathbb{P}(Z^* \geq s) \leq e^{\log d - s^2/2}$. Replace $s$ with $\sqrt{2\log d + s^2}$ and this becomes

$$\mathbb{P}\left( Z^* \geq \sqrt{2\log d} + s \right) \leq \mathbb{P}\left( Z^* \geq \sqrt{2\log d + s^2} \right) \leq e^{-s^2/2}.$$

Use the fact $-\min_{i \in [d]} Z_i \overset{d}{=} \max_{i \in [d]} Z_i$ and we complete the proof. $\qquad\square$

# B Stage 1: recovery of the subspace and directions

In this section, we consider the stage where the second layer is fixed to be a small value and the first layer is trained using online spherical SGD. Let $v$ be a first-layer neuron that is good in the sense of (9). Assume w.l.o.g. that $v_1$ is the largest. Our goal in this section is to show $v$ will converge to close to $e_1$ with probability at least $1 - \delta_{\mathbb{P}}$ at the end of Stage 1.

For notational simplicity, let $l_{\text{corr}}$ and $\mathcal{L}_{\text{corr}}$ denote the per-sample and population correlation loss, respectively. By Lemma 2.1, we can write its update rule as

$$\hat{v}_{t+1} = v_t + \eta\tilde{\nabla}\mathcal{L}_{\text{corr}} + \eta Z_{t+1}, \quad v_{t+1} = \frac{\hat{v}_{t+1}}{\|\hat{v}_{t+1}\|},$$

where $Z_{t+1} = (I - vv^\top)(\nabla_v l_{\text{corr}}(x) - \nabla_v \mathcal{L}_{\text{corr}})$ and, by Lemma 2.1,

$$-\tilde{\nabla}_v\mathcal{L}_{\text{corr}} = -(I - vv^\top)\nabla_v\mathcal{L}$$
$$= 2\hat{\phi}_2^2 \sum_{k=1}^{P} v_k(I - vv^\top)e_k + \sum_{l \geq L}\sum_{k=1}^{P} l\hat{\phi}_l^2 v_k^{l-1}(I - vv^\top)e_k.$$

In particular, for each $k \in [d]$, we have[4]

$$\hat{v}_{t+1,k} = v_{t,k} + 2\eta\hat{\phi}_2^2\left(\mathbb{1}\{k \leq P\} - \|v_{\leq P}\|^2\right)v_k + L\eta\hat{\phi}_L^2\left(\mathbb{1}\{k \leq P\}v_k^{L-2} - \|v_{\leq P}\|_L^L\right)v_k$$
$$+ \eta\sum_{l>L} l\hat{\phi}_l^2\left(\mathbb{1}\{k \leq P\}v_k^{l-2} - \|v_{\leq P}\|_l^l\right)v_k + \eta Z_{t+1,k}$$

$$= v_{t,k} + \eta\mathbb{1}\{k \leq P\}\left(2\hat{\phi}_2^2 + L\hat{\phi}_L^2 v_k^{L-2} + \sum_{l>L} l\hat{\phi}_l^2 v_k^{l-2}\right)v_k$$
$$- \eta\left(2\hat{\phi}_2^2\|v_{\leq P}\|^2 + L\hat{\phi}_L^2\|v_{\leq P}\|_L^L + \sum_{l>L} l\hat{\phi}_l^2\|v_{\leq P}\|_l^l\right)v_k + \eta Z_{t+1,k}.$$

For notation simplicity, we define

$$\rho := 2\hat{\phi}_2^2\|v_{\leq P}\|^2 + L\hat{\phi}_L^2\|v_{\leq P}\|_L^L + \sum_{l>L} l\hat{\phi}_l^2\|v_{\leq P}\|_l^l. \tag{10}$$

Note that $\rho$ is independent of the coordinate $k$, and we can write

$$\hat{v}_{t+1,k} = v_{t,k} + \eta\mathbb{1}\{k \leq P\}\left(2\hat{\phi}_2^2 + L\hat{\phi}_L^2 v_k^{L-2} + \sum_{l>L} l\hat{\phi}_l^2 v_k^{l-2}\right)v_k - \eta\rho v_k + \eta Z_{t+1,k}. \tag{11}$$

For the martingale difference term $Z$, note that by Lemma 2.1, for any $u \in \mathbb{S}^{d-1}$, $\langle Z_{t+1}, u\rangle$ is a $(M_Z^2, \theta)$-subweibull variable with $M_Z = P^{1/2}$ and $1/\theta = 2(1 + Q)$. In particular, this implies

$$|\langle Z_{t+1}, u\rangle| \lesssim_\phi M_Z \log^{2(1+Q)}\log(d/\delta_{\mathbb{P}}) =: \hat{M}_Z, \quad \text{with probability at least } 1 - \delta_{\mathbb{P}}/d^C C \tag{12}$$

where $C > 0$ is any fixed constant.

In addition, we have the following lemma on the dynamics of $v_k^2$. The proof is routine calculation and is deferred to the end of this section (cf. Section B.3).

**Lemma B.1** (Dynamics of $v_k^2$). *For any first-layer neuron $v$ and $k \in [d]$, we have*

$$\hat{v}_{t+1,k}^2 = v_{t,k}^2 + 2\eta\gamma_{t,k}v_{t,k}^2 + 2\eta v_{t,k}Z_{t+1,k} + \xi_{t+1,k},$$

*where $\gamma_{k,t} := \mathbb{1}\{k \leq P\}\left(2\hat{\phi}_2^2 + L\hat{\phi}_L^2 v_k^{L-2} + \sum_{l>L} l\hat{\phi}_l^2 v_k^{l-2}\right) - \rho$ is a $\mathcal{F}_t$-measurable random variable with $|\gamma_{t,k}| \leq 2C_\phi^2$ and $(\xi_{t+1})_{t\in[T]}$ is (uniformly) bounded by $O_\phi(\eta^2\hat{M}_Z^2)$ with probability at least $1 - \delta_{\mathbb{P}}$.*

To proceed, we split Stage 1 into two substages. In Stage 1.1, we rely on the second-order terms to learn the relevant subspace. We will also show that the gap between largest and second-largest coordinates, which can be guaranteed with certain probability at initialization, is preserved throughout Stage 1.1. These give Stage 1.2 a nice starting point. Then, we show that in Stage 1.2, online spherical SGD can recover the directions using the $L$-th order terms.

---

[4]We will often drop the subscript $t$ when it is clear from the context.

## B.1 Stage 1.1: recovery of the subspace and preservation of the gap

In this subsection, first we show that the ratio $\|\boldsymbol{v}_{\leq P}\|^2 / \|\boldsymbol{v}_{>P}\|^2$ will grow from $\Omega(P/d)$ to $\Theta(1)$ within $\tilde{O}(dP)$ iterations. We will rely on the second-order terms and bound the influence of higher-order terms. This leads to the desired complexity. The next goal to show the initial randomness can be preserved. In our case, we only need the gap between the largest and the second-largest coordinate to be preserved, which will ensure that the neurons will not collapse to one single direction. Formally, we have the following lemma.

**Lemma B.2** (Stage 1.1). *Let $\boldsymbol{v} \in \mathbb{S}^{d-1}$ be an arbitrary first-layer neuron satisfying $\|\boldsymbol{v}\|_\infty \leq \log^2 d/d$, $\|\boldsymbol{v}_{\leq P}\|^2 / \|\boldsymbol{v}_{>P}\|^2 \gtrsim P/d$, and $v_p^2 = (1 + \delta_0) \operatorname{argmax}_{q \in [P]\setminus\{p\}} v_q^2$ at initialization. Let $\delta_{\mathbb{P}}$ be given. Suppose that*

$$P \gg_\phi \log^2 d \quad and \quad \eta \lesssim_\phi \frac{\delta_0^2}{dP \log d} \left( \frac{P}{\hat{M}_Z^2} \wedge \frac{1}{M_Z^2 \log^{\theta+1}(d/\delta_{\mathbb{P}})} \right) = \Theta_\phi \left( \frac{\delta_c^2}{dP} \right).$$

*Then, with probability at least $1 - O(\delta_{\mathbb{P}})$, we have*

$$\frac{\|\boldsymbol{v}_{\leq P}\|^2}{\|\boldsymbol{v}_{<P}\|^2} \geq 1 \quad within \ T = \frac{1 + o(1)}{4\hat{\phi}_2^2 \eta} \log \left( \frac{d}{P} \right) = \tilde{\Theta}(dP) \ iterations.$$

*In addition, at the end of Stage 1.1, we have $v_p^2 = (1 + \delta_0/2) \operatorname{argmax}_{q \in [P]\setminus\{p\}} v_q^2$.*

*Proof.* It suffices to combine Lemma B.4 and Lemma B.6. $\qquad\square$

To prove this lemma, we will use stochastic induction (cf. Section E), in particular, Lemma 4.1 and Lemma E.3. For example, to analyze the dynamics of $\|\boldsymbol{v}_{\leq P}\|^2 / \|\boldsymbol{v}_{>P}\|^2$, it suffices to write down the update rule of $\|\boldsymbol{v}_{\leq P}\|^2 / \|\boldsymbol{v}_{>P}\|^2$ and decompose it into a signal growth term, a higher-order error term, and a martingale difference term as in Lemma 4.1. Then, we bound the higher-order error terms, and estimate the covariance of the martingale difference terms, assuming the induction hypotheses.

The induction hypotheses we will maintain in this substage are the following:

$$\frac{\|\boldsymbol{v}_{t,\leq P}\|^2}{\|\boldsymbol{v}_{t,>P}\|^2} = \Theta(1)(1 + 4\hat{\phi}_2^2 \eta)^t \frac{\|\boldsymbol{v}_{0,\leq P}\|^2}{\|\boldsymbol{v}_{0,>P}\|^2}, \quad v_p^2 \leq \frac{\log^2 d}{P}. \tag{13}$$

They are established in Lemma B.4 and Lemma B.7.

### B.1.1 Learning the subspace

Now, we derive formulas for the dynamics of the ratio $\|\boldsymbol{v}_{\leq P}\|^2 / \|\boldsymbol{v}_{>P}\|^2$. As we have mentioned earlier, the goal here is separate the signal terms, martingale difference terms, and higher-order error terms.

**Lemma B.3** (Dynamics of the norm ratio). *Assume the induction hypotheses (13) at time $t \leq T$. Suppose that $\eta \leq \left( d\hat{M}_Z^2 \right)^{-1}$. Let $\boldsymbol{v}$ be an arbitrary first-layer neuron. Then, at time $t$, we have*

$$\frac{\|\boldsymbol{v}_{t+1,\leq P}\|^2}{\|\boldsymbol{v}_{t+1,>P}\|^2} = \frac{\|\boldsymbol{v}_{\leq P}\|^2}{\|\boldsymbol{v}_{>P}\|^2} \left( 1 + 4\eta\hat{\phi}_2^2 + 2\eta\varepsilon_v \right) + H_{t+1} + \xi_{t+1},$$

*where $\varepsilon_v := \sum_{l \geq L} l\hat{\phi}_l^2 \|\boldsymbol{v}_{\leq P}\|_l^l / \|\boldsymbol{v}_{\leq P}\|^2$, where $(H_{t+1})_t$ is a martingale difference sequence that is conditionally $\left( O_\phi((1 + 4\hat{\phi}_2^2 \eta)^t \frac{P}{d}), \theta \right)$-subweibull, and $(\xi_t)_t$ is an adapted process with $|\xi_{t+1}| \lesssim_\phi (1 + 4\hat{\phi}_2^2 \eta)^t \eta^2 \hat{M}_Z^2 P$ for all $t \in [T]$ with probability at least $1 - \delta_{\mathbb{P}}$.*

*Proof.* First, recall from Lemma B.1 that

$$\hat{v}_{t+1,k}^2 = v_{t,k}^2 + 2\eta\gamma_{t,k} v_{t,k}^2 + 2\eta v_{t,k} Z_{t+1,k} + \eta^2 \gamma_{t,k}^2 v_{t,k}^2 + \eta^2 Z_{t+1,k}^2 + 2\eta^2 \gamma_{t,k}^2 v_{t,k} Z_{t+1,k},$$

where $\gamma_{k,t} := \mathbb{1}\{k \leq P\}\left(2\hat{\phi}_2^2 + L\hat{\phi}_L^2 v_k^{L-2} + \sum_{l>L} l\hat{\phi}_l^2 v_k^{l-2}\right) - \rho$ is a $\mathcal{F}_t$-measurable random variable with $|\gamma_{t,k}| \leq 2C_\phi^2$. First, for $\|\hat{\boldsymbol{v}}_{\leq P}\|^2$, we have

$$\|\hat{\boldsymbol{v}}_{t+1,\leq P}\|^2 = \left(1 + 4\eta\hat{\phi}_2^2 - 2\eta\rho\right)\|\boldsymbol{v}_{\leq P}\|^2 + 2\eta\sum_{l\geq L} l\hat{\phi}_l^2 \|\boldsymbol{v}_{\leq P}\|^2 + 2\eta\langle\boldsymbol{v}_{\leq P}, \boldsymbol{Z}_{\leq P}\rangle$$

$$\underbrace{\pm 8C_\phi^4\eta^2\|\boldsymbol{v}_{\leq P}\|^2 \pm 2\eta^2\|\boldsymbol{Z}_{\leq P}\|^2}_{=:\xi_{\leq P,t+1}}.$$

Similarly, for $\|\hat{\boldsymbol{v}}_{>P}\|$, we have

$$\|\hat{\boldsymbol{v}}_{t+1,>P}\|^2 = \|\boldsymbol{v}_{>P}\|^2 - 2\eta\rho\|\boldsymbol{v}_{>P}\|^2 + 2\eta\langle\boldsymbol{v}_{>P}, \boldsymbol{Z}_{>P}\rangle \underbrace{\pm 8C_\phi^4\eta^2\|\boldsymbol{v}_{>P}\|^2 \pm 2\eta^2\|\boldsymbol{Z}_{>P}\|^2}_{=:\xi_{>P,t+1}}.$$

For notational simplicity, we also write $\varepsilon_v := \sum_{l\geq L} l\hat{\phi}_l^2 \|\boldsymbol{v}_{\leq P}\|_l^l / \|\boldsymbol{v}_{\leq P}\|^2$. Note that by Assumption 1 and the fact that $\|\boldsymbol{v}_{\leq P}\|_l^l / \|\boldsymbol{v}_{\leq P}\|^2 \leq 1$, $\varepsilon_v \leq C_\phi^2$. Since $\|\boldsymbol{v}_{\leq P}\| / \|\boldsymbol{v}_{>P}\| = \|\hat{\boldsymbol{v}}_{\leq P}\| / \|\hat{\boldsymbol{v}}_{>P}\|$, we have

$$\frac{\|\boldsymbol{v}_{t+1,\leq P}\|^2}{\|\boldsymbol{v}_{t+1,>P}\|^2} = \frac{\left(1 + 4\eta\hat{\phi}_2^2 - 2\eta\rho + 2\eta\varepsilon_v\right)\|\boldsymbol{v}_{\leq P}\|^2}{(1 - 2\eta\rho)\|\boldsymbol{v}_{>P}\|^2}\left(1 - \frac{2\eta\langle\boldsymbol{v}_{>P}, \boldsymbol{Z}_{>P}\rangle}{\|\hat{\boldsymbol{v}}_{t+1,>P}\|^2} - \frac{\xi_{>P}}{\|\hat{\boldsymbol{v}}_{t+1,>P}\|^2}\right)$$

$$+ \frac{2\eta\langle\boldsymbol{v}_{\leq P}, \boldsymbol{Z}_{\leq P}\rangle}{\|\hat{\boldsymbol{v}}_{t+1,>P}\|^2} + \frac{\xi_{\leq P}}{\|\hat{\boldsymbol{v}}_{t+1,>P}\|^2}$$

$$= \frac{\left(1 + 4\eta\hat{\phi}_2^2 - 2\eta\rho + 2\eta\varepsilon_v\right)\|\boldsymbol{v}_{\leq P}\|^2}{(1 - 2\eta\rho)\|\boldsymbol{v}_{>P}\|^2}$$

$$- \frac{\left(1 + 4\eta\hat{\phi}_2^2 - 2\eta\rho + 2\eta\varepsilon_v\right)\|\boldsymbol{v}_{\leq P}\|^2}{(1 - 2\eta\rho)\|\boldsymbol{v}_{>P}\|^2}\frac{2\eta\langle\boldsymbol{v}_{>P}, \boldsymbol{Z}_{>P}\rangle}{\|\hat{\boldsymbol{v}}_{t+1,>P}\|^2} + \frac{2\eta\langle\boldsymbol{v}_{\leq P}, \boldsymbol{Z}_{\leq P}\rangle}{\|\hat{\boldsymbol{v}}_{t+1,>P}\|^2}$$

$$- \frac{\left(1 + 4\eta\hat{\phi}_2^2 - 2\eta\rho + 2\eta\varepsilon_v\right)\|\boldsymbol{v}_{\leq P}\|^2}{(1 - 2\eta\rho)\|\boldsymbol{v}_{>P}\|^2}\frac{\xi_{>P}}{\|\hat{\boldsymbol{v}}_{t+1,>P}\|^2} + \frac{\xi_{\leq P}}{\|\hat{\boldsymbol{v}}_{t+1,>P}\|^2}$$

$$=: \mathtt{T}_1\left(\frac{\|\boldsymbol{v}_{t+1,\leq P}\|^2}{\|\boldsymbol{v}_{t+1,>P}\|^2}\right) + \mathtt{T}_2\left(\frac{\|\boldsymbol{v}_{t+1,\leq P}\|^2}{\|\boldsymbol{v}_{t+1,>P}\|^2}\right) + \mathtt{T}_3\left(\frac{\|\boldsymbol{v}_{t+1,\leq P}\|^2}{\|\boldsymbol{v}_{t+1,>P}\|^2}\right),$$

where each $\mathtt{T}_i$ represents one line. Note that, up to some higher order terms, $\mathtt{T}_1$ contains the signal terms and $\mathtt{T}_2$ contains the martingale difference terms. Now, our goal is to factor out those higher order terms.

For $\mathtt{T}_1$, recall that $|\rho| \leq C_\phi^2$ and use the fact that

$$\frac{1}{1+z} = 1 - z \pm 2z^2, \quad \forall |z| \leq 1/2, \tag{14}$$

to obtain

$$\mathtt{T}_1\left(\frac{\|\boldsymbol{v}_{t+1,\leq P}\|^2}{\|\boldsymbol{v}_{t+1,>P}\|^2}\right) = \frac{\|\boldsymbol{v}_{\leq P}\|^2}{\|\boldsymbol{v}_{>P}\|^2}\left(1 + 4\eta\hat{\phi}_2^2 - 2\eta\rho + 2\eta\varepsilon_v\right)\left(1 + 2\eta\rho \pm 4\eta^2\rho\right)$$

$$= \frac{\|\boldsymbol{v}_{\leq P}\|^2}{\|\boldsymbol{v}_{>P}\|^2}\left(1 + 4\eta\hat{\phi}_2^2 + 2\eta\varepsilon_v \pm 30C_\phi^4\eta^2\right).$$

Now, we consider

$$\mathtt{T}_2 = -\frac{\|\boldsymbol{v}_{\leq P}\|^2}{\|\boldsymbol{v}_{>P}\|^2}\frac{1 + 4\eta\hat{\phi}_2^2 - 2\eta\rho + 2\eta\varepsilon_v}{1 - 2\eta\rho}\frac{2\eta\langle\boldsymbol{v}_{>P}, \boldsymbol{Z}_{>P}\rangle}{\|\hat{\boldsymbol{v}}_{t+1,>P}\|^2} + \frac{2\eta\langle\boldsymbol{v}_{\leq P}, \boldsymbol{Z}_{\leq P}\rangle}{\|\hat{\boldsymbol{v}}_{t+1,>P}\|^2}.$$

First, we estimate the $1/\|\hat{\boldsymbol{v}}_{t+1,>P}\|^2$. We write

$$
\begin{aligned}
\frac{1}{\|\hat{\boldsymbol{v}}_{t+1,>P}\|^2} &= \frac{1}{(1-2\eta\rho)\|\boldsymbol{v}_{>P}\|^2 + 2\eta\langle\boldsymbol{v}_{>P}, \boldsymbol{Z}_{>P}\rangle + \xi_{>P,t+1}} \\
&= \frac{1}{(1-2\eta\rho)\|\boldsymbol{v}_{>P}\|^2}\left(1 - \frac{2\eta\langle\boldsymbol{v}_{>P}, \boldsymbol{Z}_{>P}\rangle + \xi_{>P,t+1}}{\|\hat{\boldsymbol{v}}_{t+1,>P}\|^2}\right).
\end{aligned}
$$

By (12), we have with probability at least $1 - \delta_\mathbb{P}$ that

$$
|\overline{\boldsymbol{v}_{>P}}\cdot\boldsymbol{Z}_{>P}| \wedge |\overline{\boldsymbol{v}_{\le P}}\cdot\boldsymbol{Z}_{\le P}| \wedge \max_{k\in[d]}|Z_k| \lesssim_\phi \hat{M}_Z.
$$

Note that the above conditions also imply

$$
\begin{aligned}
|\xi_{\le P}| &\le 8C_\phi^4\eta^2\|\boldsymbol{v}_{\le P}\|^2 + 2\eta^2 P\hat{M}_Z^2 \lesssim_\phi \eta^2 P\hat{M}_Z^2, \\
|\xi_{>P}| &\le 8C_\phi^4\eta^2\|\boldsymbol{v}_{>P}\|^2 + 2\eta^2 d\hat{M}_Z^2 \lesssim_\phi \eta^2 d\hat{M}_Z^2.
\end{aligned}
$$

By our definition of Stage 1.1, we have $\|\hat{\boldsymbol{v}}_{t+1,>P}\|^2 \ge 1/2$. Then, we have

$$
\begin{aligned}
\frac{1}{\|\hat{\boldsymbol{v}}_{t+1,>P}\|^2} &= \frac{1}{(1-2\eta\rho)\|\boldsymbol{v}_{>P}\|^2}\left(1 \pm 4\eta\hat{M}_Z \pm 32C_\phi^4\eta^2 d\hat{M}_Z^2\right) \\
&= \frac{1}{(1-2\eta\rho)\|\boldsymbol{v}_{>P}\|^2}\left(1 \pm O_\phi(\eta\hat{M}_Z)\right).
\end{aligned}
$$

Using the above two estimations of $1/\|\hat{\boldsymbol{v}}_{t+1,>P}\|^2$, we can rewrite $\mathsf{T}_2$ as

$$
\begin{aligned}
\mathsf{T}_2 &= -\frac{\|\boldsymbol{v}_{\le P}\|^2}{\|\boldsymbol{v}_{>P}\|^2}\frac{1+4\eta\hat{\phi}_2^2-2\eta\rho+2\eta\varepsilon_v}{1-2\eta\rho}\frac{2\eta\langle\boldsymbol{v}_{>P},\boldsymbol{Z}_{>P}\rangle}{(1-2\eta\rho)\|\boldsymbol{v}_{>P}\|^2}\left(1\pm O_\phi(\eta\hat{M}_Z)\right) \\
&\quad + \frac{2\eta\langle\boldsymbol{v}_{\le P},\boldsymbol{Z}_{\le P}\rangle}{(1-2\eta\rho)\|\boldsymbol{v}_{>P}\|^2}\left(1\pm O_\phi(\eta\hat{M}_Z)\right) \\
&= -\frac{\|\boldsymbol{v}_{\le P}\|^2}{\|\boldsymbol{v}_{>P}\|^2}\frac{1+4\eta\hat{\phi}_2^2-2\eta\rho+2\eta\varepsilon_v}{1-2\eta\rho}\frac{2\eta\langle\boldsymbol{v}_{>P},\boldsymbol{Z}_{>P}\rangle}{(1-2\eta\rho)\|\boldsymbol{v}_{>P}\|^2} + \frac{2\eta\langle\boldsymbol{v}_{\le P},\boldsymbol{Z}_{\le P}\rangle}{(1-2\eta\rho)\|\boldsymbol{v}_{>P}\|^2} \\
&\quad \pm 20\frac{\|\boldsymbol{v}_{\le P}\|^3}{\|\boldsymbol{v}_{>P}\|^3}\eta\hat{M}_Z O_\phi(\eta\hat{M}_Z) \pm 4\frac{\|\boldsymbol{v}_{\le P}\|}{\|\boldsymbol{v}_{>P}\|}\eta\hat{M}_Z O_\phi(\eta\hat{M}_Z) \\
&= -\frac{\|\boldsymbol{v}_{\le P}\|^2}{\|\boldsymbol{v}_{>P}\|^2}\frac{1+4\eta\hat{\phi}_2^2-2\eta\rho+2\eta\varepsilon_v}{1-2\eta\rho}\frac{2\eta\langle\boldsymbol{v}_{>P},\boldsymbol{Z}_{>P}\rangle}{(1-2\eta\rho)\|\boldsymbol{v}_{>P}\|^2} + \frac{2\eta\langle\boldsymbol{v}_{\le P},\boldsymbol{Z}_{\le P}\rangle}{(1-2\eta\rho)\|\boldsymbol{v}_{>P}\|^2} \\
&\quad \pm O_\phi\left(\frac{\|\boldsymbol{v}_{\le P}\|}{\|\boldsymbol{v}_{>P}\|}\eta^2\hat{M}_Z^2\right).
\end{aligned}
$$

Finally, consider the third term

$$
\mathsf{T}_3 := -\frac{\left(1+4\eta\hat{\phi}_2^2-2\eta\rho+2\eta\varepsilon_v\right)\|\boldsymbol{v}_{\le P}\|^2}{(1-2\eta\rho)\|\boldsymbol{v}_{>P}\|^2}\frac{\xi_{>P}}{\|\hat{\boldsymbol{v}}_{t+1,>P}\|^2} + \frac{\xi_{\le P}}{\|\hat{\boldsymbol{v}}_{t+1,>P}\|^2}.
$$

By our previous bounds on $\xi$, we have

$$
|\mathsf{T}_3| \le 64\frac{\|\boldsymbol{v}_{\le P}\|^2}{\|\boldsymbol{v}_{>P}\|^2}\frac{C_\phi^4\eta^2 d\hat{M}_Z^2}{\|\hat{\boldsymbol{v}}_{t+1,>P}\|^2} + 32C_\phi^4\eta^2 P\hat{M}_Z^2 \le 100C_\phi^4\eta^2\hat{M}_Z^2\left(P \vee \frac{\|\boldsymbol{v}_{\le P}\|^2}{\|\boldsymbol{v}_{>P}\|^2}d\right).
$$

Combine the above bounds, and we get

$$\frac{\|\boldsymbol{v}_{t+1,\leq P}\|^2}{\|\boldsymbol{v}_{t+1,>P}\|^2} = \frac{\|\boldsymbol{v}_{\leq P}\|^2}{\|\boldsymbol{v}_{>P}\|^2}\left(1 + 4\eta\hat{\phi}_2^2 + 2\eta\varepsilon_v\right)$$

$$- \frac{\|\boldsymbol{v}_{\leq P}\|^2}{\|\boldsymbol{v}_{>P}\|^2}\frac{1 + 4\eta\hat{\phi}_2^2 - 2\eta\rho + 2\eta\varepsilon_v}{1 - 2\eta\rho}\frac{2\eta\langle\boldsymbol{v}_{>P}, \boldsymbol{Z}_{>P}\rangle}{(1 - 2\eta\rho)\|\boldsymbol{v}_{>P}\|^2} + \frac{2\eta\langle\boldsymbol{v}_{\leq P}, \boldsymbol{Z}_{\leq P}\rangle}{(1 - 2\eta\rho)\|\boldsymbol{v}_{>P}\|^2}$$

$$\pm \frac{\|\boldsymbol{v}_{\leq P}\|^2}{\|\boldsymbol{v}_{>P}\|^2}30C_\phi^4\eta^2 \pm 1000C_\phi^4\frac{\|\boldsymbol{v}_{\leq P}\|}{\|\boldsymbol{v}_{>P}\|}\eta^2\hat{M}_Z^2\left(1 \vee \eta d\hat{M}_Z\right)$$

$$\pm 100C_\phi^4\eta^2\hat{M}_Z^2\left(P \vee \frac{\|\boldsymbol{v}_{\leq P}\|^2}{\|\boldsymbol{v}_{>P}\|^2}d\right).$$

Let $H_{t+1}$ denote the second line and $\xi_{t+1}$ denote the last two lines. Recall our induction hypothesis $\|\boldsymbol{v}_{t,\leq P}\|^2 / \|\boldsymbol{v}_{t,>P}\|^2 = \Theta(1)(1+4\hat{\phi}_2^2\eta)^t\|\boldsymbol{v}_{0,\leq P}\|^2 / \|\boldsymbol{v}_{0,>P}\|^2 = \Theta((1+4\hat{\phi}_2^2\eta)^t P/d)$. Meanwhile, note that $\frac{\|\boldsymbol{v}_{\leq P}\|}{\|\boldsymbol{v}_{>P}\|} \leq \frac{\|\boldsymbol{v}_{\leq P}\|^2}{\|\boldsymbol{v}_{>P}\|^2}\sqrt{\frac{d}{P}}$ Then, we compute

$$|\xi_{t+1}| \lesssim_\phi (1 + 4\hat{\phi}_2^2\eta)^t\frac{P}{d}\eta^2 + (1 + 4\hat{\phi}_2^2\eta)^t\frac{P}{d}\sqrt{\frac{d}{P}}\eta^2\hat{M}_Z^2\left(1 \vee \eta d\hat{M}_Z\right) + \eta^2\hat{M}_Z^2(1 + 4\hat{\phi}_2^2\eta)^t P$$

$$\lesssim_\phi (1 + 4\hat{\phi}_2^2\eta)^t\eta^2\hat{M}_Z^2\left(\sqrt{P}d\eta\hat{M}_Z \vee P\right)$$

$$\lesssim_\phi (1 + 4\hat{\phi}_2^2\eta)^t\eta^2\hat{M}_Z^2 P$$

Then, consider $H_{t+1}$. We have

$$|H_{t+1}| \leq \left|\frac{\|\boldsymbol{v}_{\leq P}\|^2}{\|\boldsymbol{v}_{>P}\|^2}\frac{1 + 4\eta\hat{\phi}_2^2 - 2\eta\rho + 2\eta\varepsilon_v}{1 - 2\eta\rho}\frac{2\eta\langle\boldsymbol{v}_{>P}, \boldsymbol{Z}_{>P}\rangle}{(1 - 2\eta\rho)\|\boldsymbol{v}_{>P}\|^2}\right| + \left|\frac{2\eta\langle\boldsymbol{v}_{\leq P}, \boldsymbol{Z}_{\leq P}\rangle}{(1 - 2\eta\rho)\|\boldsymbol{v}_{>P}\|^2}\right|$$

$$\lesssim (1 + 4\hat{\phi}_2^2\eta)^t\frac{P}{d}\eta\,|\langle\overline{\boldsymbol{v}_{>P}}, \boldsymbol{Z}_{>P}\rangle| + \left((1 + 4\hat{\phi}_2^2\eta)^t\frac{P}{d}\right)^{1/2}\eta\,|\langle\overline{\boldsymbol{v}_{\leq P}}, \boldsymbol{Z}_{\leq P}\rangle|.$$

Since both $|\langle\overline{\boldsymbol{v}_{>P}}, \boldsymbol{Z}_{>P}\rangle|$ and $|\langle\overline{\boldsymbol{v}_{\leq P}}, \boldsymbol{Z}_{\leq P}\rangle|$ are conditionally $(P, \theta)$-subweibull, $H_{t+1}$ is conditionally $\left(O_\phi((1 + 4\hat{\phi}_2^2\eta)^t\frac{P}{d}), \theta\right)$-subweibull. $\qquad\square$

With the above formula, we can now use Lemma 4.1 to analyze the dynamics of the ratio of the norms.

**Lemma B.4** (Learning the subspace). *Suppose that*

$$P \gg_\phi \log^2 d \quad and \quad \eta \lesssim_\phi \frac{1}{dP\log d}\left(\frac{P^2}{\hat{M}_Z^2} \wedge \frac{P}{M_Z^2\log^{\theta+1}(d/\delta_\mathbb{P})}\right) = \tilde{\Theta}_\phi\left(\frac{1}{dP}\right).$$

*Then, throughout Stage 1.1, we have*

$$\frac{(1 + 4\hat{\phi}_2^2\eta)^t}{2}\frac{\|\boldsymbol{v}_{0,\leq P}\|^2}{\|\boldsymbol{v}_{0,>P}\|^2} \leq \frac{\|\boldsymbol{v}_{\leq P}\|^2}{\|\boldsymbol{v}_{>P}\|^2} \leq \frac{3(1 + 4\hat{\phi}_2^2\eta)^t}{2}\frac{\|\boldsymbol{v}_{0,\leq P}\|^2}{\|\boldsymbol{v}_{0,>P}\|^2},$$

*and Stage 1.1 takes at most $(1 + o(1))(4\hat{\phi}_2^2\eta)^{-1}\log(d/P) = \tilde{O}_\phi(dP)$ iterations. For this result to hold for the $P$ good neurons (satisfying (9)), it suffices to replace $\delta_\mathbb{P}$ with $\delta_\mathbb{P}/P$.*

*Proof.* First, by Lemma B.3, we have for any $t \leq T$,

$$\frac{\|\boldsymbol{v}_{t+1,\leq P}\|^2}{\|\boldsymbol{v}_{t+1,>P}\|^2} = \frac{\|\boldsymbol{v}_{\leq P}\|^2}{\|\boldsymbol{v}_{>P}\|^2}\left(1 + 4\eta\hat{\phi}_2^2 + 2\eta\varepsilon_v\right) + H_{t+1} + \xi_{t+1},$$

where $\varepsilon_v := \sum_{l\geq L}l\hat{\phi}_l^2\|\boldsymbol{v}_{\leq P}\|_l^l / \|\boldsymbol{v}_{\leq P}\|^2$, where $(H_{t+1})_t$ is a martingale difference sequence that is conditionally $\left(O_\phi((1 + 4\hat{\phi}_2^2\eta)^t\frac{P}{d}), \theta\right)$-subweibull, and $(\xi_t)_t$ is an adapted process with

$|\xi_{t+1}| \lesssim_\phi (1 + 4\hat{\phi}_2^2\eta)^t \eta^2 \hat{M}_Z^2 P$ for all $t \in [T]$ with probability at least $1 - \delta_\mathbb{P}$. By our induction hypothesis $v_p^2 \leq \log^2 d/P$, we have

$$0 \leq \varepsilon_v := \frac{1}{\|\boldsymbol{v}_{\leq P}\|^2} \sum_{l \geq L} l\hat{\phi}_l^2 \sum_{k=1}^{P} v_k^l \leq \sum_{l \geq L} l\hat{\phi}_l^2 \|\boldsymbol{v}_{\leq P}\|_\infty^{l-2} \leq \frac{C_\phi^2 \log^{L-2} d}{P^{L/2-1}} =: \delta_v.$$

In particular, note that $\delta_v$ does not depend on $t$ and is $o(1)$. For notational simplicity, let $X_t := \|\boldsymbol{v}_{\leq P}\|^2 / \|\boldsymbol{v}_{>P}\|^2$, $x_t^- = (1 + 4\eta)^t x_0$ and $x_t^+ = (1 + 4\eta(1 + \delta_v))^t x_0$. $x^\pm$ will serve as the lower and upper bounds for the deterministic counterpart of $X$, since

$$\left(1 + 4\hat{\phi}_2^2\eta\right) X_t + \xi_{t+1} + H_{t+1} \leq X_{t+1} \leq \left(1 + 4\hat{\phi}_2^2\eta(1 + \delta_v)\right) X_t + \xi_{t+1} + H_{t+1}.$$

Moreover, note that for any $t \leq T$, we have

$$\frac{x_t^+}{x_t^-} = \left(\frac{1 + 4\hat{\phi}_2^2\eta(1 + \delta_v)}{1 + 4\hat{\phi}_2^2\eta}\right)^t = \left((1 + 4\eta(1 + \delta_v))\left(1 - 4\hat{\phi}_2^2\eta \pm O_\phi(\eta^2)\right)\right)^t$$

$$\leq \left(1 + 4\hat{\phi}_2^2\eta\delta_v \pm O_\phi(\eta^2)\right)^t$$

$$\leq \exp\left(O_\phi(1)\eta T (\delta_v + \eta)\right).$$

Since $T \lesssim_\phi \log d/\eta$, the above implies

$$1 \leq \frac{x_t^+}{x_t^-} \leq \exp\left(O_\phi(1)\log d (\delta_v + \eta)\right) \leq 1 + O_\phi(1)\log d (\delta_v + \eta) = 1 + o(1),$$

where the last (approximate) identity holds whenever

$$\delta_v \ll \frac{1}{\log d} \quad \Leftarrow \quad \frac{C_\phi^2 \log^{L-2} d}{P^{L/2-1}} \ll \frac{1}{\log d} \quad \Leftarrow \quad P \gg_\phi \log^2 d.$$

In particular, this implies that the (multiplicative) difference between $x_t^+$ and $x_t^-$ is small. Now, we apply Lemma 4.1 to $X_t$. In our case, we have

$$\Xi \lesssim_\phi \eta^2 \hat{M}_Z^2 P, \quad \sigma_Z^2 \lesssim_\phi \eta^2 \frac{P}{d} M_Z^2,$$

$\alpha = 4(1 + o(1))\hat{\phi}_2^2\eta$ and $X_0 = \Theta(P/d)$. Recall that $T \lesssim_\phi \log d/\eta$. Hence, to meet the conditions of Lemma 4.1, it suffices to choose

$$\eta^2 P\hat{M}_Z^2 \lesssim_\phi \frac{X_0}{T} \quad \Leftarrow \quad \eta \lesssim_\phi \frac{1}{d\hat{M}_Z^2 \log d},$$

$$\eta^2 \frac{P}{d} M_Z^2 \lesssim_\phi \frac{x_0^2}{T \log^{\theta+1}(T/\delta_\mathbb{P})} \quad \Leftarrow \quad \eta \lesssim_\phi \frac{P}{dM_Z^2 \log d \log^{\theta+1}(T/\delta_\mathbb{P})}.$$

Then, by Lemma 4.1, we have, with probability at least $1 - \Theta(\delta_\mathbb{P})$, $0.5x_t^- \leq X_t \leq 1.5x_t^+$. Since $x_t^+ = (1 + o(1))x_t^-$, this implies $0.5x_t \leq X_t \leq 2x_t$. To complete the proof, it suffices to note that for $x_t$ to grow from $\Theta(P/d)$ to $1$, the number of iterations needed is bounded by $(1 + o(1))(4\phi_2^2\eta)^{-1} \log (d/P)$. $\qquad\square$

### B.1.2 Preservation of the gap

Now, we show that the gap between the largest coordinate and the second-largest coordinate can be preserved in Stage 1.1. Let $p = \operatorname{argmax}_{i \in [P]} v_i^2(0)$ and consider the ratio $v_q^2/v_p^2$, where $q \in [P]$ is arbitrary. The proof is conceptually very similar to the previous one, except that we will use Lemma E.3 instead of Lemma 4.1.

**Lemma B.5.** *For $p = \operatorname{argmax}_{i \in [P]} v_i^2(0)$ and any $q \in [P]$, we have*

$$\frac{v_{t+1,q}^2}{v_{t+1,p}^2} \leq \frac{v_{t,q}^2}{v_{t,p}^2} + H_{t+1} + \xi_{t+1},$$

*where $(H_{t+1})_t$ is a martingale difference sequence that is conditionally $\left(O_\phi(\eta^2 dM_Z^2), \theta\right)$-subweibull, and $(\xi_t)_t$ is an adapted process that is uniformly bounded by $O_\phi(\eta^2 d\hat{M}_Z^2)$ with probability at least $1 - \delta_\mathbb{P}$.*

*Proof.* For notational simplicity, define $\rho_{t,q/p} := v_{t,q}^2/v_{t,p}^2$, Our goal is to upper bound $\rho_{t,q/p}$. Recall from Lemma B.1 that for any $k \leq P$, we have

$$\hat{v}_{t+1,k}^2 = v_{t,k}^2 + 2\eta\gamma_{t,k}v_{t,k}^2 + 2\eta v_{t,k}Z_{t+1,k} + \underbrace{\eta^2\gamma_{t,k}^2 v_{t,k}^2 + \eta^2 Z_{t+1,k}^2 + 2\eta^2\gamma_{t,k}v_{t,k}Z_{t+1,k}}_{=:\, \xi_{t+1,k}},$$

where $\gamma_{k,t} := 2\hat{\phi}_2^2 + L\hat{\phi}_L^2 v_k^{L-2} + \sum_{l>L} l\hat{\phi}_l^2 v_k^{l-2} - \rho$. Then, we compute

$$
\begin{aligned}
\rho_{t+1,q/p} &\leq \frac{(1+2\eta\gamma_{t,q})\,v_{t,q}^2 + 2\eta v_{t,q}Z_{t+1,q} + \xi_{t+1,q}}{(1+2\eta\gamma_{t,p})\,v_{t,p}^2 + 2\eta v_{t,p}Z_{t+1,p} + \xi_{t+1,p}} \\
&= \frac{(1+2\eta\gamma_{t,q})\,v_{t,q}^2}{(1+2\eta\gamma_{t,p})\,v_{t,p}^2} - \frac{(1+2\eta\gamma_{t,q})\,v_{t,q}^2}{(1+2\eta\gamma_{t,p})\,v_{t,p}^2}\frac{2\eta v_{t,p}Z_{t+1,p}}{\hat{v}_{t+1,p}^2} + \frac{2\eta v_{t,q}Z_{t+1,q}}{(1+2\eta\gamma_{t,p})\,v_{t,p}^2} \\
&\quad + \frac{\xi_{t+1,q}}{\hat{v}_{t+1,p}^2} - \frac{(1+2\eta\gamma_{t,q})\,v_{t,q}^2}{(1+2\eta\gamma_{t,p})\,v_{t,p}^2}\frac{\xi_{t+1,p}}{\hat{v}_{t+1,p}^2} - \frac{2\eta v_{t,q}Z_{t+1,q}}{(1+2\eta\gamma_{t,p})\,v_{t,p}^2}\frac{2\eta v_{t,p}Z_{t+1,p} + \xi_{t+1,p}}{\hat{v}_{t+1,p}^2} \\
&=: \mathtt{T}_1(\rho_{t+1,q/p}) + \mathtt{T}_2(\rho_{t+1,q/p}) + \mathtt{T}_3(\rho_{t+1,q/p}),
\end{aligned}
$$

where $\mathtt{T}_1$ contains the first term (signal term), $\mathtt{T}_2$ contains the next two terms (approximate martingale difference terms), and $\mathtt{T}_3$ contains the last line (higher order error terms).

First, for the first term, we compute

$$
\begin{aligned}
\mathtt{T}_1 &= \rho_{t,q/p}\,(1+2\eta\gamma_{t,q})\left(1 - 2\eta\gamma_{t,p}\left(1 - \frac{2\eta\gamma_{t,p}}{1+2\eta\gamma_{t,p}}\right)\right) \\
&= \rho_{t,q/p}\,(1+2\eta\gamma_{t,q})\left(1 - 2\eta\gamma_{t,p} \pm 5\eta^2\gamma_{t,p}^2\right) \\
&= \rho_{t,q/p}\left(1 + 2\eta\,(\gamma_{t,q} - \gamma_{t,p}) \pm 20\eta^2\left(\gamma_{t,p}^2 \vee \gamma_{t,q}^2\right)\right).
\end{aligned}
$$

Recall that $|\gamma_{t,k}| \leq 2C_\phi^2$ and note that $\gamma_{t,q} - \gamma_{t,p} = \sum_{l\geq L} l\hat{\phi}_l^2 v_q^{l-2} - \sum_{l\geq L} l\hat{\phi}_l^2 v_p^{l-2} \leq 0$. Hence,

$$\mathtt{T}_1 \leq \rho_{t,q/p}\left(1 + 80C_\phi^4\eta^2\right).$$

Now, consider the martingale difference term

$$\mathtt{T}_2 := -\rho_{t,q/p}\frac{1+2\eta\gamma_{t,q}}{1+2\eta\gamma_{t,p}}\frac{2\eta v_{t,p}Z_{t+1,p}}{\hat{v}_{t+1,p}^2} + \frac{2\eta v_{t,q}Z_{t+1,q}}{(1+2\eta\gamma_{t,p})\,v_{t,p}^2}.$$

We rewrite the denominator as

$$
\begin{aligned}
\frac{1}{\hat{v}_{t+1,p}^2} &= \frac{1}{v_{t,p}^2 + 2\eta\gamma_{t,p}v_{t,p}^2 + 2\eta v_{t,p}Z_{t+1,p} + \xi_{t+1,p}} \\
&= \frac{1}{v_{t,p}^2 + 2\eta\gamma_{t,p}v_{t,p}^2} - \frac{1}{v_{t,p}^2 + 2\eta\gamma_{t,p}v_{t,p}^2}\frac{2\eta v_{t,p}Z_{t+1,p} + \xi_{t+1,p}}{\hat{v}_{t+1,p}^2}.
\end{aligned}
$$

By (12), with probability at least $1 - \delta_{\mathbb{P}}/d^C$, we have

$$|\xi_{p,t+1}| \lesssim_\phi \eta^2 v_{t,p}^2 + \eta^2 \hat{M}_Z^2 + \eta^2|v_{t,p}|\hat{M}_Z \lesssim_\phi \eta^2\hat{M}_Z^2.$$

Therefore,

$$\frac{1}{\hat{v}_{t+1,p}^2} = \frac{1}{v_{t,p}^2 + 2\eta\gamma_{t,p}v_{t,p}^2} \pm O_\phi\left(\frac{1}{v_{t,p}^2}\frac{\eta\hat{M}_Z}{v_{t,p}}\right).$$

Then, we can rewrite $\mathtt{T}_2$ as

$$
\begin{aligned}
\mathtt{T}_2 &= -\rho_{t,q/p}\frac{1+2\eta\gamma_{t,q}}{1+2\eta\gamma_{t,p}}\frac{2\eta v_{t,p}Z_{t+1,p}}{v_{t,p}^2 + 2\eta\gamma_{t,p}v_{t,p}^2} + \frac{2\eta v_{t,q}Z_{t+1,q}}{(1+2\eta\gamma_{t,p})\,v_{t,p}^2} \pm O_\phi\left(\eta^2\frac{\hat{M}_Z^2}{v_{t,p}^2}\right) \\
&=: H_{t+1} \pm O_\phi\left(\eta^2 d\hat{M}_Z^2\right).
\end{aligned}
$$

Note that $H_{t+1}$ is a martingale difference term with

$$|H_{t+1}| \lesssim_\phi \frac{\eta|Z_{t+1,p}|}{v_{t,p}} + \frac{\eta v_{t,q}|Z_{t+1,q}|}{v_{t,p}^2} \lesssim_\phi \eta\sqrt{d}\,(|Z_{t+1,p}| + |Z_{t+1,q}|).$$

Since $Z_{t+1,p}$ and $Z_{t+1,q}$ are both conditionally $(M_Z^2, \theta)$-subweibull, $H_{t+1}$ is conditionally $\left(O_\phi(\eta^2 d M_Z^2), \theta\right)$-subweibull. Finally, consider

$$\mathrm{T}_3 := \frac{\xi_{t+1,q}}{\hat{v}_{t+1,p}^2} - \frac{(1 + 2\eta\gamma_{t,q}) v_{t,q}^2}{(1 + 2\eta\gamma_{t,p}) v_{t,p}^2} \frac{\xi_{t+1,p}}{\hat{v}_{t+1,p}^2} - \frac{2\eta v_{t,q} Z_{t+1,q}}{(1 + 2\eta\gamma_{t,p}) v_{t,p}^2} \frac{2\eta v_{t,p} Z_{t+1,p} + \xi_{t+1,p}}{\hat{v}_{t+1,p}^2}.$$

Since $|\xi_{q,t+1}| \vee |\xi_{p,t+1}| \leq \eta^2 \hat{M}_Z^2$ and $|Z_{t+1,p}| \vee |Z_{t+1,q}| \leq \hat{M}_Z$, we have

$$|\mathrm{T}_3| \lesssim_\phi \eta^2 d \hat{M}_Z^2.$$

Combining the above bounds, we get

$$\rho_{t+1,q/p} \leq \rho_{t,q/p} \left(1 + 80 C_\phi^4 \eta^2\right) + H_{t+1} + O_\phi\left(\eta^2 d \hat{M}_Z^2\right) = \rho_{t,q/p} + H_{t+1} + O_\phi\left(\eta^2 d \hat{M}_Z^2\right).$$

$\square$

**Lemma B.6** (Preservation of the gap). *Consider* $\delta_c \in (0,1)$, $p = \mathrm{argmax}_{i \in [P]} v_i^2(0)$ *and any* $q \in [P]$. *Suppose that*

$$\eta \lesssim_\phi \frac{\delta_c^2}{dP \log d} \left(\frac{P}{\hat{M}_Z^2} \wedge \frac{P}{M_Z^2 \log^{\theta+1}(T/\delta_\mathbb{P})}\right) = \tilde{\Theta}_\phi\left(\frac{\delta_c}{dP}\right).$$

*Then, we have*

$$\sup_{t \leq T} \left(\frac{v_{t,q}^2}{v_{t,p}^2} - \frac{v_{0,q}^2}{v_{0,p}^2}\right) \leq \delta_c \quad \text{with probability at least } 1 - \delta_\mathbb{P}.$$

*Proof.* By Lemma B.5, we have

$$\frac{v_{t+1,q}^2}{v_{t+1,p}^2} \leq \frac{v_{t,q}^2}{v_{t,p}^2} + H_{t+1} + \xi_{t+1},$$

where $(H_{t+1})_t$ is a martingale difference sequence that is conditionally $\left(O_\phi(\eta^2 d M_Z^2), \theta\right)$-subweibull, and $(\xi_t)_t$ is an adapted process that is uniformly bounded by $O_\phi(\eta^2 d \hat{M}_Z^2)$ with probability at least $1 - \delta_\mathbb{P}$. Hence, by Lemma E.3, we have

$$\sup_{t \leq T} \left(\frac{v_{t,q}^2}{v_{t,p}^2} - \frac{v_{0,q}^2}{v_{0,p}^2}\right) \lesssim_\phi T\eta^2 d \hat{M}_Z^2 + \sqrt{\eta^2 d M_Z^2 T \log^{\theta+1}(T/\delta_\mathbb{P})}$$

$$\lesssim_\phi \eta d \hat{M}_Z^2 \log d + \sqrt{\eta d M_Z^2 \log^{\theta+1}(T/\delta_\mathbb{P}) \log d}.$$

For the RHS to be bounded by $\delta_c \in (0,1)$, it suffices to require

$$\eta d \hat{M}_Z^2 \log d \lesssim_\phi \delta_c \quad \Leftarrow \quad \eta \lesssim_\phi \frac{\delta_c}{d \hat{M}_Z^2 \log d},$$

$$\sqrt{\eta d M_Z^2 \log^{\theta+1}(T/\delta_\mathbb{P}) \log d}. \lesssim_\phi \delta_c \quad \Leftarrow \quad \eta \lesssim_\phi \frac{\delta_c^2}{d M_Z^2 \log^{\theta+1}(T/\delta_\mathbb{P}) \log d}.$$

$\square$

### B.1.3 Other induction hypotheses

In this subsection, we verify the induction hypothesis: $v_p^2 \lesssim \log^2 d / P$ for all $p \in [P]$. This condition is used to ensure the influence of the higher-order term is small compared to the influence of the second-order terms.

**Lemma B.7** (Upper bound on $v_p^2$). *Suppose that*

$$\eta \lesssim_\phi \frac{\log d}{dP} \left(\frac{P}{\hat{M}_Z^2} \wedge \frac{P}{M_Z^2 \log^{\theta+1}(d/\delta_\mathbb{P})}\right).$$

*Then, throughout Stage 1, we have* $v_p^2 \lesssim \log^2 d / P$.

*Proof.* First, by Lemma B.1, for any $p \leq P$, we have

$$\hat{v}_{t+1,p}^2 \leq v_{t,p}^2 + 2\eta\left(2\hat{\phi}_2^2 + \sum_{l \geq L} l\hat{\phi}_l^2 v_k^{l-2}\right)v_{t,p}^2 + 2\eta v_{t,p}Z_{t+1,p}$$

$$+ \underbrace{\eta^2 \gamma_{t,p}^2 v_{t,p}^2 + \eta^2 Z_{t+1,p}^2 + 2\eta^2 \gamma_{t,p} v_{t,p} Z_{t+1,p}}_{=:\xi_{t+1}}$$

$$\leq v_{t,p}^2 + 4\hat{\phi}_2^2 \eta\left(1 + \underbrace{\frac{C_\phi^2}{2\hat{\phi}_2^2}\frac{\log^{L-2}d}{P^{L/2-1}}}_{=:\,\delta_v}\right)v_{t,p}^2 + 2\eta v_{t,p}Z_{t+1,p} + \xi_{t+1}$$

where $\gamma_{p,t} := 2\hat{\phi}_2^2 + L\hat{\phi}_L^2 v_k^{L-2} + \sum_{l>L} l\hat{\phi}_l^2 v_k^{l-2} - \rho$ is a $\mathcal{F}_t$-measurable random variable with $|\gamma_{t,p}| \leq 2C_\phi^2$. By (12), with probability at least $1 - \delta_{\mathbb{P}}/T$, we have

$$|\xi_{t+1}| \lesssim_\phi \eta^2 v_{t,p}^2 + \eta^2 \hat{M}_Z^2 + \eta^2 |v_{t,p}|\hat{M}_Z \lesssim_\phi \eta^2 \hat{M}_Z^2$$

We maintain the induction hypothesis $v_{t,p}^2 \leq 2(1 + 4\hat{\phi}_2^2\eta(1 + \delta_v))^t \log^2 d/d$. Under this induction hypothesis, we have

$$|2\eta v_{t,p}Z_{t+1,p}| \lesssim \eta\sqrt{(1 + 4\bar{\phi}_2^2\eta(1 + \delta_v))\log^2 d/d}|Z_{t+1,p}|,$$

and therefore, is $\left(O_\phi\left(\eta^2(1 + 4\bar{\phi}_2^2\eta(1 + \delta_v))^t v_{0,p}^2 M_Z^2, \theta\right)\right)$-subweibull. Using the language of Lemma 4.1, we have

$$\Xi \lesssim_\phi \eta^2 \hat{M}_Z^2 \quad \text{and} \quad \sigma_Z^2 \lesssim_\phi \eta^2(1 + 4\bar{\phi}_2^2\eta(1 + \delta_v))^t \frac{\log^2 d}{d}M_Z^2.$$

Therefore, as long as

$$\eta^2 \hat{M}_Z^2 \lesssim_\phi \frac{\log^2 d/d}{T} \quad \Leftarrow \quad \eta \lesssim_\phi \frac{\log d}{d\hat{M}_Z^2},$$

$$\eta^2 \frac{\log^2 d}{d}M_Z^2 \lesssim_\phi \frac{x_0^2}{T\log^{\theta+1}(T/\delta_{\mathbb{P}})} \quad \Leftarrow \quad \eta \lesssim_\phi \frac{\log d}{dM_Z^2 \log^{\theta+1}(T/\delta_{\mathbb{P}})},$$

we have $v_{t,p}^2 \leq 2(1 + 4\hat{\phi}_2^2\eta(1 + \delta_v))^t \log^2 d/d$ throughout Stage 1. In particular, by Lemma B.4, this implies

$$v_{t,p}^2 \lesssim \exp^{1+\delta_v}\left(4\hat{\phi}_2^2\eta T\right)\frac{\log^2 d}{d} \lesssim \frac{\log^2 d}{P}.$$

$\square$

## B.2 Stage 1.2: recovery of the directions

Let $v$ be an arbitrary first-layer neuron. Assume w.l.o.g. that $v_1^2$ is the largest at initialization and $v_{0,1}^2/\max_{2\leq k\leq P} v_{0,k}^2 \geq 1 + \delta_0$. By Lemma B.2, we know this gap can be approximately preserved in the sense that $v_{0,1}^2/\max_{2\leq k\leq P} v_{0,k}^2 \geq 1 + \delta_0/2$ holds. For notational simplicity, we will drop the factor $1/2$ in the sequel. Moreover, since we use symmetric initialization, we can further assume that $v_1 > 0$. In this subsection, we show that $v_1^2$ will grow from $\Omega(1/P)$ to $3/4$ and then to close to 1. Formally, we prove the following lemma.

**Lemma B.8** (Stage 1.2). *Let $v \in \mathbb{S}^{d-1}$ be an arbitrary first-layer neuron satisfying $v_{T_1,1}^2 \geq c/P$ and $v_{T_1,1}^2/\max_{2\leq k\leq P} v_{T_1,k}^2 \geq 1 + c$ for some small universal constant $c > 0$. Let $\delta_{\mathbb{P}} \in (0,1)$ and $\varepsilon_v > 0$ be given. Suppose that we choose*

$$\eta \lesssim_\phi \frac{\delta_0}{dP^{L/2}}\left(\frac{P}{\hat{M}_z^2} \wedge \frac{d}{M_Z^2}\frac{1}{\log^{\theta+1}(d/\delta_{\mathbb{P}})}\right) \wedge \frac{\varepsilon_*}{dP}\left(\frac{P}{\hat{M}_Z^2 \log(1/\varepsilon_*)} \wedge \frac{\varepsilon_* dP}{M_Z^2 \log d \log^{\theta+1}(d/\delta_{\mathbb{P}})}\right)$$

*Then, with probability at least $1 - O(\delta_{\mathbb{P}})$, we have $v_1^2 \geq 1 - \varepsilon_v$ within $O_\phi\left(\left(P^{L/2-1} + \log(1/\varepsilon_v)\right)/\eta\right)$ iterations.*

*Proof.* It suffices to combine Lemma B.10 and Lemma B.11. □

**Lemma B.9** (Dynamics of $v_1^2$)**.** *We have*

$$v_{t+1,1}^2 \geq v_{t,1}^2 \left( 1 + 2\eta \sum_{l \geq L} l\hat{\phi}_l^2 v_1^{l-2} - 2\eta \sum_{l \geq L} l\hat{\phi}_l^2 \left\| \boldsymbol{v}_{t,\leq P} \right\|_l^l \right) + H_{t+1} + \tilde{\xi}_{t+1},$$

*where $H_{t+1}$ is a martingale difference term that is conditionally $(O_\phi(\eta^2 v_{t,1}^2 M_Z^2), \theta)$-subweibull and $\xi_{t+1}$ is bounded by $O_\phi(\eta^2 d\hat{M}_Z^2 v_{t,1}^2)$ uniformly over $t \in [T]$ with probability at least $1 - \delta_{\mathbb{P}}$.*

*Proof.* Recall from Lemma B.1 that

$$\hat{v}_{t+1,k}^2 = v_{t,k}^2 + 2\eta\gamma_{t,k}v_{t,k}^2 + 2\eta v_{t,k}Z_{t+1,k} + \xi_{t+1,k},$$

where $\gamma_{k,t} := \mathbb{1}\{k \leq P\} \left( 2\hat{\phi}_2^2 + L\hat{\phi}_L^2 v_k^{L-2} + \sum_{l>L} l\hat{\phi}_l^2 v_k^{l-2} \right) - \rho$ is a $\mathcal{F}_t$-measurable random variable with $|\gamma_{t,k}| \leq 2C_\phi^2$ and $(\xi_{t+1})_{t\in[T]}$ is (uniformly) bounded by $O_\phi(\eta^2 \hat{M}_Z^2)$ with probability at least $1 - \delta_{\mathbb{P}}$. Sum over $k \in [d]$ and we get

$$\left\| \hat{\boldsymbol{v}}_{t+1} \right\|^2 = 1 + 2\eta \sum_{k=1}^d \left( \mathbb{1}\{k \leq P\} \left( 2\hat{\phi}_2^2 + L\hat{\phi}_L^2 v_k^{L-2} + \sum_{l>L} l\hat{\phi}_l^2 v_k^{l-2} \right) - \rho \right) v_{t,k}^2 + 2\eta \langle \boldsymbol{v}_t, \boldsymbol{Z}_{t+1} \rangle + \xi_{t+1}'$$

$$= 1 + 2\eta \left( 2\hat{\phi}_2^2 \left\| \boldsymbol{v}_{t,\leq P} \right\|^2 + \sum_{l \geq L} l\hat{\phi}_l^2 \left\| \boldsymbol{v}_{t,\leq P} \right\|_l^l - \rho \left\| \boldsymbol{v}_t \right\|^2 \right) + 2\eta \langle \boldsymbol{v}_t, \boldsymbol{Z}_{t+1} \rangle + \xi_{t+1}'$$

$$\leq \underbrace{1 + 2\eta \left( 2\hat{\phi}_2^2 - \rho \right) + 2\eta \sum_{l \geq L} l\hat{\phi}_l^2 \left\| \boldsymbol{v}_{t,\leq P} \right\|_l^l}_{=:N_v^2} + 2\eta \langle \boldsymbol{v}_t, \boldsymbol{Z}_{t+1} \rangle + \xi_{t+1}',$$

where $\xi_{t+1}$ is bounded by $O_\phi(\eta^2 d\hat{M}_Z^2)$. Recall from (12) that $|\langle \boldsymbol{v}_t, \boldsymbol{Z}_{t+1} \rangle| \lesssim_\phi \hat{M}_Z$ and choose $\eta \leq (d\hat{M}_Z^2)^{-1}$. As a result,

$$\frac{1}{\left\| \hat{\boldsymbol{v}}_{t+1} \right\|^2} \geq \frac{1}{N_v^2} \left( 1 - \frac{2\eta \langle \boldsymbol{v}_t, \boldsymbol{Z}_{t+1} \rangle + \xi_{t+1}'}{N_v^2} \left( 1 - \frac{2\eta \langle \boldsymbol{v}_t, \boldsymbol{Z}_{t+1} \rangle + \xi_{t+1}'}{N_v^2 + 2\eta \langle \boldsymbol{v}_t, \boldsymbol{Z}_{t+1} \rangle + \xi_{t+1}'} \right) \right)$$

$$\geq \frac{1}{N_v^2} - \frac{1}{N_v^2} \frac{2\eta \langle \boldsymbol{v}_t, \boldsymbol{Z}_{t+1} \rangle}{N_v^2} \pm O_\phi(\eta^2 d\hat{M}_Z^2).$$

Meanwhile, we have

$$\hat{v}_{t+1,1}^2 = v_{t,1}^2 + 2\eta \left( 2\hat{\phi}_2^2 - \rho \right) v_{t,1}^2 + 2\eta \sum_{l \geq L} l\hat{\phi}_l^2 v_1^l + 2\eta v_{t,1}Z_{t+1,1} + \xi_{t+1,1},$$

where $|\xi_{t+1,1}| \lesssim_\phi \eta^2 \hat{M}_Z^2$. Therefore,

$$v_{t+1,1}^2 \geq \hat{v}_{t+1,1}^2 \left( \frac{1}{N_v^2} - \frac{1}{N_v^2} \frac{2\eta \langle \boldsymbol{v}_t, \boldsymbol{Z}_{t+1} \rangle}{N_v^2} \pm O_\phi(\eta^2 d\hat{M}_Z^2) \right)$$

$$= \frac{\hat{v}_{t+1,1}^2}{N_v^2} - \frac{\hat{v}_{t+1,1}^2}{N_v^2} \frac{2\eta \langle \boldsymbol{v}_t, \boldsymbol{Z}_{t+1} \rangle}{N_v^2} \pm O_\phi(\eta^2 d\hat{M}_Z^2 v_{t,1}^2)$$

$$= \frac{v_{t,1}^2 + 2\eta \left( 2\hat{\phi}_2^2 - \rho \right) v_{t,1}^2 + 2\eta \sum_{l \geq L} l\hat{\phi}_l^2 v_1^l}{N_v^2}$$

$$+ \frac{2\eta v_{t,1}Z_{t+1,1}}{N_v^2} - \frac{v_{t,1}^2 + 2\eta \left( 2\hat{\phi}_2^2 - \rho \right) v_{t,1}^2 + 2\eta \sum_{l \geq L} l\hat{\phi}_l^2 v_1^l}{N_v^2} \frac{2\eta \langle \boldsymbol{v}_t, \boldsymbol{Z}_{t+1} \rangle}{N_v^2}$$

$$\pm O_\phi \left( \eta^2 d\hat{M}_Z^2 v_{t,1}^2 \right)$$

$$=: \mathtt{T}_1 \left( v_{t+1,1}^2 \right) + \mathtt{T}_2 \left( v_{t+1,1}^2 \right) \pm O_\phi \left( \eta^2 d\hat{M}_Z^2 v_{t,1}^2 \right),$$

where we have used the fact that $v_{t,1}^2 \gtrsim 1/P$ to merge error terms of form $\eta^2 \hat{M}_Z^2$ into $O_\phi\left(\eta^2 d\hat{M}_Z^2 v_{t,1}^2\right)$. Meanwhile, for $\texttt{T}_2$, we have $|\texttt{T}_2| \lesssim_\phi |\eta v_{t,1} Z_{t+1,1}| + \left|\eta v_{t,1}^2 \langle \boldsymbol{v}_t, \boldsymbol{Z}_{t+1}\rangle\right|$. Therefore, it is a $(O_\phi(\eta^2 v_{t,1}^2 M_Z^2), \theta)$-subweibull. For the signal term $\texttt{T}_1$, by (14), we have

$$
\begin{aligned}
\texttt{T}_1 &= \frac{v_{t,1}^2 + 2\eta\left(2\hat{\phi}_2^2 - \rho\right)v_{t,1}^2 + 2\eta \sum_{l\geq L} l\hat{\phi}_l^2 v_1^l}{1 + 2\eta\left(2\hat{\phi}_2^2 - \rho\right) + 2\eta \sum_{l\geq L} l\hat{\phi}_l^2 \|\boldsymbol{v}_{t,\leq P}\|_l^l} \\
&= v_{t,1}^2 \left(1 + 2\eta\left(2\hat{\phi}_2^2 - \rho\right) + 2\eta \sum_{l\geq L} l\hat{\phi}_l^2 v_1^{l-2}\right)\left(1 - 2\eta\left(2\hat{\phi}_2^2 - \rho\right) - 2\eta \sum_{l\geq L} l\hat{\phi}_l^2 \|\boldsymbol{v}_{t,\leq P}\|_l^l \pm O_\phi\left(\eta^2\right)\right) \\
&= v_{t,1}^2 \left(1 + 2\eta \sum_{l\geq L} l\hat{\phi}_l^2 v_1^{l-2} - 2\eta \sum_{l\geq L} l\hat{\phi}_l^2 \|\boldsymbol{v}_{t,\leq P}\|_l^l \pm O_\phi\left(\eta^2\right)\right).
\end{aligned}
$$

Combine the above estimations, set $H_{t+1} = \texttt{T}_2$, and we complete the proof. $\qquad\square$

**Lemma B.10** (Weak reocovery of directions). *Suppose that we choose*

$$
\eta \lesssim_\phi \frac{\delta_0}{dP^{L/2}} \left(\frac{P}{\hat{M}_z^2} \wedge \frac{d}{M_Z^2} \frac{1}{\log^{\theta+1}(d/\delta_{\mathbb{P}})}\right).
$$

*Then with probability at least $1 - O(\delta_{\mathbb{P}})$, we will have $v_1^2 \geq 3/4$ within the following number of iterations:*

$$
O_\phi\left(\frac{P^{L/2-1}}{\eta}\right) = O_\phi\left(PdP^{L-2}\left(\frac{P}{\hat{M}_z^2} \wedge \frac{d}{M_Z^2} \frac{1}{\log^{\theta+1}(d/\delta_{\mathbb{P}})}\right)^{-1}\right).
$$

**Remark.** Note that when $\hat{M}_Z^2, M_Z^2 = \tilde{O}_\phi(P)$, then the above is roughly $P \times (dP^{L-2})$. The $dP^{L-2}$ is the usual bound for online SGD when the noise has order $d$ instead of $P$. The first $P$ comes from the fact that there are $P$ directions. $\qquad\clubsuit$

*Proof.* By Lemma B.9, we have

$$
v_{t+1,1}^2 \geq v_{t,1}^2 \left(1 + 2\eta \sum_{l\geq L} l\hat{\phi}_l^2 v_1^{l-2} - 2\eta \sum_{l\geq L} l\hat{\phi}_l^2 \|\boldsymbol{v}_{t,\leq P}\|_l^l\right) + H_{t+1} + \tilde{\xi}_{t+1},
$$

where $H_{t+1}$ is a martingale difference term that is conditionally $(O_\phi(\eta^2 M_Z^2 v_{t,1}^2), \theta)$-subweibull, and $\xi_{t+1}$ is bounded by $O_\phi(\eta^2 d\hat{M}_Z^2 v_{t,1}^2)$ for all $t \in [T]$ with probability at least $1 - \delta_{\mathbb{P}}$. For the signal term, we write

$$
v_1^{l-2} - \|\boldsymbol{v}\|_l^l = v_1^{l-2} - v_1^l - \sum_{k=2}^P v_k^l = v_1^{l-2}(1 - v_1^2) - \left(\|\boldsymbol{v}_{\leq P}\|^2 - v_1^2\right)\sum_{k=2}^P \frac{v_k^2}{\|\boldsymbol{v}_{\leq P}\|^2 - v_1^2} v_k^{l-2}.
$$

Note that the last term is a weighted average of $v_k^{l-2}$. Similar to the proof in Section B.1.2, one can show that the induction hypothesis $v_1^2 / \max_{2\leq k\leq P} v_k^2 \geq 1 + \delta_0/2$ remains true,[5] which gives

$$
\sum_{k=2}^P \frac{v_k^2}{\|\boldsymbol{v}_{\leq P}\|^2 - v_1^2} v_k^{l-2} \leq \left(\max_{2\leq k\leq P} v_k^2\right)^{L/2-1} \leq \left(\frac{v_1^2}{1 + \delta_0/2}\right)^{L/2-1} \leq \frac{v_1^{L-2}}{1 + \delta_0/2}.
$$

---

[5]The only difference is that now the $L$-th order terms cannot be simply ignored as we no longer have the induction hypothesis $v_p^2 \leq \log^2 d/P$. To handle them, it suffices to note that if $v_1^2 \geq v_q^2$, then those $L$-th order terms of $v_1^2$ are also larger, which will even lead to an amplification of the gap. In fact, this is why we can recover the directions using them.

Therefore,

$$v_1^{l-2} - \|\boldsymbol{v}\|_l^l \geq v_1^{l-2}(1 - v_1^2) - \left(\|\boldsymbol{v}_{\leq P}\|^2 - v_1^2\right)\frac{v_1^{L-2}}{1 + \delta_0/2}$$

$$= \frac{v_1^{l-2}}{1 + \delta_0/2}\left(1 + \delta_0(1 - v_1^2) - \|\boldsymbol{v}_{\leq P}\|^2\right) \geq \frac{\delta_0}{2}v_1^{l-2}\left(1 - v_1^2\right).$$

As a result, for the signal term, we have

$$v_1^2\left(1 + 2\eta\sum_{l\geq L}l\hat{\phi}_l^2 v_1^{l-2} - 2\eta\sum_{l\geq L}l\hat{\phi}_l^2\|\boldsymbol{v}_{\leq P}\|_l^l\right) \geq v_1^2 + L\hat{\phi}_L^2\delta_0\left(1 - v_1^2\right)\eta v_1^L.$$

In particular, when $v_1^2 \leq 3/4$, we have

$$v_{t+1,1}^2 \geq v_1^2 + \frac{L\hat{\phi}_L^2}{4}\delta_0\eta v_1^L + H_{t+1} + \xi_{t+1}.$$

Thus, using the notations of Lemma E.5, we have

$$\alpha = \frac{L\hat{\phi}_L^2}{4}\delta_0\eta, \quad \Xi \lesssim_\phi \eta^2 d\hat{M}_z^2, \quad \sigma_Z^2 \lesssim_\phi \eta^2 M_Z^2, \quad p = L/2, \quad x_0 = \Omega(1/P).$$

To meet the conditions of Lemma E.5, it suffices to choose

$$\alpha \lesssim x_0^{p-1} \quad \Leftarrow \quad \alpha \lesssim \delta_0^{-1}x_0^{p-1},$$

$$\Xi \lesssim \alpha x_0^{p-1} \quad \Leftarrow \quad \eta \lesssim_\phi \frac{\delta_0}{d\hat{M}_z^2 P^{L/2-1}},$$

$$\sigma_Z^2 \lesssim_\theta \frac{\alpha x_0^p}{\log^{\theta+1}\left(\log(1/x_0)/(\alpha x_0^{p-1}\delta_\mathbb{P})\right)} \quad \Leftarrow \quad \eta \lesssim_\phi \frac{\delta_0}{M_Z^2 P^{L/2}}\frac{1}{\log^{\theta+1}(d/\delta_\mathbb{P})}.$$

Combine the above and we get the condition

$$\eta \lesssim_\phi \frac{\delta_0}{dP^{L/2}}\left(\frac{P}{\hat{M}_z^2} \wedge \frac{d}{M_Z^2}\frac{1}{\log^{\theta+1}(d/\delta_\mathbb{P})}\right).$$

Finally, we apply Lemma E.5 to complete the proof. $\square$

**Lemma B.11** (Strong recovery of directions). *Let $\boldsymbol{v} \in \mathbb{S}^{d-1}$ be an arbitrary first-layer neuron. Let $\delta_\mathbb{P}$ and $\varepsilon_*$ be given. Suppose that we choose*

$$\eta \lesssim_\phi \frac{\varepsilon_*}{dP}\left(\frac{P}{\hat{M}_Z^2\log(1/\varepsilon_*)} \wedge \frac{\varepsilon_* dP}{M_Z^2\log d\log^{\theta+1}(d/\delta_\mathbb{P})}\right).$$

*Then, with probability at least $1 - O(\delta_\mathbb{P})$, we have $v_1^2 \geq 1 - \varepsilon_*$ within $O_\phi(\log(1/\varepsilon_*)/\eta)$ iterations.*

*Proof.* Again, By Lemma B.9, we have

$$v_{t+1,1}^2 \geq v_{t,1}^2\left(1 + 2\eta\sum_{l\geq L}l\hat{\phi}_l^2 v_1^{l-2} - 2\eta\sum_{l\geq L}l\hat{\phi}_l^2\|\boldsymbol{v}_{t,\leq P}\|_l^l\right) + H_{t+1} + \tilde{\xi}_{t+1},$$

where $H_{t+1}$ is a martingale difference term that is conditionally $(O_\phi(\eta^2 M_Z^2 v_{t,1}^2), \theta)$-subweibull, and $\xi_{t+1}$ is bounded by $O_\phi(\eta^2 d\hat{M}_Z^2 v_{t,1}^2)$ for all $t \in [T]$ with probability at least $1 - \delta_\mathbb{P}$. Meanwhile, by the proof of the previous lemma, we have

$$v_1^2\left(1 + 2\eta\sum_{l\geq L}l\hat{\phi}_l^2 v_1^{l-2} - 2\eta\sum_{l\geq L}l\hat{\phi}_l^2\|\boldsymbol{v}_{\leq P}\|_l^l\right) \geq v_1^2 + 2L\hat{\phi}_L^2\frac{c_{g,L}}{1 + c_{g,L}}\left(1 - v_1^2\right)\eta v_1^L$$

$$\geq v_1^2 + \eta\hat{\phi}_L^2\frac{2Lc_{g,L}}{1 + c_{g,L}}\left(\frac{3}{4}\right)^L\left(1 - v_1^2\right)$$

$$=: v_1^2 + \eta c_{g,\phi}\left(1 - v_1^2\right),$$

for some constant $c_{g,\phi} > 0$ that depends only on $c_{g,L}$ and $\phi$. Thus,
$$1 - v_{t+1,1}^2 \geq \left(1 - v_1^2\right) - \eta c_{g,L,\phi}\left(1 - v_1^2\right) - H_{t+1} - \xi_{t+1}.$$
In the language of Lemma 4.1,[6] we have
$$\alpha = -\eta c_{g,L,\phi}, \quad \eta T \lesssim_\phi \log(1/\varepsilon_*), \quad \sigma_Z^2 \lesssim_\phi \eta^2 M_Z^2, \quad \Xi \lesssim_\phi \eta^2 d\hat{M}_Z^2.$$
To meet the conditions of Lemma 4.1, it suffices to choose
$$\Xi \lesssim \frac{\varepsilon_*}{T} \quad \Leftarrow \quad \eta \lesssim_\phi \frac{\varepsilon_*}{d\hat{M}_Z^2 \log(1/\varepsilon_*)},$$
$$\sigma_Z^2 \lesssim \frac{x_0^2}{T \log^{\theta+1}(T/\delta_\mathbb{P})} \quad \Leftarrow \quad \eta \lesssim \frac{\varepsilon_*^2}{M_Z^2 \log d \log^{\theta+1}(d/\delta_\mathbb{P})}.$$
Under the above conditions, by Lemma 4.1, we have $v_1^2 \geq 1 - \varepsilon_*$ within $T = O_\phi(\log(1/\varepsilon_*)/\eta)$ iterations with probability at least $1 - O(\delta_\mathbb{P})$. $\qquad\square$

## B.3 Deferred proofs in this section

*Proof of Lemma B.1.* First, recall from (11) that
$$\hat{v}_{t+1,k} = v_{t,k} + \eta\mathbb{1}\{k \leq P\}\left(2\hat{\phi}_2^2 + L\hat{\phi}_L^2 v_k^{L-2} + \sum_{l>L} l\hat{\phi}_l^2 v_k^{l-2}\right)v_k - \eta\rho v_k + \eta Z_{t+1,k}.$$
Therefore,
$$\hat{v}_{t+1,k}^2 = v_{t,k}^2 + 2\eta v_{t,k}\left(\mathbb{1}\{k \leq P\}\left(2\hat{\phi}_2^2 + L\hat{\phi}_L^2 v_k^{L-2} + \sum_{l>L} l\hat{\phi}_l^2 v_k^{l-2}\right)v_k - \rho v_k + Z_{t+1,k}\right)$$
$$+ \eta^2\left(\mathbb{1}\{k \leq P\}\left(2\hat{\phi}_2^2 + L\hat{\phi}_L^2 v_k^{L-2} + \sum_{l>L} l\hat{\phi}_l^2 v_k^{l-2}\right)v_k - \rho v_k + Z_{t+1,k}\right)^2$$
$$=: v_{t,k}^2 + \mathtt{T}_1\left(\hat{v}_{t+1,k}^2\right) + \mathtt{T}_2\left(\hat{v}_{t+1,k}^2\right).$$
For the first term, we rewrite it as
$$\mathtt{T}_1 = 2\eta v_{t,k}^2\left(\mathbb{1}\{k \leq P\}\left(2\hat{\phi}_2^2 + L\hat{\phi}_L^2 v_k^{L-2} + \sum_{l>L} l\hat{\phi}_l^2 v_k^{l-2}\right) - \rho\right) + 2\eta v_{t,k} Z_{t+1,k}.$$
Consider the second term. For notation simplicity, put
$$\gamma_{k,t} := \mathbb{1}\{k \leq P\}\left(2\hat{\phi}_2^2 + L\hat{\phi}_L^2 v_k^{L-2} + \sum_{l>L} l\hat{\phi}_l^2 v_k^{l-2}\right) - \rho.$$
Note that $\gamma_{k,t}$ is $\mathcal{F}_t$-measurable and by Assumption 1, we have
$$2\hat{\phi}_2^2 + L\hat{\phi}_L^2 v_k^{L-2} + \sum_{l>L} l\hat{\phi}_l^2 v_k^{l-2} \leq 2\hat{\phi}_2^2 + L\hat{\phi}_L^2 + \sum_{l>L} l\hat{\phi}_l^2 \leq C_\phi^2,$$
$$\rho := 2\hat{\phi}_2^2 \|\boldsymbol{v}_{\leq P}\|^2 + L\hat{\phi}_L^2 \|\boldsymbol{v}_{\leq P}\|_L^L + \sum_{l>L} l\hat{\phi}_l^2 \|\boldsymbol{v}_{\leq P}\|_l^l \leq 2\hat{\phi}_2^2 + L\hat{\phi}_L^2 + \sum_{l>L} l\hat{\phi}_l^2 \leq C_\phi^2,$$
and therefore $|\zeta_{k,t}| \leq 2C_\phi^2$. Then, we compute
$$\frac{\mathtt{T}_2}{\eta^2} = (\gamma_{t,k} v_k + Z_{t+1,k})^2 = \gamma_{t,k}^2 v_{t,k}^2 + Z_{t+1,k}^2 + 2\gamma_{t,k} v_{t,k} Z_{t+1,k}.$$
Combine the above two bounds and we get
$$\hat{v}_{t+1,k}^2 = v_{t,k}^2 + 2\eta\gamma_{t,k} v_{t,k}^2 + 2\eta v_{t,k} Z_{t+1,k} + \eta^2\gamma_{t,k}^2 v_{t,k}^2 + \eta^2 Z_{t+1,k}^2 + 2\eta^2\gamma_{t,k} v_{t,k} Z_{t+1,k}.$$
Now, consider the last three terms. By (12), we have $|Z_{t+1,k}| \lesssim_\phi \hat{M}_Z$ with probability at least $1 - \delta_\mathbb{P}$ for all $t \in [T]$. Thus,
$$\left|\eta^2\gamma_{t,k}^2 v_{t,k}^2 + \eta^2 Z_{t+1,k}^2 + 2\eta^2\gamma_{t,k} v_{t,k} Z_{t+1,k}\right| \lesssim_\phi \eta^2\hat{M}_Z^2.$$
$\qquad\square$

---

[6]When $\alpha$ is negative, it suffices to replace $x_0$ with our target $\varepsilon_*$.

# C  Stage 2: training the second layer

**Lemma C.1.** *Suppose that for each $p \in [P]$, there exists a first-layer neuron $\boldsymbol{v}_{i_p}$ with $v_{i_p,p} \geq \sqrt{1 - \varepsilon_v}$ for some small positive $\varepsilon_v = O(1/P)$, then we can choose $\boldsymbol{a}_* \in \mathbb{R}^m$ with $\|\boldsymbol{a}_*\| = \sqrt{P}$ such that*

$$\mathcal{L}(\boldsymbol{a}_*, \boldsymbol{V}) := \mathbb{E}\left(f_*(\boldsymbol{x}) - f(\boldsymbol{x}; \boldsymbol{a}_*, \boldsymbol{V})\right)^2 \leq 20 C_\phi^2 P^2 \varepsilon_v.$$

*Proof.* Choose one $\boldsymbol{v}_{i_p}$ for each $p \in [P]$. Then, we set the $i_p$-th entries of $\boldsymbol{a}_*$ to be 1 and all other entries 0. Then, we write

$$\left(f_*(\boldsymbol{x}) - f(\boldsymbol{x}; \boldsymbol{a}_*, \boldsymbol{V})\right)^2 = \left(\sum_{k=1}^P \left(\phi(x_k) - \phi(\boldsymbol{v}_{i_k} \cdot \boldsymbol{x})\right)\right)^2$$

$$= \sum_{k,l=1}^P \left(\phi(x_k) - \phi(\boldsymbol{v}_{i_k} \cdot \boldsymbol{x})\right)\left(\phi(x_l) - \phi(\boldsymbol{v}_{i_l} \cdot \boldsymbol{x})\right).$$

By expanding $\phi$ in the Hermite basis, for any $\boldsymbol{v}, \boldsymbol{v}' \in \mathbb{S}^{d-1}$, we have

$$\mathbb{E}_{\boldsymbol{x} \sim \mathcal{N}(0, \boldsymbol{I})}[\phi(\boldsymbol{v} \cdot \boldsymbol{x})\phi(\boldsymbol{v}' \cdot \boldsymbol{x})] = \sum_{i=0}^\infty \hat{\phi}_i^2 \langle \boldsymbol{v}, \boldsymbol{v}' \rangle^i = \sum_{i=2}^\infty \hat{\phi}_i^2 \langle \boldsymbol{v}, \boldsymbol{v}' \rangle^i.$$

Hence, for $k = l$, we have

$$\mathbb{E}\left(\phi(x_k) - \phi(\boldsymbol{v}_{i_k} \cdot \boldsymbol{x})\right)^2 = \mathbb{E}\,\phi^2(x_k) + \mathbb{E}\,\phi^2(\boldsymbol{v}_{i_k} \cdot \boldsymbol{x}) - 2\,\mathbb{E}\,\phi(x_k)\phi(\boldsymbol{v}_{i_k} \cdot \boldsymbol{x})$$

$$= 2\sum_{i=2}^\infty \hat{\phi}_i^2 \left(1 - \langle \boldsymbol{e}_k, \boldsymbol{v}_{i_k} \rangle^i\right)$$

$$\leq 2 C_\phi^2 \varepsilon_v.$$

Meanwhile, for $k \neq l$, we have

$$\mathbb{E}\left(\phi(x_k) - \phi(\boldsymbol{v}_{i_k} \cdot \boldsymbol{x})\right)\left(\phi(x_l) - \phi(\boldsymbol{v}_{i_l} \cdot \boldsymbol{x})\right)$$
$$= \mathbb{E}\,\phi(x_k)\phi(x_l) + \mathbb{E}\,\phi(\boldsymbol{v}_{i_k} \cdot \boldsymbol{x})\phi(\boldsymbol{v}_{i_l} \cdot \boldsymbol{x}) - \mathbb{E}\,\phi(x_k)\phi(\boldsymbol{v}_{i_l} \cdot \boldsymbol{x}) - \mathbb{E}\,\phi(\boldsymbol{v}_{i_k} \cdot \boldsymbol{x})\phi(x_l)$$
$$= \sum_{i=2}^\infty \hat{\phi}_i^2 \left(\langle \boldsymbol{v}_{i_k}, \boldsymbol{v}_{i_l} \rangle^i - v_{i_l,k}^i - v_{i_k,l}^i\right).$$

Note that $v_{i_l,k}^2 \vee v_{i_k,l}^2 \leq \varepsilon_v$ and

$$\langle \boldsymbol{v}_{i_k}, \boldsymbol{v}_{i_l} \rangle^2 \leq 2v_{i_l,k}^2 + 2\langle \boldsymbol{v}_{i_k} - \boldsymbol{e}_k, \boldsymbol{v}_{i_l} \rangle^2 \leq 2\varepsilon_v + 2\|\boldsymbol{v}_{i_k} - \boldsymbol{e}_k\|^2 = 2\varepsilon_v + 4(1 - v_{i_k,k}) \leq 6\varepsilon_v.$$

As a result,

$$\mathbb{E}\left(\phi(x_k) - \phi(\boldsymbol{v}_{i_k} \cdot \boldsymbol{x})\right)\left(\phi(x_l) - \phi(\boldsymbol{v}_{i_l} \cdot \boldsymbol{x})\right) \leq 10 C_\phi^2 \varepsilon_v.$$

Combining these two cases, we obtain

$$\mathcal{L} = \mathbb{E}\left(f_*(\boldsymbol{x}) - f(\boldsymbol{x}; \boldsymbol{a}_*, \boldsymbol{V})\right)^2 \leq 20 C_\phi^2 P^2 \varepsilon_v.$$

$\square$

Now, we are ready to prove the following generalization bound for Stage 2. The proof of it is adapted from Section B.8 of [OSSW24], which in turn is based on ([DLS22, AAM22, BES$^+$22]).

**Lemma C.2.** *Suppose that for each $p \in [P]$, there exists a first-layer neuron $\boldsymbol{v}_{i_p}$ with $v_{i_p,p}^2 \geq 1 - \varepsilon_v$ for some small positive $\varepsilon_v = O(1/P)$. Then, there exists some $\lambda > 0$ such that the ridge estimator $\hat{\boldsymbol{a}}$ we obtain in Stage 2 satisfies*

$$\|f(\cdot; \hat{\boldsymbol{a}}, \boldsymbol{V}) - f_*\|_{L^1(D)} \leq \frac{8\|\boldsymbol{a}_*\|\sqrt{m}}{\sqrt{N}\delta_{\mathbb{P}}} + \sqrt{10LP^2\varepsilon_v},$$

*with probability at least $1 - 2\delta_{\mathbb{P}}$.*

*Proof.* For notational simplicity, let $D = \mathcal{N}(0, 1)$ and $\hat{D} = \frac{1}{N} \sum_{n=1}^{N} \delta_{\boldsymbol{x}_{T+n}}$ denote the empirical distribution of the samples we use in Stage 2. In addition, we write $f_{\boldsymbol{a}}$ for $f(\cdot; \boldsymbol{a}, \boldsymbol{V})$ where $\boldsymbol{V}$ is the first-layer weights we have obtained in Stage 1 and $\boldsymbol{X} = (\boldsymbol{x}_{T+n})_{n=1}^{N}$.

Let $\boldsymbol{a}_* \in \mathbb{R}^m$ denote the second-layer weights we constructed in Lemma C.1 and $\hat{\boldsymbol{a}} \in \mathbb{R}^m$ denote the ridge estimator obtained via minimizing $\boldsymbol{a} \mapsto \|f_* - f_{\boldsymbol{a}}\|_{L^2(\hat{D})}^2 + \lambda \|\boldsymbol{a}\|^2$. By the equivalence between norm-constrained linear regression and ridge regression, there exists $\lambda > 0$ such that

$$\|f_* - f_{\hat{\boldsymbol{a}}}\|_{L^2(\hat{D})}^2 \le \|f_* - f_{\boldsymbol{a}_*}\|_{L^2(\hat{D})}^2 \quad \text{and} \quad \|\hat{\boldsymbol{a}}\| \le \|\boldsymbol{a}_*\|.$$

Choose this $\lambda$ and let $\mathcal{F} := \{f(\cdot; \boldsymbol{a}) : \|\boldsymbol{a}\| \le \|\boldsymbol{a}_*\|\}$ be our hypothesis class. Note that $f_{\hat{\boldsymbol{a}}} \in \mathcal{F}$. Moreover, we have

$$\begin{aligned}
\|f_{\hat{\boldsymbol{a}}} - f_*\|_{L^1(D)} &= \left( \|f_{\hat{\boldsymbol{a}}} - f_*\|_{L^1(D)} - \|f_{\hat{\boldsymbol{a}}} - f_*\|_{L^1(\hat{D})} \right) + \|f_{\hat{\boldsymbol{a}}} - f_*\|_{L^1(\hat{D})} \\
&\le \sup_{\boldsymbol{a} : \|\boldsymbol{a}\| \le \|\boldsymbol{a}_*\|} \left( \|f_{\boldsymbol{a}} - f_*\|_{L^1(D)} - \|f_{\boldsymbol{a}} - f_*\|_{L^1(\hat{D})} \right) + \|f_{\hat{\boldsymbol{a}}} - f_*\|_{L^1(\hat{D})} \\
&\le \sup_{\boldsymbol{a} : \|\boldsymbol{a}\| \le \|\boldsymbol{a}_*\|} \left( \|f_{\boldsymbol{a}} - f_*\|_{L^1(D)} - \|f_{\boldsymbol{a}} - f_*\|_{L^1(\hat{D})} \right) + \|f_{\boldsymbol{a}_*} - f_*\|_{L^2(\hat{D})},
\end{aligned}$$

where we used the fact that $\|f_{\hat{\boldsymbol{a}}} - f_*\|_{L^1(\hat{D})} \le \|f_{\hat{\boldsymbol{a}}} - f_*\|_{L^2(\hat{D})} \le \|f_{\boldsymbol{a}_*} - f_*\|_{L^1(\hat{D})}$ in the last line.

Now, we bound the first term. Let $\boldsymbol{\sigma} := (\sigma_n)_{n=1}^{N}$ be i.i.d. Rademacher variables that are also independent of everything else. By symmetrization and Theorem 7 of [MZ03], we have

$$\mathbb{E}_{\boldsymbol{X}} \left[ \sup_{\boldsymbol{a} : \|\boldsymbol{a}\| \le \|\boldsymbol{a}_*\|} \left( \|f_{\boldsymbol{a}} - f_*\|_{L^1(D)} - \|f_{\boldsymbol{a}} - f_*\|_{L^1(\hat{D})} \right) \right]$$

$$\le 2 \mathbb{E}_{\boldsymbol{X}, \boldsymbol{\sigma}} \sup_{\boldsymbol{a} : \|\boldsymbol{a}\| \le \|\boldsymbol{a}_*\|} \frac{1}{N} \sum_{t=1}^{N} \sigma_t |f_a(\boldsymbol{x}_{T+n}) - f_*(\boldsymbol{x}_{T+n})|$$

$$\le 2 \mathbb{E}_{\boldsymbol{X}, \boldsymbol{\sigma}} \sup_{\boldsymbol{a} : \|\boldsymbol{a}\| \le \|\boldsymbol{a}_*\|} \frac{1}{N} \sum_{t=1}^{N} \sigma_t \left( f_a(\boldsymbol{x}_{T+n}) - f_*(\boldsymbol{x}_{T+n}) \right)$$

$$\le \frac{2}{N} \mathbb{E}_{\boldsymbol{X}, \boldsymbol{\sigma}} \sup_{\boldsymbol{a} : \|\boldsymbol{a}\| \le \|\boldsymbol{a}_*\|} \sum_{t=1}^{N} \sigma_t f_a(\boldsymbol{x}_{T+n}) + 2 \underbrace{\mathbb{E}_{\boldsymbol{X}, \boldsymbol{\sigma}} \frac{1}{N} \sum_{t=1}^{N} \sigma_t f_*(\boldsymbol{x}_{T+n})}_{0}.$$

Note that the first term is two times the Rademacher complexity $\operatorname{Rad}_N(\mathcal{F})$ of $\mathcal{F}$ (see, for example, Chapter 4 of [Wai19]). By (the proof of) Lemma 48 of [DLS22], we have

$$\begin{aligned}
\operatorname{Rad}_N(\mathcal{F}) &\le \frac{\|\boldsymbol{a}_*\|}{\sqrt{N}} \sqrt{\mathbb{E}_{\boldsymbol{x} \sim \mathcal{N}(0, \boldsymbol{I}_d)} \|\phi(\boldsymbol{V}\boldsymbol{x})\|^2} = \frac{\|\boldsymbol{a}_*\|}{\sqrt{N}} \sqrt{\sum_{k=1}^{m} \mathbb{E}_{\boldsymbol{x} \sim \mathcal{N}(0, \boldsymbol{I}_d)} \phi^2(\boldsymbol{v}_k \cdot \boldsymbol{x})} \\
&= \frac{\|\boldsymbol{a}_*\| \sqrt{m}}{\sqrt{N}} \sqrt{\mathbb{E}_{x_1 \sim \mathcal{N}(0, 1)} \phi^2(x_1)} \\
&= \frac{2 \|\boldsymbol{a}_*\| \sqrt{m}}{\sqrt{N}}.
\end{aligned}$$

In other words, we have

$$\mathbb{E} \sup_{\boldsymbol{a} : \|\boldsymbol{a}\| \le \|\boldsymbol{a}_*\|} \left( \|f_{\boldsymbol{a}} - f_*\|_{L^1(D)} - \|f_{\boldsymbol{a}} - f_*\|_{L^1(\hat{D})} \right) \le \frac{4 \|\boldsymbol{a}_*\| \sqrt{m}}{\sqrt{N}}.$$

Hence, for any $\delta_{\mathbb{P}} \in (0, 1)$, by Markov's inequality, we have

$$\sup_{\boldsymbol{a} : \|\boldsymbol{a}\| \le \|\boldsymbol{a}_*\|} \left( \|f_{\boldsymbol{a}} - f_*\|_{L^1(D)} - \|f_{\boldsymbol{a}} - f_*\|_{L^1(\hat{D})} \right) \le \frac{4 \|\boldsymbol{a}_*\| \sqrt{m}}{\sqrt{N} \delta_{\mathbb{P}}},$$

with probability at least $1 - \delta_{\mathbb{P}}$. Apply the same argument to $\|f_{\boldsymbol{a}_*} - f_*\|_{L^2(\hat{D})}$ and recall from Lemma C.1 that $\|f_{\boldsymbol{a}_*} - f_*\|_{L^2(D)}^2 \le 10LP^2 \varepsilon_v$, and we obtain

$$\|f_{\hat{\boldsymbol{a}}} - f_*\|_{L^1(D)} \le \frac{8 \|\boldsymbol{a}_*\| \sqrt{m}}{\sqrt{N} \delta_{\mathbb{P}}} + \sqrt{10LP^2 \varepsilon_v},$$

with probability at least $1 - 2\delta_{\mathbb{P}}$. $\qquad \square$

## D  Proof of the main theorem

**Theorem 2.1** (Main Theorem). *Consider the setting and algorithm described above. Let $C > 0$ be a large universal constant. Suppose that $\log^C d \le P \le d$ and $\{\boldsymbol{v}_k^*\}_{k \in [P]}$ are orthonormal. Let $\delta_{\mathbb{P}} \in (\exp(-\log^C d), 1)$ and $\varepsilon_* > 0$ be given. Suppose that we choose $a_0, \eta, T, N$ satisfying*

$$m = \tilde{\Theta}(P), \quad N = \tilde{\Theta}\left(\frac{P^2}{\varepsilon_*^2 \delta_{\mathbb{P}}^2}\right), \quad \eta = \tilde{\Theta}_\phi\left(\frac{\varepsilon_*^2 \delta_{\mathbb{P}}}{PdP^{L/2-1}}\right), \quad T = \tilde{O}_\phi\left(\frac{P^{L/2-1}}{\eta \varepsilon_*^4 \delta_{\mathbb{P}}}\right).$$

*Then, there exists some $\lambda > 0$ such that at the end of training, we have $\mathcal{L}_{\mathrm{MSE}}(\boldsymbol{a}, \boldsymbol{V}) \le \varepsilon_*$ with probability at least $1 - O(\delta_{\mathbb{P}})$.*

*Proof.* First, by Lemma A.8, we should choose $m = \Theta\left(P\log(P/\delta_{\mathbb{P}})\right)$ and the $\delta_0$ in Lemma B.2 and Lemma B.8 can be chosen to be $\Theta(1/\log P)$. Meanwhile, by Lemma C.2, to achieve target $L^1$-error $\varepsilon_*$ with probability at least $1 - O(\delta_{\mathbb{P}})$, we need

$$N \gtrsim \frac{Pm}{\varepsilon_*^2 \delta_{\mathbb{P}}^2} = \Theta\left(\frac{P^2 \log(P/\delta_{\mathbb{P}})}{\varepsilon_*^2 \delta_{\mathbb{P}}^2}\right), \quad \varepsilon_v = O_\phi\left(\frac{\varepsilon_*^2}{P^2}\right).$$

By Lemma 2.1, we have $M_Z \lesssim_\phi P^{1/2}$ and $\hat{M}_Z \lesssim_\phi P^{1/2} \log^\theta \log(P/\delta_{\mathbb{P}})$ where $\theta = 1/(2(1+q))$. Then, to meet the conditions of Lemma B.2 and Lemma B.8 (uniformly over those $P$ good neurons), it suffices to choose

$$\eta \lesssim_\phi \frac{1}{\log^{2\theta+3}(d/\delta_{\mathbb{P}})}\left(\frac{1}{dP^{L/2}} \wedge \frac{\varepsilon_*^2}{P\log(1/\varepsilon_*)}\right)$$

Then, by Lemma B.2 and Lemma B.8, the numbers of iterations needed for Stage 1.1 and Stage 1.2 are $O_\phi(\log(d/P)/\eta)$ and $O_\phi\left(\left(P^{L/2-1} + \log(1/\varepsilon_v)\right)/\eta\right)$, respectively. Thus, the total number of iterations is bounded by

$$T = O_\phi\left(\frac{\log d + P^{L/2-1} + \log(P/\varepsilon_*)}{\eta}\right) = \tilde{O}_\phi\left(dP^{L-1} \vee \frac{P^{L/2}\log(1/\varepsilon_*)}{\varepsilon_*^2}\right).$$

$\qquad \square$

## E  Stochastic Induction

Our proof is essentially a large induction: When certain properties hold, we know how to analyze the dynamics and can show certain quantities are bounded with high probability. Meanwhile, certain properties hold as long as those quantities are still well-controlled. In the deterministic setting, this seemingly looped argument can be made formal by either mathematical induction (in discrete time) or the continuity argument (in continuous time). In this subsection, we show the same can also be done in the presence of randomness and derive a stochastic version of Gronwall's lemma and its generalizations.

We start with an example where Doob's submartingale inequality can be directly used. Let $(\Omega, \mathcal{F}, (\mathcal{F}_t)_t, \mathbb{P})$ be our filtered probability space and $(Z_t)_t$ be a martingale difference sequence. Suppose that $\mathbb{E}[Z_{t+1}^2 \mid \mathcal{F}_t]$ is uniformly bounded by $\sigma_Z^2$. Then, by Doob's submartingale inequality, for any $M > 0$ and $T > 0$, we have

$$\mathbb{P}\left[\sup_{t \le T}\left|\sum_{s=1}^{t} Z_s\right| \ge M\right] \le M^{-2}\,\mathbb{E}\left(\sum_{s=1}^{T} Z_s\right)^2 = \frac{T\sigma_Z^2}{M^2}.$$

In particular, this implies that when $M = \omega(\sigma_Z\sqrt{T})$, we have $\sup_{t \le T}\left|\sum_{s=1}^{t} Z_s\right| \le M$ with high probability.

Note that there is no need to do any kind of "induction" in the above example because of the unconditional uniform bound on $\mathbb{E}[Z_{t+1}^2 \mid \mathcal{F}_t]$. However, things become subtle if instead of assuming

$\mathbb{E}[Z_{t+1}^2 \mid \mathcal{F}_t]$ is always bounded by $\sigma_Z^2$, we assume it to be bounded by $\sigma_Z^2$ when $\sup_{s \leq t} |\sum_{r=1}^s Z_r| \leq M$. Intuitively, since $M$ is chosen so that $\sup_{t \leq T} \left| \sum_{s=1}^t Z_s \right| \leq M$ holds with high probability, the bounds $\mathbb{E}[Z_{t+1}^2 \mid \mathcal{F}_t] \leq \sigma_Z^2$ should also hold with high probability and we can still use Doob's submartingale inequality as before. Now, we formalize this argument.

**Lemma E.1.** *Let $(Z_t)_t$ be a martingale difference sequence. Suppose that there exists $M, \sigma_Z > 0$ such that if $\sup_{s \leq t} |\sum_{r=1}^s Z_s| \leq M$, then we have $\mathbb{E}[Z_{t+1}^2 \mid \mathcal{F}_t] \leq \sigma_Z^2$. Then, we have*

$$\mathbb{P}\left[ \sup_{t \leq T} \left| \sum_{s=1}^t Z_s \right| > M \right] \leq \frac{T \sigma_Z^2}{M^2}.$$

*Note that this bound is the same as the one we obtained with the assumption that $\mathbb{E}[Z_{t+1}^2 \mid \mathcal{F}_t] \leq \sigma_Z^2$ always holds.*

*Proof.* Consider the stopping time $\tau := \inf\{t \geq 0 \; : \; \left| \sum_{s=1}^t Z_s \right| > M\}$. By definition, we have $\sup_{s \leq t} |\sum_{r=1}^s Z_s| \leq M$ for all $t \leq \tau$. Then, we define $Y_{t+1} = Z_{t+1}\mathbb{1}\{t < \tau\}$. Note that $(Y_t)$ is a martingale difference sequence with $\mathbb{E}[Y_{t+1}^2 \mid \mathcal{F}_t] \leq \sigma_Z^2$. As a result, by Doob's submartingale inequality, we have $\mathbb{P}\left[ \sup_{t \leq T} \left| \sum_{s=1}^t Y_s \right| > M \right] \leq T\sigma_Z^2/M^2$. To relate it to $(Z_t)_t$, we compute

$$
\mathbb{P}\left[ \sup_{t \leq T} \left| \sum_{s=1}^t Z_s \right| > M \right] = \mathbb{P}\left[ \sup_{t \leq T} \left| \sum_{s=1}^t Z_s \right| > M \wedge \tau \leq T \right] = \mathbb{P}\left[ \left| \sum_{s=1}^\tau Z_s \right| > M \wedge \tau \leq T \right]
$$

$$
= \mathbb{P}\left[ \left| \sum_{s=1}^\tau Y_s \right| > M \wedge \tau \leq T \right]
$$

$$
\leq \frac{T\sigma_Z^2}{M^2},
$$

where the first and second identities comes from the definition of $\tau$ and the third from the fact $Z_t = Y_t$ for all $t \leq \tau$. $\qquad\square$

Now, we consider a more complicated case, where the process of interest is not a pure martingale. Suppose that the process $(X_t)_t$ satisfies

$$X_{t+1} = (1+\alpha)X_t + \xi_{t+1} + Z_{t+1}, \quad X_0 = x_0 > 0,$$

where the signal growth rate $\alpha > 0$ and initialization $x_0 > 0$ are given and fixed, $(\xi_t)_t$ is an adapted process, and $(Z_t)_t$ is a martingale difference sequence. In most cases, $(\xi_t)_t$ will represent the higher-order error terms.

Our goal is control the difference between $X_t$ and its deterministic counterpart $x_t = (1+\alpha)^t x_0$. To this end, we recursively expand the RHS to obtain

$$X_{t+1} = (1+\alpha)^2 X_{t-1} + (1+\alpha)\xi_t + \xi_{t+1} + (1+\alpha)Z_t + Z_{t+1}$$

$$= (1+\alpha)^{t+1} x_0 + \sum_{s=1}^t (1+\alpha)^{t-s}\xi_{s+1} + \sum_{s=1}^t (1+\alpha)^{t-s} Z_{s+1}.$$

Divide both sides with $(1+\alpha)^{t+1}$ and replace $t+1$ with $t$. Then, the above becomes

$$X_t(1+\alpha)^{-t} = x_0 + \sum_{s=1}^t (1+\alpha)^{-s}\xi_s + \sum_{s=1}^t (1+\alpha)^{-s} Z_s.$$

Note that $((1+\alpha)^{-t} Z_t)_t$ is still a martingale difference sequence. Ideally, $|\xi_t|$ should be small as it represents the higher-order error terms, and we have bounds on the conditional variance of $Z_t$ so that we can apply Doob's submartingale inequality to the last term. Unfortunately, in many cases, since $\xi_{t+1}$ and $Z_{t+1}$, particularly their maximum and (conditional) variance, can potentially depend on $(X_s)_{s \leq t}$, we may only be able to assume $|\xi_{t+1}| \leq (1+\alpha)^t \Xi$ with probability at least $1 - \delta_{\mathbb{P},\xi}$ (for each $t$) and $\mathbb{E}[Z_{t+1}^2 \mid \mathcal{F}_t] \leq (1+\alpha)^t \sigma_Z^2$ for some $\xi_{\mathbb{P},\xi}, \Xi$ and $\sigma_Z^2$ when, say, $X_t = (1 \pm 0.5)x_t$. Still,

we can use the previous argument to estimate the probability that $X_t \notin (1 \pm 0.5)x_t$ for some $t \leq T$. We now formalize this argument. In addition, instead of Doob's $L^2$ submartingale inequality, we will use the following extension of Freedman's inequality, which allows us to improve the dependence on failure probability from linear to poly-logarithmic. The proof of this lemma is deferred to the end of this section.

**Lemma E.2** (Freedman's inequality with subweibull variables). *Let $\{Z_t\}_t$ be a martingale difference sequence that is conditionally $(\sigma^2, \theta)$-subweibull, i.e.,*

$$\mathbb{P}\left[|Z_t| \geq M \mid \mathcal{F}_{t-1}\right] \leq C \exp\left(-(M/\sigma)^{1/\theta}\right), \quad \forall M \geq 0,$$

*for some universal constant $C > 0$. Then, for any $\delta_{\mathbb{P}} \in (0, 1)$, we have*

$$\left|\sum_{t=1}^{T} Z_t\right| \lesssim_\theta \sigma\sqrt{T \log^{\theta+1}(T/\delta_{\mathbb{P}})}, \quad \text{with probability at least } 1 - \delta_{\mathbb{P}}.$$

**Lemma 4.1** (Stochastic Gronwall's lemma). *Suppose that $(X_t)_t$ satisfies*

$$X_{t+1} = (1 + \alpha)X_t + \xi_{t+1} + Z_{t+1}, \quad X_0 = x_0 > 0, \tag{5}$$

*where the signal growth rate $\alpha > 0$ and initialization $x_0 > 0$ are given, $(\xi_t)_t$ is an adapted process, and $(Z_t)_t$ is a martingale difference sequence. Define $x_t = (1 + \alpha)^t x_0$.*

*Let $T > 0$ and $\delta_{\mathbb{P}} \in (0, 1)$ be given. Suppose that there exists some $\delta_{\mathbb{P},\xi} \in (0, 1)$ and $\Xi, \sigma_Z > 0$ such that for every $t \geq 0$, if $X_t = (1 \pm 0.5)x_t$, then we have $|\xi_{t+1}| \leq (1 + \alpha)^t \Xi$ with probability at least $1 - \delta_{\mathbb{P},\xi}$ and $Z_{t+1}$ is conditionally $((1 + \alpha)^t \sigma_Z^2, \theta)$-subweibull. Then, if*

$$\Xi \lesssim \frac{x_0}{T} \quad \text{and} \quad \sigma_Z^2 \lesssim \frac{x_0^2}{T \log^{\theta+1}(T/\delta_{\mathbb{P}})}, \tag{6}$$

*we have $X_t = (1 \pm 0.5)x_t$ for all $t \in [T]$ with probability at least $1 - \delta_{\mathbb{P}} - T\delta_{\mathbb{P},\xi}$.*

**Remark.** This lemma can be easily generalized to cases where we have multiple induction hypotheses. For example, if we have another process $X'_{t+1} = (1 + \alpha')X'_t + \xi'_{t+1} + Z'_{t+1}$ and we need both $X_t = (1 \pm 0.5)x_t$ and $X'_t = (1 \pm 0.5)x'_t$ for the bounds on $|\xi_{t+1}|, |\xi'_{t+1}|$, $\mathbb{E}[Z_{t+1}^2 \mid \mathcal{F}_t], \mathbb{E}[(Z'_{t+1})^2 \mid \mathcal{F}_t]$ to hold. In this case, the final failure probability will be bounded by $T(\delta_{\mathbb{P},\xi} + \delta_{\mathbb{P},\xi'}) + 2\delta_{\mathbb{P}}$. ♣

**Remark.** If the recurrence relationship is $X_{t+1} \leq (1 + \alpha)X_t + \xi_{t+1} + Z_{t+1}$, and we only want an upper bound, then we can replace $x_0$ with any $x_0^+ \geq x_0$ in (6) and the definition of the deterministic process $(x_t)$. ♣

*Proof.* Let $\tau := \inf\{t \geq 0 : X_t \notin (1 \pm \delta)x_t\}$ and set $\hat{\xi}_{t+1} := \xi_{t+1}\mathbb{1}\{t \leq \tau\}$ and $\hat{Z}_{t+1} := Z_{t+1}\mathbb{1}\{t \leq \tau\}$. Clear that $\tau$ is a stopping time, $\hat{\xi}$ is adapted, and $\hat{Z}$ is still a martingale difference sequence. Moreover, by our hypotheses, we have $|\hat{\xi}_t| \leq (1 + \alpha)^t \Xi$ with probability at least $1 - \delta_{\mathbb{P},\xi}$ and $\hat{Z}_{t+1}$ is conditionally $((1 + \alpha)^t \sigma_Z^2, \theta)$-subweibull. As a result,

$$\left|\sum_{s=1}^{t}(1 + \alpha)^{-s}\hat{\xi}_s\right| \leq \Xi t \leq T\Xi \quad \text{with probability at least } 1 - T\delta_{\mathbb{P},\xi},$$

and by Lemma E.2,

$$\sup_{t \in [T]}\left|\sum_{s=1}^{t}(1 + \alpha)^{-s}Z_s\right| \leq \sigma_Z\sqrt{T \log^{\theta+1}(T/\delta_{\mathbb{P}})} \quad \text{with probability at least } 1 - \delta_{\mathbb{P}}.$$

Hence, for any $\delta_{\mathbb{P}} \in (0, 1)$, if we assume

$$\Xi \lesssim \frac{x_0}{T} \quad \text{and} \quad \sigma_Z^2 \lesssim \frac{x_0^2}{T \log^{\theta+1}(T/\delta_{\mathbb{P}})},$$

then with probability at least $1 - \delta_{\mathbb{P}} - T\delta_{\mathbb{P},\xi}$, we have

$$\left| \sum_{s=1}^{t} (1+\alpha)^{-s} \hat{\xi}_s + \sum_{s=1}^{t} (1+\alpha)^{-s} \hat{Z}_s \right| \leq \frac{x_0}{2}, \quad \forall t \in [T].$$

Recall that

$$X_t = (1+\alpha)^t \left( x_0 + \sum_{s=1}^{t} (1+\alpha)^{-s} \xi_s + \sum_{s=1}^{t} (1+\alpha)^{-s} Z_s. \right) \quad \text{and} \quad x_t = (1+\alpha)^t x_0.$$

Then, we compute

$$\begin{aligned}
\mathbb{P}\left[\exists t \in [T], X_t \notin (1 \pm 0.5)x_t\right] &= \mathbb{P}\left[\exists t \in [T], X_t \notin (1 \pm 0.5)x_t \wedge \tau \leq T\right] \\
&= \mathbb{P}\left[X_\tau \notin (1 \pm 0.5)x_\tau \wedge \tau \leq T\right] \\
&= \mathbb{P}\left[\left| \sum_{s=1}^{\tau} (1+\alpha)^{-s} \xi_s + \sum_{s=1}^{\tau} (1+\alpha)^{-s} Z_s \right| \geq 0.5x_0 \wedge \tau \leq T\right] \\
&= \mathbb{P}\left[\left| \sum_{s=1}^{\tau} (1+\alpha)^{-s} \hat{\xi}_s + \sum_{s=1}^{\tau} (1+\alpha)^{-s} \hat{Z}_s \right| \geq 0.5x_0 \wedge \tau \leq T\right] \\
&\leq \delta_{\mathbb{P}} + T\delta_{\mathbb{P},\xi}.
\end{aligned}$$

$\square$

The above lemmas will be used in Stage 1.1 to estimate the growth rate of the signals. The next lemma considers the case where $\alpha$ is 0 and will be used to show the gap between the largest and the second-largest coordinates can be preserved during Stage 1.1.

**Lemma E.3.** *Suppose that $(X_t)_t$ satisfies*

$$X_{t+1} \leq X_t + \xi_{t+1} + Z_{t+1}, \quad X_0 = x_0 > 0,$$

*where the signal growth rate $\alpha > 0$ and initialization $x_0 > 0$ are given and fixed, $(\xi_t)_t$ is an adapted process, and $(Z_t)_t$ is a martingale difference sequence.*

*Let $T > 0$ and $\delta_{\mathbb{P}} \in (0,1)$ be given. Suppose that there exists some $\delta_{\mathbb{P},\xi} \in (0,1)$ and $\Xi, \sigma_Z > 0$ such that for every $t \leq T$, $|\xi_t| \leq \Xi$ with probability at least $1 - \delta_{\mathbb{P},\xi}$ and $Z_{t+1}$ is conditionally $(\sigma_Z^2, \theta)$-subweibull. Then, we have*

$$\sup_{t \leq T} |X_t - x_0| \leq T\Xi + \sigma_Z \sqrt{T \log^{\theta+1}(T/\delta_{\mathbb{P}})} \quad \text{with probability at least } 1 - T\delta_{\mathbb{P},\xi} - \delta_{\mathbb{P}}.$$

*Proof.* Recursively expand the RHS, and we obtain

$$X_t \leq x_0 + \sum_{s=1}^{t} \xi_s + \sum_{s=1}^{t} Z_s.$$

Clear that

$$\sup_{t \leq T} \left| \sum_{s=1}^{t} \xi_t \right| \leq T\Xi \quad \text{with probability at least } 1 - T\delta_{\mathbb{P},\xi}.$$

Meanwhile, by Lemma E.2, we have

$$\sup_{t \leq T} \left| \sum_{s=1}^{t} \hat{Z}_s \right| \leq \sigma_Z \sqrt{T \log^{\theta+1}(T/\delta_{\mathbb{P}})} \quad \text{with probability at least } 1 - \delta_{\mathbb{P}}.$$

Combine the above bounds and we complete the proof. $\square$

Now, we consider the case where the signal grows at a polynomial instead of linear rate. This lemma will be used in Stage 1.2, where the $L$-th order terms dominate. We will need the following estimations on the corresponding deterministic process. Its proof is deferred to the end of this section.

**Lemma E.4.** *Consider the process* $x_{t+1} = x_t + \alpha x_t^p$ *where* $x_0, \alpha$ *are small positive real numbers and* $p > 1$. *Let* $T$ *be the time* $x_t$ *first goes above* 1. *We have*

$$T \lesssim \frac{1}{(p-1)\alpha x_0^{p-1}} \quad and \quad \sum_{t=0}^{T-1} x_t \lesssim \frac{1}{p\alpha x_0^{p-2}}, \quad if \ \alpha \lesssim x_0^{p-1}/p.$$

**Remark.** This lemma provides upper bounds on the time needed for $x_t$ to grow from $x_0 = o(1)$ to 1 and the sum of $x_t$ in this process. Note that the second upper bound is essentially $Tx_0$. Intuitively, this is because due to the sharp transition behavior of this polynomial system, $x_t \approx x_0$ for most of the time. ♣

**Lemma E.5.** *Let* $(X_t)_t$ *be a non-negative stochastic process satisfying*

$$X_{t+1} \geq X_t + \alpha X_t^p + Z_{t+1} + \xi_{t+1}, \quad X_0 = x_0 > 0, \tag{15}$$

*where* $\alpha > 0$, $(Z_{t+1})_t$ *is a martingale difference sequence, and* $(\xi_t)_t$ *is an adapted process. Let* $\hat{x}_t$ *be the solution to the deterministic recurrence relationship* $\hat{x}_{t+1} = \hat{x}_t + \alpha\hat{x}_t^p, \hat{x}_0 = x_0/2$.

*Let* $\delta_{\mathbb{P}} \in (0, 1)$ *be given and* $T := \inf\{t \geq 0 : X_t \geq 1\}$. *Suppose that there exists* $\Xi, \sigma_Z > 0$ *and* $\delta_{\mathbb{P},\xi} \in (0, 1)$ *such that if* $X_t \geq \hat{x}_t$ *and* $t \leq T$, *we have* $|\xi_t| \leq \Xi X_t$ *with probability at least* $1 - \delta_{\mathbb{P},\xi}$ *and* $Z_{t+1}$ *is conditionally* $(\sigma_Z^2 X_t, \theta)$*-subweibull. Then, if*

$$\alpha \lesssim x_0^{p-1}/p, \quad \Xi \lesssim p\alpha x_0^{p-1}, \quad \sigma_Z^2 \lesssim_\theta \frac{\alpha x_0^p}{\log^{\theta+1}\left(\log(1/x_0)/(\alpha x_0^{p-1}\delta_{\mathbb{P}})\right)},$$

*then with probability at least* $1 - \delta_{\mathbb{P},\xi}/\left(\alpha(x_0/2)^{p-1}\right) - \delta_{\mathbb{P}}$, *we have* $T \lesssim \left(p\alpha(x_0/2)^{p-1}\right)^{-1}$ *and* $X_t \geq \hat{x}_t$ *for all* $t \leq T$.

*Proof.* Note that we can rewrite (15) as $X_{t+1} \geq X_t(1 + \alpha X_t^{p-1}) + \xi_t + Z_t$ and view it as the linear recurrence relationship in Lemma 4.1 with a non-constant growth rate. This suggests defining the counterpart of $(1 + \alpha)^t$ as

$$P_{s,t} := \begin{cases} \prod_{r=s}^{t-1}(1 + \alpha X_r^{p-1}), & t > s, \\ 1, & t = s. \end{cases}$$

Then, we can unroll (15) as

$$\begin{aligned}
X_1 &\geq X_0\left(1 + \alpha X_0^{p-1}\right) + \xi_1 + Z_1, \\
X_2 &\geq \left(X_0\left(1 + \alpha X_0^{p-1}\right) + \xi_1 + Z_1\right)\left(1 + \alpha X_1^{p-1}\right) + \xi_2 + Z_2 \\
&\geq X_0\left(1 + \alpha X_0^{p-1}\right)\left(1 + \alpha X_1^{p-1}\right) + \left(1 + \alpha X_1^{p-1}\right)(\xi_1 + Z_1) + \xi_2 + Z_2 \\
&= X_0 P_{0,2} + P_{1,2}(\xi_1 + Z_1) + \xi_2 + Z_2, \\
X_3 &\geq X_2\left(1 + \alpha X_2^{p-1}\right) + \xi_3 + Z_3 \\
&\geq \left(X_0 P_{0,2} + P_{1,2}(\xi_1 + Z_1) + \xi_2 + Z_2\right)\left(1 + \alpha X_2^{p-1}\right) + \xi_3 + Z_3 \\
&= X_0 P_{0,3} + P_{1,3}(\xi_1 + Z_1) + P_{2,3}(\xi_2 + Z_2) + \xi_3 + Z_3.
\end{aligned}$$

Continue the above expansion, and eventually we obtain

$$X_t \geq X_{t_0} P_{t_0,t} + \sum_{s=t_0}^{t-1} P_{s+1,t}(\xi_{s+1} + Z_{s+1}), \quad \forall t \geq t_0 \geq 0.$$

Since $X$ is non-negative, we have $P_{s,t} \geq 1 > 0$. Hence, we can rewrite the above as

$$P_{t_0,t}^{-1} X_t \geq X_{t_0} + \sum_{s=t_0}^{t-1} P_{t_0,s+1}^{-1}(\xi_{s+1} + Z_{s+1}), \quad \forall t \geq t_0 \geq 0.$$

We wish the repeat the argument in the proof of Lemma 4.1, showing that the last term is smaller than $x_0/2$. Unfortunately, this approach will not work directly. We have only assumed $|\xi_{t+1}| \leq \Xi X_t$ and $\mathbb{E}[Z_{t+1}^2|\mathcal{F}_t] \leq \sigma_Z^2 X_t$. Since $X_t$ can be much larger than $\hat{x}_t$, we cannot directly use our assumption to control the size of noises. On the other hand, note that if $X_t \gg \hat{x}_t$, the induction hypothesis will less likely be violated, so in principle, $X_t \gg \hat{x}_t$ should help us. To "enforce" the $X_t \lesssim \hat{x}_t$ condition, we consider the following recoupling strategy: whenever $X_t \geq 4\hat{x}_t$, we restart $\hat{x}_t$ at $X_t/2$. This recoupling will only increase the value of $\hat{x}_t$, and it ensures $X_t \lesssim \hat{x}_t$ always hold.

We now formalize the above argument. To this end, let $\Phi_t : \mathbb{R}_{>0} \to \mathbb{R}_{>0}$ be the flow map of the recurrence relationship $x_{t+1} = x_t + \alpha x_t^p$. That is, $\Phi_s(x)$ is the value of $x_s$ if $(x_s)_s$ is generated by $x_{t+1} = x_t + \alpha x_t^p$ with $x_0 = x$. Then, we inductively define the following sequences of "deterministic" processes and stopping times:

$$\hat{x}_t^{(0)} = \Phi_t(X_0/2), \qquad \iota^{(1)} = \inf\left\{t \geq 0 \,:\, X_t \geq 4\hat{x}_t^{(0)}\right\},$$

$$\hat{x}_t^{(k)} = \Phi_{t-\iota^{(k)}}\left(X_{\iota^{(k)}}/2\right), \qquad \iota^{(k+1)} = \inf\left\{t > \iota^{(k)} \,:\, X_t \geq 4\hat{x}_t^{(k)}\right\}, \quad \forall k \geq 1.$$

In words, $\iota^{(k)}$ is the time we switch to the $k$th coupling. By construction, By construction, $\hat{x}_t^{(k)}$ in non-decreasing in both $t$ and $k$, $0 =: \iota^{(0)} < \cdots < \iota^{(k)} < \cdots$, and $\hat{x}_{\iota^{(k)}}^{(k)} = X_{\iota^{(k)}}/2 \geq 2\hat{x}_{\iota^{(k)}}^{(k-1)} \geq 2\hat{x}_{\iota^{(k-1)}}^{(k-1)} \geq \cdots \geq 2^{k-1}x_0$. In particular, the last property implies that there are only finitely many couplings before $\hat{x}_t^{(k)}$ reaches any fixed constant.

Then, we abuse notations, redefining

$$\hat{x}_t := \sum_{k=0}^{\infty} \mathbb{1}\left\{\iota^{(k)} \leq t < \iota^{(k+1)}\right\} \hat{x}_t^{(k)}.$$

Clear that at each $t$, only one summand is nonzero. By construction, we always have $X_t \leq 4\hat{x}_t$. Since this $\hat{x}_t$ is no smaller than the original one, it suffices to bound the probability that $X_t \leq \hat{x}_t$ for some $t \leq T$. Note that $X_t \leq \hat{x}_t$ if and only if there exists some $k \in \mathbb{N}_{\geq 0}$ with $t \in \left[\iota^{(k)}, \iota^{(k+1)}\right)$ such that

$$X_{\iota^{(k)}} + \sum_{\iota^{(k)}=0}^{t-1} P_{\iota^{(k)},s+1}^{-1}(\xi_{s+1} + Z_{s+1}) \leq P_{\iota^{(k)},t}^{-1}\hat{x}_t^{(k)}.$$

In addition, note that if $X_s \geq \hat{x}_s$ for all $s < t$, then we have $P_{\iota^{(k)},t}^{-1}\hat{x}_t^{(k)} \leq \hat{x}_{\iota^{(k)}}^{(k)} = X_{\iota^{(k)}}/2$. Therefore,

$$\exists t, X_t \leq \hat{x}_t \;\Rightarrow\; \exists k \in \mathbb{N}_{\geq 0}, t \in \left[\iota^{(k)}, \iota^{(k+1)}\right) \text{ s.t. } \begin{cases} X_s \geq \hat{x}_s, \forall s < t, \\ \displaystyle\sum_{s=\iota^{(k)}}^{t-1} P_{\iota^{(k)},s+1}^{-1}(\xi_{s+1} + Z_{s+1}) \leq -\hat{x}_{\iota^{(k)}}^{(k)}. \end{cases}$$

In other words, it suffices to upper bound the probability that RHS happens before $t$. To this end, we define $\tau := \inf\{t \geq 0 \,:\, X_t \leq \hat{x}_t\}$, $\hat{\xi}_{t+1} = \xi_{t+1}\mathbb{1}\{t < \tau\}$, and $\hat{Z}_{t+1} = Z_{t+1}\mathbb{1}\{t < \tau\}$. Then, we can further rewrite the above as

$$\exists t \leq T, X_t \leq \hat{x}_t$$
$$\Rightarrow \exists k \in \mathbb{N}_{\geq 0}, t \in \left[\iota^{(k)}, \iota^{(k+1)}\right) \text{ s.t. } \left|\sum_{s=\iota^{(k)}}^{(t \wedge T)-1} P_{\iota^{(k)},s+1}^{-1}\hat{\xi}_{s+1}\right| + \left|\sum_{s=\iota^{(k)}}^{(t \wedge T)-1} P_{\iota^{(k)},s+1}^{-1}\hat{Z}_{s+1}\right| \geq \hat{x}_{\iota^{(k)}}^{(k)}.$$

We now estimate the last term as follows. First, for $(\xi_t)_t$, we have $|\hat{\xi}_{t+1}| \leq \Xi X_t \leq 4\Xi\hat{x}_t^{(k)}$ if $t \in [\iota^{(k)}, \iota^{(k+1)})$. Therefore,

$$\left|\sum_{s=\iota^{(k)}}^{(t \wedge T)-1} P_{\iota^{(k)},s+1}^{-1}\hat{\xi}_{s+1}\right| \leq 4\Xi\left|\sum_{s=\iota^{(k)}}^{(t \wedge T)-1} \hat{x}_{s+1}^{(k)}\right| \lesssim \frac{\Xi}{p\alpha[\hat{x}_{\iota^{(k)}}^{(k)}]^{p-2}},$$

where the second inequality comes from Lemma E.4. For the RHS to be smaller than $\hat{x}_{\iota^{(k)}}^{(k)}$, it suffices to require

$$\Xi \lesssim p\alpha[\hat{x}_{\iota^{(k)}}^{(k)}]^{p-1} \quad \Leftrightarrow \quad \Xi \lesssim p\alpha x_0^{p-1}.$$

Also, by Lemma E.4, when the induction hypothesis is true, we have $T \lesssim \left(\alpha x_0^{p-1}\right)^{-1}$. Thus, the above implies that with probability at least $1 - \delta_{\mathbb{P},\xi}/(\alpha x_0^{p-1})$, the total contribution of $(\xi_t)_t$ is small, as long as $\Xi \lesssim p\alpha x_0^{p-1}$.

Then, we consider the martingale difference terms. Note that $P_{\iota^{(k)},t+1}^{-1}\hat{Z}_{t+1}$ is a martingale difference sequence that is conditionally $(4\sigma_Z^2\hat{x}_{\iota^{(k)}}^{(k)},\theta)$-subweibull. Hence, for each $k$, by Lemma E.2, we have

$$\left|\sum_{s=\iota^{(k)}}^{(t\wedge T)-1} P_{\iota^{(k)},s+1}^{-1}\hat{Z}_{s+1}\right| \lesssim_\theta \sqrt{\sigma_Z^2\hat{x}_{\iota^{(k)}}^{(k)} \frac{\log^{\theta+1}\left(\left(\alpha[\hat{x}_{\iota^{(k)}}^{(k)}]^{p-1}\delta_{\mathbb{P},Z}\right)^{-1}\right)}{\alpha[\hat{x}_{\iota^{(k)}}^{(k)}]^{p-1}}}$$

$$\lesssim_\theta \sigma_Z\sqrt{\frac{\log^{\theta+1}\left(1/(\alpha x_0^{p-1}\delta_{\mathbb{P},Z})\right)}{\alpha x_0^{p-2}}},$$

with probability at least probability at least $1 - \delta_{\mathbb{P},Z}$. For the RHS to be smaller than $x_0$, it suffices to require

$$\sigma_Z^2 \lesssim_\theta \frac{\alpha x_0^p}{\log^{\theta+1}\left(1/(\alpha x_0^{p-1}\delta_{\mathbb{P},Z})\right)}.$$

Recall that we recouple at most $O(\log(1/x_0))$ times. Hence, it suffices to replace $\delta_{\mathbb{P},Z}$ with $\delta_{\mathbb{P}}/\log(1/x_0)$ to ensure the total contribution of $(Z_t)_t$ is small. $\qquad\square$

### E.1 Deferred proofs

*Proof of Lemma E.2.* First, consider the case where $\sigma = 1$. Let $M \geq 1$ be a parameter to be chosen later. Then, define $\hat{Z}_t = Z_t\mathbb{1}\{|Z_t| \leq M\}$ and write

$$\sum_{t=1}^T Z_t = \sum_{t=1}^T \left(\hat{Z}_t - \mathbb{E}\left[\hat{Z}_t \mid \mathcal{F}_{t_1}\right]\right) + \sum_{t=1}^T \mathbb{E}\left[\hat{Z}_t \mid \mathcal{F}_{t_1}\right] + \sum_{t=1}^T Z_t\mathbb{1}\{|Z_t| > M\} =: \mathtt{T}_1 + \mathtt{T}_2 + \mathtt{T}_3.$$

Since $Z_t$ is conditionally $(1,\theta)$-subweibull, we have

$$\mathbb{P}(\mathtt{T}_3 \neq 0) \leq \mathbb{P}\left(\exists t \in [T], |Z_t| \geq M\right) \leq CT\exp\left(-M^{1/\theta}\right).$$

For the last term to be bounded by $\delta_{\mathbb{P}}$, it suffices to choose

$$M \geq \log^\theta\left(CT/\delta_{\mathbb{P}}\right).$$

Then, we consider $\mathtt{T}_2$. Since $\mathbb{E}[Z_t \mid \mathcal{F}_{t-1}] = 0$, we have

$$\mathbb{E}\left[\hat{Z}_t \mid \mathcal{F}_{t-1}\right] = \mathbb{E}\left[\hat{Z}_t - Z_t \mid \mathcal{F}_{t-1}\right] = \mathbb{E}\left[Z_t\mathbb{1}\{|Z_t| > M\} \mid \mathcal{F}_{t-1}\right].$$

For the last term, using the layer cake representation, we obtain

$$\begin{aligned}
|\mathbb{E}\left[Z_t\mathbb{1}\{|Z_t| > M\} \mid \mathcal{F}_{t-1}\right]| &\leq \mathbb{E}\left[|Z_t|\mathbb{1}\{|Z_t| \geq M\} \mid \mathcal{F}_{t-1}\right] \\
&= \int_0^\infty \mathbb{P}\left(|Z_t|\mathbb{1}\{|Z_t| \geq M\} \geq s \mid \mathcal{F}_{t-1}\right)\,\mathrm{d}s \\
&= \int_0^\infty \mathbb{P}\left(|Z_t| \geq M \vee s \mid \mathcal{F}_{t-1}\right)\,\mathrm{d}s \\
&= M\,\mathbb{P}\left(|Z_t| \geq M \mid \mathcal{F}_{t-1}\right) + \int_M^\infty \mathbb{P}\left(|Z_t| \geq s \mid \mathcal{F}_{t-1}\right)\,\mathrm{d}s.
\end{aligned}$$

Therefore, for each summand in $\mathsf{T}_2$, we have

$$\left| \mathbb{E}\left[ \hat{Z}_t \mid \mathcal{F}_{t-1} \right] \right| \leq CM \exp\left(-M^{1/\theta}\right) + \int_M^\infty C \exp\left(-s^{1/\theta}\right) \mathrm{d}s$$

$$= CM \exp\left(-M^{1/\theta}\right) + C\sigma \int_{M^{1/\theta}}^\infty e^{-s} s^{\theta-1} \, \mathrm{d}s.$$

Note that if $s/\log s \geq 2(\theta-1)$, we have $e^{-s} s^{\theta-1} = e^{-s+(\theta-1)\log s} \leq e^{-s/2}$. Hence, if we choose $M$ such that

$$\frac{M^{1/\theta}}{\log\left(M^{1/\theta}\right)} \geq 2(\theta-1) \quad \Leftarrow \quad \frac{M}{\log^\theta M} \geq \left(\frac{2(\theta-1)}{\theta}\right)^\theta,$$

then we have

$$C \int_{M^{1/\theta}}^\infty e^{-s} s^{\theta-1} \, \mathrm{d}s \leq C \int_{M^{1/\theta}}^\infty e^{-s/2} \, \mathrm{d}s = 2C \exp\left(-\frac{1}{2} M^{1/\theta}\right).$$

As a result, we have

$$|\mathsf{T}_2| \leq CT \left( M \exp\left(-M^{1/\theta}\right) + 2\exp\left(-\frac{1}{2} M^{1/\theta}\right) \right) \leq 4CMT \exp\left(-\frac{1}{2} M^{1/\theta}\right).$$

Finally, consider $\mathsf{T}_1$. Note that $(\hat{Z}_t - \mathbb{E}[\hat{Z}_t \mid \mathcal{F}_{t-1}])_t$ is a martingale difference that is bounded by $2M$ and has conditional variance bounded by $C_\theta$ for some $C_\theta > 0$. Therefore, by Bernstein's inequality, we have

$$\mathbb{P}(|\mathsf{T}_1| \geq K) \leq 2\exp\left(-\frac{(K/\sqrt{T})^2}{C_\theta + 2M}\right).$$

For the RHS to be bounded by $\delta_{\mathbb{P}}$, it suffices to require

$$K \geq \sqrt{T}\sqrt{(C_\theta + 2M)\log\left(2/\delta_{\mathbb{P}}\right)}$$

Finally, combining the above analysis, we obtain

$$\left| \sum_{t=1}^T Z_t \right| \leq 4CMT \exp\left(-M^{1/\theta}/2\right) + \sqrt{T}\sqrt{(C_\theta + 2M)\log\left(2/\delta_{\mathbb{P}}\right)},$$

with probability at least $1 - 2\delta_{\mathbb{P}}$, where $M \geq \log^\theta\left(CT/\delta_{\mathbb{P}}\right)$ and $M/\log^\theta M \geq \left(\frac{2(\theta-1)}{\theta}\right)^\theta$. Now, we simplify the RHS as follows. Note that

$$4CMT \exp\left(-\frac{1}{2} M^{1/\theta}\right) \leq \sqrt{T}\sqrt{2M\log\left(2/\delta_{\mathbb{P}}\right)}$$

$$\Leftarrow \quad \exp\left(\frac{1}{2}\log M - \frac{1}{2} M^{1/\theta}\right) \leq \frac{\sqrt{2\log\left(2/\delta_{\mathbb{P}}\right)}}{4C\sqrt{T}}$$

$$\Leftarrow \quad \frac{M}{\log^\theta M} \geq 2^\theta, \quad M \geq 4^\theta \log^\theta\left(\frac{8C^2 T}{\log\left(2/\delta_{\mathbb{P}}\right)}\right).$$

In other words, we can choose

$$M = \Theta_\theta\left(\log^\theta(T/\delta_{\mathbb{P}})\right),$$

and obtain

$$\left| \sum_{t=1}^T Z_t \right| \lesssim_\theta \sqrt{T \log^{\theta+1}\left(T/\delta_{\mathbb{P}}\right)}.$$

Finally, for general $\sigma > 0$, it suffices to note that if $X$ is $(\sigma^2, \theta)$-subweibull, then $X/\sigma$ is $(1, \theta)$-subweibull.

$\square$

*Proof of Lemma E.4.* First, we consider the upper bound on $T$. We compute

$$\alpha = \frac{x_{t+1} - x_t}{x_t^p} = \frac{x_{t+1}^p}{x_t^p} \frac{x_{t+1} - x_t}{x_{t+1}^p} = \frac{x_{t+1}^p}{x_t^p} \int_{x_t}^{x_{t+1}} \frac{1}{x_{t+1}^p} \, dy$$

$$\leq \frac{x_{t+1}^p}{x_t^p} \int_{x_t}^{x_{t+1}} \frac{1}{y^p} \, dy = \frac{x_{t+1}^p}{x_t^p} \frac{1}{p-1} \left( \frac{1}{x_t^{p-1}} - \frac{1}{x_{t+1}^{p-1}} \right).$$

In addition, note that $x_{t+1}^p / x_t^p = \left( 1 + \alpha x_t^{p-1} \right)^p \leq (1 + \alpha)^p \leq e^{\alpha p}$. Therefore,

$$\alpha \leq \frac{e^{\alpha p}}{p-1} \left( \frac{1}{x_t^{p-1}} - \frac{1}{x_{t+1}^{p-1}} \right) \quad \Rightarrow \quad \frac{1}{x_{t+1}^{p-1}} \leq \frac{1}{x_t^{p-1}} - \frac{(p-1)\alpha}{e^{\alpha p}}.$$

Sum both sides from 0 to $t-1$ and we get

$$\frac{1}{x_t^{p-1}} \leq \frac{1}{x_0^{p-1}} - \frac{t(p-1)\alpha}{e^{\alpha p}} \quad \Rightarrow \quad x_t \geq \left( \frac{1}{x_0^{p-1}} - e^{-\alpha p}(p-1)\alpha t \right)^{-\frac{1}{p-1}}.$$

In particular, this implies

$$T \leq \left( \frac{1}{x_0^{p-1}} - 1 \right) \frac{e^{\alpha p}}{(p-1)\alpha}.$$

Now, we consider the upper bound on $\sum_t x_t$. Let $(\tilde{x}_h)_h$ be the solution to the continuous-time ODE $\frac{d}{dt}\tilde{x}_h = \tilde{x}_h^p$ with $\tilde{x}_0 = x_0$. Note that $\tilde{x}$ is increasing and therefore

$$\tilde{x}_{(t+1)\alpha} = \tilde{x}_{t\alpha} + \int_0^\alpha \tilde{x}_{t\alpha+r}^p \, dr \geq \tilde{x}_{t\alpha} + \alpha a \tilde{x}_{t\alpha}^p.$$

Hence, by induction, we have $\tilde{x}_{t\alpha} \geq x_t$ for all $t$. In addition, $\tilde{x}_h$ has the closed-form formula:

$$\tilde{x}_h = \left( \frac{1}{x_0^{p-1}} - (p-1)h \right)^{-\frac{1}{p-1}}.$$

Thus,

$$\sum_{t=0}^{T-1} x_t \leq \sum_{t=0}^{T-1} \tilde{x}_{t\alpha} \leq \alpha^{-1} \sum_{t=0}^{T-1} \int_{t\alpha}^{(t+1)\alpha} \tilde{x}_s \, ds = \frac{1}{\alpha} \int_0^{T\alpha} \tilde{x}_h \, dh$$

$$= \frac{1}{\alpha} \int_0^{T\alpha} \left( \frac{1}{x_0^{p-1}} - (p-1)h \right)^{-\frac{1}{p-1}} \, dh.$$

When $p = 2$, we have

$$\sum_{t=0}^{T-1} x_t \leq \frac{1}{\alpha} \int_0^{T\alpha} \left( \frac{1}{x_0} - h \right)^{-1} \, dh = \frac{1}{\alpha} \log \left( \frac{1}{1 - x_0 T\alpha} \right) \leq \frac{1}{\alpha} \log \left( \frac{1}{1 - e^{2\alpha} + x_0 e^{2\alpha}} \right) \leq \frac{2}{\alpha},$$

as long as $\alpha \lesssim x_0$ so that $\tilde{x}_{T\alpha} = O(1)$. When $p > 2$, to have $\tilde{x}_{T\alpha} \leq 2$, it suffices to have

$$\frac{1}{x_0^{p-1}} - (p-1)\alpha T \geq \frac{1}{2^{p-1}} \quad \Leftarrow \quad e^{\alpha p} - \frac{e^{\alpha p} - 1}{x_0^{p-1}} \geq \frac{1}{2^{p-1}} \quad \Leftarrow \quad \alpha \lesssim x_0^{p-1}/p.$$

Let $\tilde{T}$ be the time $\tilde{x}_h$ reaches 2. We have $\tilde{T} \le \frac{1}{(p-1)x_0^{p-1}}$ and

$$
\begin{aligned}
\sum_{t=0}^{T-1} x_t &\le \frac{1}{\alpha} \int_0^{\tilde{T}} \left( \frac{1}{x_0^{p-1}} - (p-1)h \right)^{-\frac{1}{p-1}} \mathrm{d}h \\
&= \frac{1}{\alpha} \frac{1}{p-2} \left( \frac{1}{x_0^{p-1}} - (p-1)\tilde{T} \right)^{-\frac{1}{p-1}} \\
&\quad \times \left( (p-1)\tilde{T} + \frac{1}{x_0^{p-1}} \left( -1 + \left( \frac{1}{x_0^{p-1}} \right)^{-\frac{1}{p-1}} \left( \frac{1}{x_0^{p-1}} - (p-1)T \right)^{\frac{1}{p-1}} \right) \right) \\
&\lesssim \frac{1}{(p-2)\alpha} \frac{1}{x_0^{p-2}}.
\end{aligned}
$$

$\square$

## F   Simulation

We include simulation results for Stage 1 in this section. The goal here is to provide empirical evidence that (i) if we have both the second- and $L$-th order terms, then the sample complexity of online SGD scales linearly with $d$ and (ii) without the higher-order terms, online SGD cannot recovery the exact directions.

The setting is the same as the one we have described in Section 2. We choose the hyperparameters roughly according to Theorem 2.1. To reduce the demand of computational resources, we choose $m = \Theta(P^2)$ instead of $\tilde{\Omega}(P^8)$. Note that by the Coupon Collector problem, we need $m = \Omega(P \log P)$ to ensure that for each $p \in [P]$, there exists at least one neuron $\boldsymbol{v}$ with $v_p^2 \ge \max_{q \le P} v_q^2$. Since we are mostly interested in the dependence on $d$, for the learning rate, we choose $\eta = c/d$, where $c$ is a tunable constant that is independent of $d$ but can depend on everything else. $T$ is chosen according to Theorem 2.1 and we early-stop the training when for all $p \in [P]$, there exists a neuron with $v_p^2 \ge 0.95$ (in the moving average sense).

All experiments are performed on the authors' laptop without using GPUs, and it takes less than one day to complete the experiments.

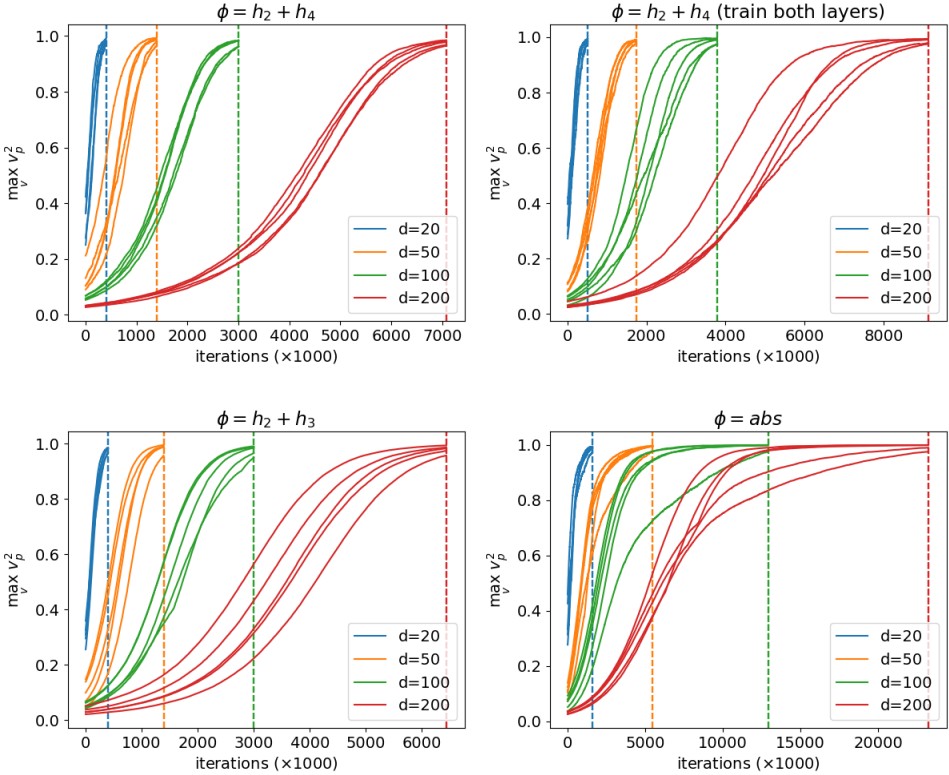

Figure 1: Recovery of directions. The above plots show the evolution of the correlation with each of the ground-truth directions. We fix the relevant dimension $P = 5$ and vary the ambient dimension $d$. Different colors represent different $d$. For each color, one curve represents $\max_{\boldsymbol{v}} v_p^2$ for one $p \in [P]$. In the first row, the link function is $\phi = h_2 + h_4$. In the left plot, we use the algorithm (1), while in the right plot, we train both layers simultaneously. The second row contains simulation results for other link functions.

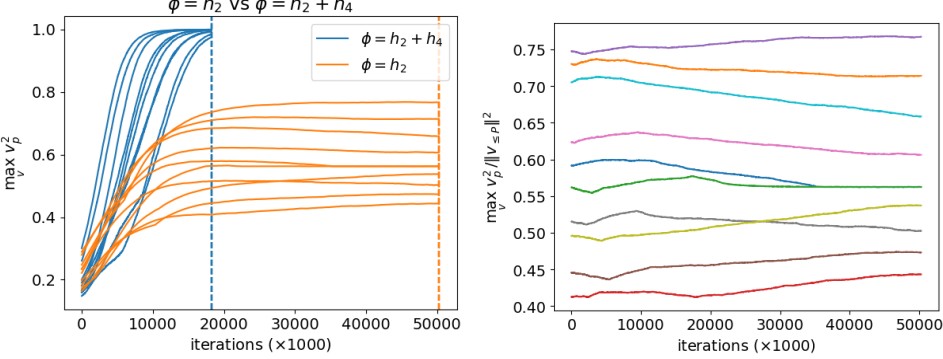

Figure 2: Necessity of the higher order terms. In these two figures, we choose $P = 10$ and $d = 100$. The left plot shows the maximum correlation each of the ground-truth directions (also see Figure 1). We can see that in the isotropic case, whether online SGD can recover the ground-truth directions is determined by the presence/absence of the higher-order terms. The right plot shows the change of $\max_{\boldsymbol{v}} v_p^2 / \|\boldsymbol{v}_{\leq P}\|^2$ for each $p \in [P]$ in Stage 1 when the link function is $h_2$. One can observe that they are almost unchanged throughout training. This, together with the left plot, shows that the increase of the correlation is caused by learning the subspace instead of the actual directions.

