# OpenReview forum: "Learning Orthogonal Multi-Index Models: A Fine-Grained Information Exponent Analysis"
_NeurIPS.cc/2025/Conference — NeurIPS 2025 poster_

### Official Review · Reviewer_9mhx · 2025-06-30

**Clarity:** 3
**Significance:** 2
**Originality:** 3
**Rating:** 4
**Confidence:** 2

**Summary:**

PAPER SUMMARY:
The information exponent (IE) of a multi-index model $f(x) = \sum_i \phi(v_i \cdot x)$ is the order of the first nonzero term in the Hermite expansion of $\phi$ around $0$.
The IE and its generalization, the generative exponent (GE), provide characterizations for the sample complexity of learning multi-index models via SGD. In particular, roughly $d^{IE -1}$ samples are needed if $IE \geq 3$.

When $IE = 2$, rotational invariance makes it impossible to recover the exact orthonormal vectors $\{v_i\}_i$ and only $Span(\{v_i\}_i)$ can be recovered. This paper investigates the statistical complexity of recovering all the orthonormal vectors $\{v_i\}_i$ as a function of $L$, where $IE = 2$, and $L$ is the the next non-zero term in the Hermite expansion of $\phi$.

This work proves that $O(PdP^{L-2})$ samples are sufficient for recovering all vectors $\{v_i\}_i$.
Here is the catch: even if it does not allow recovery of all the vectors $\{v_i\}_i$, having a non-zero second-order term is advantageous, as $d^{L-1} \gg PdP^{L-2}$ samples would be required if the second-order term was null. In particular, the IE does not tell the full statistical complexity story when $IE = 2$.

**Questions:**

I probably missed something, but I am not sure why you write $PdP^{L-2}$ rather than $dP^{L-1}$.

**Ethical Concerns:**

["NO or VERY MINOR ethics concerns only"]

**Final Justification:**

I confirm my evaluation and recommend acceptance.

**Limitations:**

Yes.

**Paper Formatting Concerns:**

I confirm my evaluation and recommend acceptance, as long as there is room.

**Quality:**

3

**Strengths And Weaknesses:**

I believe that the authors could have done a better job at articulating why this particular problem should be interesting for ML theorists. Overall, it seems like the studied phenomenon manifests itself in the particular case $IE = 2$, when we want to recover the exact vectors $\{v_i\}_i$, and does not add much to the bigger picture of multi-index models.

This work claims that "focusing solely on the lowest degree can miss key structural details of the model and result in suboptimal rates".
It seems to me that the main message of this work is that, for the particular case of $IE = 2$, looking at the next non-zero term is informative. This does not detract from the way more general statistical bound for $IE \geq 3$, and does not add much depth to that line of work.

---

> ### Author Rebuttal · Authors · 2025-07-30
>
> Thank you for the reviews! We'd like to address your concerns as follows.
>
> * **On the scope of the paper.** We focus mainly on the case where $\mathrm{IE}=2$ because $\mathrm{IE}=2$ is special
>   in the following ways: (1) the IE of even functions is at least $2$, (2) most link functions/activations have IE
>   at most $2$, and (2') the IE of many functions, including all finite degree polynomials, can be reduced to $2$ by a
>   proper label transformation.
>
>   Moreover, we believe our *hierarchical* feature learning analysis can potentially shed some light on the much harder non-orthogonal
>   version of this problem.  In the non-orthogonal case, the minimizers of the correlation loss, truncated at any finite order, are not aligned with the ground truth directions.
>   However, as the order increases, they will become localized around the ground-truth directions, as the influence decay
>   faster (for example, the "infinite-order" terms are nonzero only if the learner neuron is fully aligned with a ground
>   truth direction). Intuitively, one might be able to use the lower order terms
>   first to obtain warm starts, and then gradually move to higher order terms to move toward the ground truth
>   directions. Formalizing this or finding conditions under which this is possible is challenging, and the orthogonal
>   $h_2 + h_L + \cdots$ setting considered in this paper provides a simple example where this type of argument is
>   doable.
>
>   Finally, our results can be (partially) extended to other settings as well (polynomially bounded link functions,
>   including linear terms, non-isotropic, and/or misspecified). You could check the responses to Reviewer 4V3d, x9Fa and uJL9 for details.
> * **$P d P^{L-2}$ vs $d P^{L-1}$.** We write $P d P^{L-2}$ instead of $d P^{L-1}$ to emphasize the different sources of
>   $P$ and $P^{L-1}$. The first $P$ comes from the fact we have $P$ different directions, while the second part
>   $d P^{L-1}$ comes from learning a single-index model in a $P$-dimensional subspace with noises scaling with $d$.

---

### Official Review · Reviewer_uJL9 · 2025-07-02

**Clarity:** 3
**Significance:** 2
**Originality:** 2
**Rating:** 4
**Confidence:** 4

**Summary:**

The authors propose to understand the sample complexity of learning a multivariate function of the form $x \to \sum_{i = 1}^P \phi(v_k \cdot x)$. The input $x \sim N(0,1)$ and the learning is performed via a modified version of "SGD" with a two-layer neural network.

In the article, the information exponent is $2$, but the aim of the paper is to show that while this first order term $h_2$ has a rotational invariance that prevent the $v_k$ to be retrieved (only the span of it), higher order terms can help having a perfect recovery, without paying any cost.

The main theorem is presented in Section 2, while the sketch of the proof is divided between the two subsequent parts : the gradient flow study and then the discretization via SGD.

**Questions:**

The aim of the paper is to show that the network is able to retrieve the $v_k$, and not only the span of it. In the theorem, it is not clear to me that it is the case since it is stated that the population loss is small. I understand that it could be that since there are higher order terms, this implies that the $v_k$ are retrieved but this needs to be clarified

I have trouble with the claim that *the information/generative exponent is insufficient* ? For me it still appears as the integer driving the complexity of the dynamics?

**Ethical Concerns:**

["NO or VERY MINOR ethics concerns only"]

**Final Justification:**

I have read the answer of the authors. It is concise and precise and I understand the goal of the paper perfectly. Due to the limitation of the scope of the result, I am not willing to raise my score, however, I tend to a 4+.

**Limitations:**

Already discussed

**Quality:**

3

**Strengths And Weaknesses:**

Strengths:

- the paper seems technically very solid, with clear sketch of proofs and a concrete goal. Analyzing sample complexities in such tasks if often quite technical
- the aim of the paper is clear and it is overall well written

Weaknesses:
- the scope of the paper is quite narrow : it deals with a specific problem in the multi-index model (a separable sum with the same link function), and the link function has a very specific structure : $h_2 + h_L + \dots$.
- every thing is sort of tailored for the result to hold : the neural network is adapted to the problem in the sense that the link function which could be a priori unknown is the non-linearity of the Neural Network. The algorithm is also very specific : freezing layers after the other ones. I understand these assumptions and algorithms are abundant in the literature, yet I have to signal this as a weakness.

---

> ### Author Rebuttal · Authors · 2025-07-30
>
> Thank you for the reviews! We'd like to address your concerns as follows.
>
> * **Recovering the ground truth directions.** We show in Lemma B.8 that at the end of Stage 1.2, every ground truth
>   direction will be recovered by a neuron, in the sense that the correlation is at least $1 - \epsilon$. The only
>   thing Stage 2 does is picking out these neurons, or, more accurately, find a combination of the neurons that is
>   at least as good as the one consists of these neurons. If one really wants to recover the directions explicitly,
>   one may add a $l^1$ regularizer $\lambda \sum_{k=1}^m |a_k|$ and do lasso regression, which returns a sparse solution.
>   Alternatively, this step can also be done with $l^1$-regularized SGD via a local convergence argument similar to
>   the one in [Chi20].
> * **On the scope of the paper.** Our results can be extended to the misspecified case, where the link functions along
>   different directions are different and unknown to the learner, as long as the activation of the learner has some
>   universal approximation property, and the target link functions still have isotropic 2nd order terms, and the
>   condition number of the $L$-th order terms is constant. This is similar to the situation in [BBSS22]. Recovering
>   the subspace/directions does not require the activation to be exactly the same as the target link function, and
>   after learning the directions, fitting the target link function can be done with a one-dimensional random feature
>   argument.
>
>   Our results can be (partially) extended to other settings as well (polynomially bounded link functions,
>   including linear terms, and/or non-isotropic). You could check the responses to Reviewer 4V3d and x9Fa for details.
> * **Issue of IE/GE.** We are not claiming that the IE/GE theory is wrong. Instead, our point is that
>   the vanilla IE/GE theory, which considers only the lowest order, is insufficient to characterize the richer behavior
>   of multi-index models. For example, we cannot distinguish between $h_2$ and $h_2 + h_4$ when looking only at the
>   lowest order. On the other hand, SGD's behaviors for these target functions are different, as directional convergence
>   is possible only in the 2nd case. We propose to look beyond the lowest order and show that, in the setting considered
>   in this work, this leads to a sharper characterization.
>
>   In addition, we believe this point of view might shed some light on the much harder non-orthogonal setting.
>   In this case, the minimizers of the correlation loss, truncated at any finite order, are not aligned with the
>   ground truth directions. However, as the order increases, they will become localized around the ground-truth
>   directions, as the influence decay faster (for example, the "infinite-order" terms are nonzero only if the learner
>   neuron is fully aligned with a ground truth direction). Intuitively, one might be able to use the lower order terms
>   first to obtain warm starts, and then gradually move to higher order terms to move toward the ground truth
>   directions. Formalizing this or finding conditions under which this is possible is challenging, and the orthogonal
>   $h_2 + h_L + \cdots$ setting considered in this paper provides a simple example where this type of argument is
>   doable.
>
>
> [Chi20] Lenaic Chizat. Sparse Optimization on Measures with Over-parameterized Gradient Descent. 2020.
>
> [BBSS22] Alberto Bietti, Joan Bruna, Clayton Sanford, Min Jae Song. Learning Single-Index Models with Shallow Neural Networks. 2022.

---

> > ### Comment · Reviewer_uJL9 · 2025-08-05
> > **Answer after rebuttal**
> >
> > I have read the answer of the authors. It is concise and precise and I understand the goal of the paper perfectly. Due to the limitation of the scope of the result, I am not willing to raise my score, however, I tend to a 4+.

---

### Official Review · Reviewer_x9Fa · 2025-07-06

**Clarity:** 3
**Significance:** 3
**Originality:** 3
**Rating:** 5
**Confidence:** 3

**Summary:**

This paper studied the problem of learning orthogonal multi-index models under Gaussian distribution. The paper assumes the multi-index model takes the specific form of $f^\ast(x) = \sum_{k=1}^P \phi(v_k^\ast\cdot x)$, where $\phi$ are non-linear activations that have the form of $\phi=he_2 +\sum_{l\geq L}\hat{\phi}_l he_l$. The authors showed that, using online sgd algorithm, using $dP^{O(L)}$ samples suffices to learn the target $f^\ast$ to $\epsilon$-$L_2^2$ error. The authors provided a set of frameworks and tools for the analysis of sgd dynamics on such objectives, and provided a white-box, detailed analysis of the learning algorithm.

**Questions:**

See weakness

**Ethical Concerns:**

["NO or VERY MINOR ethics concerns only"]

**Final Justification:**

I think the authors cleared my concersn. I maintain my evaluation.

**Limitations:**

no negative limitations

**Paper Formatting Concerns:**

no concerns

**Quality:**

3

**Strengths And Weaknesses:**

Strengths
1. I think the paper conducted a thorough study of an interesting special case of multi-index models. The paper is solid in techniques.
2. I think the characterization of the sample complexity, in general, is not surprising, as similar (qualitatively) results have been derived in prior literature like [DLS22]. However, I think it is still an interesting contribution to obtain such a result for the special case.

Weakness
1. I think the condition that $P\geq \log^C(d)$ is a weird condition that seemly most likely to be a side-effect of the analysis. Is it possible to remove this by refining your arguments?
2. In [DLS22], the authors studied learning MIMs of the form $g(v^\ast_1\cdot x,\dots,v^\ast_p\cdot x)$ where $g$ is a polynomial of bounded degree, using two layer neural nets. Though [DLS22] has sample complexity $d^2 P + d P^L$ (consider $g$ being a degree $L$ polynomial), this paper considers a much simpler form of the multi-index model. I wonder if it is possible to relax the orthogonality condition to the condition of [DLS22], i.e., $E[\nabla^2 f^*(x)]$ has rank $P$. Otherwise, it seems the major contribution of this paper is to improve the degree of $d^2$ to $d$, with the sacrifice of model generality.

---

> ### Author Rebuttal · Authors · 2025-07-30
>
> Thank you for your feedback! We'd like to address your concerns as follows.
>
> * **The condition on $P$.** The condition $P \ge \log^C d$ is mostly an artifact of our analysis. First, we
>   want to ensure certain concentration properties hold in the relevant subspace (such as the norm in the relevant
>   subspace). We do not want to explicitly track the failure probability of these events, so we assume $P \ge \log^C d$,
>   so that the failure is inverse super-polynomially small. In addition, this also creates a cleaner separation between
>   the two stages. For example, at the end of the first stage, the 2nd order terms have order $\| v\_{\le P} \|\_2^2 \approx 1$,
>   while the higher order terms have order $\| v\_{\le P} \|\_L^L$. The higher order terms are much smaller than the 2nd
>   order terms when $P \gg 1$, and therefore can be safely ignored. We believe that this condition can be removed at the
>   cost of a messier proof.
> * **Comparison with [DLS22].** First, we wish to stress that our sample complexity of the second stage is also tighter
>   than the one in [DLS22]. They use a random feature argument in Stage 2, which leads to the sample complexity $d r^q$,
>   where $q$ is the degree of the polynomial. In our case, we show that SGD can do feature learning, and the sample
>   complexity is $d r^L$, where $L$ is the information exponent, which can be much smaller than the degree. In addition,
>   our results can be extended to any link functions with a polynomially bounded derivative (see the response to
>   Reviewer 4V3d), while their results only deal with polynomials. Moreover, their width complexity is $P^q$ and ours is
>   $P \log P$ (cf. the statement of Theorem 2.1 in Appendix D). In some sense, [DLS22] shows that SGD can learn the
>   subspace using $d^2$ samples, while our results show that, at least in a more restricted but still non-trivial
>   setting, SGD can automatically do *hierarchical* feature learning, which yields sample/width complexities that are
>   (almost) tight, in the sense that they match the complexities of separately learning the subspace, and then
>   independently learning $P$ single-index models in the relevant subspace.
> * **Relaxing the orthonormality condition.** First, we believe that our results can be extended to the near orthogonal
>   case. However, we expect the non-orthogonal case to be hard, as it is known that non-orthogonal tensor decomposition
>   is hard in general, whence we cannot expect SGD to learn the directions with polynomial samples/iterations,
>   even when the relevant subspace is given.
>
>   Things are trickier if we consider only the 2nd order terms (which is always "orthogonal" because of the spectral
>   theorem), but do not assume isotropicity. In this case, correlation loss will fail, since all neurons will simply
>   converge to the largest eigendirection (similar to what will happen in the matrix power iteration). This can be
>   fixed by using the MSE loss and an initialization whose scale matches the target's. Our analysis does not directly
>   apply to this case due to the interaction between the learner neurons, but the behaviors should be similar.
>   That is, the 2nd order terms drive the distribution of learner neurons to (close to) the Gaussian in the relevant
>   subspace, with the correct covariance matrix, and then the higher order terms become non-negligible and allow the
>   neurons to recover the exact directions.

---

> ### Comment · Reviewer_x9Fa · 2025-08-08
>
> I thank the authors for their response and I do not have further questions. I would like to maintain my score.

---

### Official Review · Reviewer_4V3d · 2025-07-07

**Clarity:** 3
**Significance:** 3
**Originality:** 3
**Rating:** 5
**Confidence:** 3

**Summary:**

The paper analyzes the sample complexity of learning multi-index models of the form $\sum_{k=1}^P \phi(v^\star_k . x)$ with orthogonal $v^\star_k$  through online-SGD on a two-layer neural network model. Unlike existing works focusing on the recovery of the target subspace and subsequent learning of the target non-linearity, the main results show that with sufficient overparameterization, the neurons of the trained model exactly recover individual target directions $v^\star_k$ with $\mathcal{O}(P dP^{L-2})$ sample-complexity, where $L$ denotes the information exponent. Crucially, the sample-complexity for recovering individual directions remains linear in the dimension. The exact recovery of directions proceeds through higher-order terms in the gradient dynamics, which amplify the neurons' initial difference in overlaps across different directions.

**Questions:**

* What factors does the polynomial in the Poly(P) sample-complexity depend on? Can it be well-specified?
* Can assumption 1(c) be extended to targets with polynomially-bounded derivatives?
* where does the diverging width requirement $ \log d^c < P$ arise from?
* what would happen if Assumption 1(a) is relaxed to allow a linear-component in the target non-linearity?

Including the above clarifications would strengthen the paper.

**Ethical Concerns:**

["NO or VERY MINOR ethics concerns only"]

**Final Justification:**

I recommend acceptance based on the strengths highlighted in my review, in particular the insights gained from the analysis.

**Limitations:**

The work is primarily theoretical and does not pose direct societal risks. The paper would benefit from a more detailed discussion of the limitations of their model listed under "weaknesses".

**Paper Formatting Concerns:**

I have not noticed any major formatting issues in the paper.

**Quality:**

3

**Strengths And Weaknesses:**

# Strengths:
- The paper is well-written and organized with adequate discussion of prior work.
- The analysis is insightful. In particular, the insights into how higher-order information in the gradients leads to the exact recovery of individual directions are expected to carry over to general setups, such as while jointly training all the layers, or for deeper networks.
- The proofs are simple and the paper provides readable proof-sketches explaining the key ideas.

# Weaknesses
- The width-complexity or overparameterization in the main result is specified as Poly(P). A precise description of this polynomial dependence would greatly strengthen the results.
- The paper suffers from the limitations of using correlation loss (which prevents interactions between neurons). However, since this simplifying assumption is used by several related works in the literature, I consider it a justifiable starting point.
- Another related limitation is the absence of a linear component in the target non-linearity. With correlation loss, this presumably leads to a spike from the linear components dominating the dynamics. A discussion of possible directions to relax such limitations would strengthen the work.

---

> ### Author Rebuttal · Authors · 2025-07-30
>
> Thank you for the reviews and suggestions! We'd like to answer your questions as follows.
>
> * **Width complexity.** We realized that the width can be as small as $O(P \log P)$ after submitting the main text. This
>   is tight up to the logarithmic term and matches the coupon collector bound. We have included this in the appendix.
>   See Lemma A.7 and the statement of the main theorem in Appendix D.
> * **Relaxing Assumption 1(c).** Thank you this question! It makes us realize that polynomial boundedness is sufficient.
>   Assumption 1(c) is only used to get a tail bound on the per-sample gradient, which amounts to controlling the tail of
>   $f_* = \sum_p \phi(v_p^* \cdot x)$ (times the derivative of a single neuron). We can use the polynomial boundedness
>   to bound the $L^p$ norm of each $\phi(v_p^* \cdot x)$ and then show that they are sub-Weibull.
>   Since $v_p^\*$ are orthogonal, $\phi(v_p^\* \cdot x)$ are independent, and therefore, the sum of them is still
>   sub-Weibull and this leads to the desired tail bound. We will add this into the revision!
> * **The diverging width assumption.** The condition $P \ge \log^C d$ is mostly an artifact of our analysis. First, we
>   want to ensure certain concentration properties hold in the relevant subspace (such as the norm in the relevant
>   subspace). We do not want to explicitly track the failure probability of these events, so we assume $P \ge \log^C d$,
>   so that the failure is inverse super-polynomially small. In addition, this also creates a cleaner separation between
>   the two stages. For example, at the end of the first stage, the 2nd order terms have order $\| v\_{\le P} \|\_2^2 \approx 1$,
>   while the higher order terms have order $\| v\_{\le P} \|\_L^L$. The higher order terms are much smaller than the 2nd
>   order terms when $P \gg 1$, and therefore can be safely ignored. We believe that this condition can be removed at the
>   cost of a messier proof.
> * **Allowing linear terms.** First, we wish to stress that we consider the case IE=2 since it is the lowest possible when
>   the link function is even, or equivalently, both $v_k^\*$ and $-v_k^\*$ appear in the teacher network. Now, back to
>   the question of the influence of linear components. You are right that there will be a spike (which is the all 1
>   direction if we assume w.l.o.g. that the targets are $\{ e_p \}_p$) and the all neurons will be attracted by that
>   direction.
>
>   To fix this issue, the easiest approach is to first train one neuron to fit that direction to remove it.
>   Due to the linearity, learning the linear term is roughly equivalent to learning a single-index model with IE=1,
>   and it costs only $O(d)$ samples.
>
>   The more natural but technically more challenging approach is to use MSE loss. Similar to the 2nd vs 4th order terms
>   argument, one should be able to show that the first order terms will get learned before the higher terms become large.
>   The tricky thing here is that after learning the first order term, the learner distribution will be shifted to be
>   centered at the all-one point (assuming $v_k^* = e_p$ for simplicity), instead of one single neuron learning that
>   direction with all other neurons staying close to the initialization, as the gradient of the linear term depends only
>   on the average of the neurons and not on any individual neuron. The analysis become tricky after that, since we can
>   no longer ignore the interaction between learner neurons, though in principle, a finite-width mean-field type analysis
>   similar to the one in [LMZ20] can potentially be used to analyze the dynamics of neurons in those later stages.

---

> > ### Comment · Reviewer_4V3d · 2025-08-07
> >
> > I thank the authors for their clear responses and for describing the improvements. I maintain my score and recommend acceptance.

---

### Decision · Program_Chairs · 2025-09-17

**Decision:**

Accept (poster)

**Comment:**

The paper analyzes the sample complexity of learning multi-index models of the form    $\sum_{k=1}^P\phi(v_k^{*}\cdot x)$

with orthogonal directions  $v_k^{*}$,  through online-SGD on a two-layer neural network model. The authors demonstrate that information/generative exponent alone is insufficient to characterize certain structures in the learning task and show that, in this specific orthogonal setting, by considering both second and higher-order terms, the sample complexity of directional recovery using online SGD can be improved over the vanilla information exponent-based analysis.

The paper has ratings: Accept, Accept, Borderline Accept, Borderline Accept.

Strengths: Reviewers all appreciate the technical contributions of the paper and agree that the presented theoretical analysis is insightful.

Weaknesses: Reviewer uJL9 is concerned that the scope of the paper is limited, since it deals with a specific problem in the multi-index model (a separable sum with the same link function), and the link function has a very specific structure. Nevertheless, Reviewer uJL9 still leans towards acceptance.

Base on these reasons, the final recommendation is: Accept

Note: I agree with Reviewer 9mhx that writing $PdP^{L-2}$ instead of $dP^{L-1}$ can be potentially confusing to readers.